# Approximating 3D bedrock deformation in an Antarctic ice-sheet model for projections

Caroline J. van Calcar[1,2], Pippa L. Whitehouse[3], Roderik S.W. van de Wal[1,4] & Wouter van der Wal[2]

[1]Institute for Marine and Atmospheric research Utrecht, Utrecht University, Utrecht, 3508 TA, The Netherlands
[2]Faculty of Aerospace Engineering, Delft University of Technology, Delft, 2629 HS, The Netherlands
[3]Department of Geography, Durham University, Durham, UK
[4]Department of Physical Geography, Utrecht University, Utrecht, 3584 CB, The Netherlands

*Correspondence to*: Caroline J. van Calcar (c.j.vancalcar@tudelft.nl)

**Abstract.** The bedrock deformation in response to a melting ice sheet provides a negative feedback on ice mass loss. When modelling the future behaviour of the Antarctic Ice Sheet, the impact of bed deformation on ice dynamics varies but can reduce projections of future sea-level rise by up to 40% in comparison with scenarios that assume a rigid Earth. The rate of the solid Earth response is mainly dependent on the viscosity of the Earth's mantle, which varies laterally and radially with several orders of magnitude across Antarctica. Because modelling the response for a varying viscosity is computationally expensive
and has only recently been shown to be necessary over centennial time scales, sea-level projection ensembles often exclude the Earth's response or apply a globally constant relaxation time or viscosity. We use a coupled model to investigate the accuracy of various approaches to modelling the bedrock deformation to ice load change. Specifically, we compare the sea-level projections from an ice-sheet model coupled to (i) an elastic lithosphere, relaxed asthenosphere (ELRA) model, with either uniform and laterally varying relaxation times, (ii) a glacial isostatic adjustment (GIA) model with a radially varying
Earth structure (1D GIA model), and (iii) a GIA model with laterally varying earth structures (3D GIA model). Furthermore, using the 3D GIA model we determine a relation between relaxation time and viscosity which can be used in ELRA and 1D models. We conduct 500 year projections of Antarctic Ice Sheet evolution using two different climate models and two emissions scenarios: the high emission scenario SSP5-8.5 and the low emission scenario SSP1-2.6. Using a rigid Earth model, this results in ~3-7.5 m of barystatic sea-level rise with significant retreat in various basins due to marine ice sheet instability.
The results show that using a uniform relaxation time of 300 years in an ELRA model leads to a total sea-level rise that deviates less than 40 cm (6%) from the average of the 3D GIA models in 2500. This difference in the projected sea-level rise can be further reduced to 20 cm (4%) by using an upper mantle viscosity of $10^{19}$ Pa·s in the 1D GIA model, and to 10 cm (2%) in 2500 by using a laterally varying relaxation time map in an ELRA model. Our results show that the Antarctic Ice Sheet contribution to sea-level rise can be approximated sufficiently accurate using ELRA or a 1D GIA model when the
recommended parameters derived from the full 3D GIA model are used.

# 1 Introduction

The Antarctic Ice Sheet (AIS) might contribute several meters to global mean sea-level rise by the year 2300 (Fox-Kemper et al., 2021; Coulon et al., 2024; Klose et al., 2024; Seroussi et al., 2024). The rate of ice loss is influenced by many processes such as atmospheric and oceanic processes, and ice dynamics, which can lead to an uncertainty in sea-level change of up to 1.5 meters in 2300 (Seroussi et al., 2024). Accurately representing these processes in models, along with their associated uncertainties, presents a significant challenge for projecting the ice sheet evolution. To address this, a wide range of parameters in the ice-sheet model must be explored, requiring ensemble simulations to produce robust projections of potential sea-level rise over the coming centuries (Seroussi et al., 2020).

One of the main uncertainties in projecting the evolution of the AIS over the next centuries is the response of the solid Earth to future changes in ice mass (Fox-Kemper et al., 2021). The bedrock experiences uplift due to the loss of ice mass at the surface, a process known as glacial isostatic adjustment (GIA). The bedrock uplift can delay grounding line retreat and thereby stabilize the ice sheet (Gomez et al., 2015; Whitehouse et al., 2019). The rate of the bedrock uplift depends on the viscosity of the Earth's mantle, which varies both radially and laterally by several orders of magnitude beneath the AIS (Kaufmann et al., 2005; Ivins et al., 2023), as derived from seismic models (e.g. Lloyd et al., 2019). Therefore, bedrock uplift is influenced not only by the amount of ice mass loss but also by the solid Earth properties of the region where the loss occurs. Models that include the bedrock deformation for a 3D Earth structure project a maximum of 23-40% reduction in sea-level rise over the coming centuries compared with models that assume a rigid Earth (Gomez et al., 2024; van Calcar et al., 2025), and a delay in grounding line retreat in the Amundsen Sea Embayment by up to 130 years (van Calcar et al., 2025). However, it is currently unfeasible to include a 3D GIA model in a large ensemble of sea-level projections that use dynamic ice-sheet models due to the long computation time involved (van Calcar et al., 2023). Therefore, projections of AIS evolution by ice-sheet models either omit bedrock uplift or use simplified Earth models (Levermann et al., 2020).

One such simplified model that is commonly coupled to ice-sheet models is the elastic lithosphere, relaxed asthenosphere (ELRA) model (Le Meur and Huybrechts, 1996). ELRA models are computationally cheap, easy to implement in ice-sheet models, and can be used in combination with a range of ice models, allowing large ensembles of sea-level projections to be simulated (Bulthuis et al., 2019; Levermann et al., 2020; DeConto et al., 2021; Coulon et al., 2024). Typically, ELRA is used with a uniform relaxation time of 3000 years and a flexural rigidity of $10^{25}$ kg m$^2$ s$^{-2}$, which roughly corresponds to a mantle viscosity of $10^{21}$ Pa·s and a lithospheric thickness of 100 km, respectively (Le Meur and Huybrechts, 1996; Bulthuis et al., 2019; Levermann et al., 2020; DeConto et al., 2021; Coulon et al., 2024). The relaxation time of the Earth's mantle, which is a characteristic time scale that expresses how fast the mantle responds to changes in surface loads, serves as a proxy for mantle viscosity because it also reflects how fast the viscous mantle flows under stress. Furthermore, ELRA includes the flexural elastic response of the lithosphere, but it neglects the elastic part of the viscoelastic response.

Alternatively, some sea-level projections use ice-sheet models coupled with a 1D GIA model. A GIA model includes the bedrock deformation due to changes in ice loading, and can additionally solve the sea-level equation to include changes in

ocean loading. In this study, we use the term GIA model for a model that computes deformation based on ice loading only. A 1D GIA model includes an Earth structure where viscosity varies radially and not laterally, equivalent to a self-gravitating viscoelastic Earth (SGVE) model (Le Meur and Huybrechts, 1996). Current existing ice sheet projections that are derived in conjunction with a 1D GIA model or bed deformation model with a viscoelastic half-space use a homogeneous upper mantle viscosity of $10^{21}$ Pa·s (Gomez et al., 2015; Konrad et al., 2015; Rodehacke et al., 2020; Golledge et al., 2015; Klose et al., 2024). However, using such a viscosity value, or a relaxation time of 3000 years, does not affect sea-level rise projections significantly compared with excluding bedrock deformation entirely, and it overestimates sea-level rise by up to 20% by the year 2500 compared with projections that use GIA models that consider 3D Earth structure, which we refer to as 3D GIA models (van Calcar et al., 2025).

In the Amundsen Sea embayment, mantle viscosity can be as low as $10^{18}$ Pa·s (Barletta et al., 2018). Incorporating a low-viscosity zone in the upper mantle within a 1D GIA model leads to a significant stabilizing effect on the ice sheet over thousands of years (Pollard et al., 2017). However, the same study showed that different relaxation times for East and West Antarctica did not contribute significantly to the uncertainty in the multi-centennial response of the AIS to climate change. This can be explained by the chosen relaxation times for West Antarctica which were longer than 1000 years, while they might be a lot shorter in this region (Bulthuis et al., 2019). Other research has demonstrated that using a laterally varying relaxation time in ELRA (LVELRA) with a shorter relaxation time under West Antarctica results in a significantly reduced sea-level contribution from Antarctica on multicentennial-to-millennial timescales for four different warming scenarios of 5000 years (Coulon et al., 2021). While it has long been possible to determine relaxation time spectra for radially varying viscosity profiles (McConnell, 1965), such calculations have not been performed for profiles with both lateral and radial variations in viscosity. For a viscous half-space with uniform mantle viscosity, the relaxation time is equal to the Maxwell relaxation time that can be computed directly from the viscosity and shear modulus (e.g. Turcotte and Schubert, 2002). However, in a laterally and radially varying Earth structure, there is no longer a simple relation between Maxwell time and viscosity. In analytic GIA models based on the normal mode method (e.g. Wu & Peltier, 1982), each eigenmode has a characteristic relaxation time, but the complete response is controlled by a weighted combination of modes that depends on the spatial scale of the load and the properties of the lithosphere. This implies that the relaxation time that is induced by a certain change in ice load for a given viscosity profile depends on the size of the ice (un)loading. Thus, a single, uniform relaxation time constant in time cannot be directly derived from a local viscosity. Consequently, the variation in mantle relaxation times across Antarctica remains unknown.

A laterally varying relaxation time can be implemented in a straight-forward way in an ELRA model (Oude Egbrink, 2018; Coulon et al., 2021). However, sea-level projections generated using coupled ELRA-ice-sheet models have not been compared with the output from coupled ice sheet-3D GIA models, leaving it unclear how well different relaxation times and 1D mantle viscosity profiles are able to approximate the deformation simulated by more complex models that include 3D Earth structures. Relatively fast GIA models that incorporate laterally varying viscosity and could, in principle, be coupled to ice-sheet models do exist (Nield et al., 2018; Book et al., 2022; Weerdesteijn et al., 2023). However, these are regional, flat-Earth models, which can introduce substantial errors when applied to Antarctic-wide simulations. A computationally efficient Earth model based

on fast Fourier transforms has been developed that approximates lateral variations in mantle viscosity and lithospheric thickness in the Earth structure and takes into account the effect of a spatially and time varying sea level on deformation (Swierczek-Jereczek et al., 2024). While containing multiple advantages over ELRA, such as including the effect of load wavelength, this model has only been evaluated over a full glacial cycle and not for future projections, and it shows notable discrepancies compared to the 3D GIA model it was benchmarked against. Coulon et al. (2021) coupled ELRA with a gravitationally consistent geoid calculation so compute near-field relative sea-level changes. Furthermore, ELRA uses a single relaxation time and is therefore independent of load wavelength, but the framework could in principle be extended to a scale-dependent formulation where the relaxation time becomes a function of wavenumber.

In this study, we focus on assessing the performance of the Earth models already used in ice-sheet modelling in current literature, which is ELRA, LVELRA, 1D GIA, and 3D GIA models. We use the average barystatic sea-level contribution from the ice-sheet model IMAU-ICE coupled to a 3D GIA model with two different realizations of 3D Earth structures as a reference, and we use the ice-sheet model coupled to ELRA and a 1D GIA model to answer the following research questions:

1. What is the best parameter choice for a coupled ice sheet – ELRA model using uniform relaxation time to approximate the ice sheet evolution resulting from the reference model?

2. What is the best parameter choice for a coupled ice sheet – ELRA model using laterally varying relaxation time (LVELRA) to approximate the ice sheet evolution resulting from the reference model?

3. What is the best parameter choice for a coupled ice sheet – 1D GIA model to approximate the ice sheet evolution resulting from the reference model?

To address these questions, 3D GIA simulations are conducted using a global spherical finite element model (van der Wal et al., 2013; van der Wal et al., 2015; Blank et al., 2021, van Calcar et al., 2023) coupled to the ice-sheet model IMAU-ICE (Berends et al., 2022; van Calcar et al., 2025). We use constraints from seismic velocity studies to determine the spatially-varying rheological properties of the mantle (Wu et al., 2013). Output from models that employ 3D and 1D Earth structures, and maps of different relaxation times are compared in terms of barystatic sea-level rise, grounding line position, ice thickness and bedrock uplift. As a result, we recommend values for uniform relaxation times in combination with a flexural rigidity that results in a barystatic sea-level rise close to the average barystatic sea-level rise resulting from two 3D Earth structures. One structure is based on a viscosity constraint in the Amundsen Sea Embayment, and one is based on a constraint in the Weddell Sea Embayment and Palmer Land in the Antarctic Peninsula. Furthermore, we derive a relation between relaxation time and viscosity and recommend a laterally varying relaxation time map in combination with a flexural rigidity. Last, we recommend a 1D viscosity profile to approximate a 3D viscosity profile.

**2 Method**

To compare the performance of the ELRA, LVELRA and 1D GIA models with that of a 3D GIA model, we conduct sea-level projections using the ice-sheet model IMAU-ICE coupled to all three of these Earth models. We compare the AIS evolution

over the next 500 years under different warming scenarios and climate models using a variety of Earth structures. We use the projections of two climate models from the sixth phase of the Coupled Model Intercomparison Project (Eyring et al., 2016), namely CESM2-WACCM (hereafter referred to as CESM, Danabasoglu et al., 2020) and IPSL-CM6A-LR (hereafter referred to as IPSL, Lurton et al., 2020), under a low emission scenario (SSP1-2.6) and a high emission scenario (SSP5-8.5) (Coulon et al. (2024); Klose et al., 2024). These two climate models both show warming around the whole West Antarctic Ice Sheet, but forcing magnitudes, and long-term projections of precipitation, atmospheric temperature, and oceanic temperature and salinity differ. In CESM, ocean warming mainly occurs in the Weddell Sea, whereas in IPSL, the warming mainly occurs in the Amundsen Sea. Warming is projected in the Ross Sea for both climate models. The climate models provide ocean temperature, salinity and atmospheric temperature anomalies, and precipitation ratios until the year 2300, which are used to force the ice-sheet model. Since there are no climate projections available beyond 2300, the forcing is kept constant between 2300 and 2500. The ocean temperature anomalies are shown in Supplementary Fig. 1 for each climate model and emission scenario.

The thermomechanically coupled model IMAU-ICE is based on the shallow ice and shallow shelf approximations (Morland, 1985; Bueler & Brown, 2009; Berends et al., 2022). Ice velocities are computed on a 16 km grid resolution. At the grounding line, we applied the flotation condition melt parameterization. The position of the grounding-line can freely evolve from the physics and numerics of the model without explicitly forcing a flux. Basal sliding follows the regularized Coulomb law (Zoet & Iverson, 2020). Basal melt at the ice shelf is computed using a quadratic local law (Favier et al., 2019) and the surface mass balance is computed using a temperature and radiation parametrization (Berends et al., 2022). The present-day bedrock and ice surface topography are taken from Bedmachine version 3 (Morlighem et al., 2020). The model does simulate marine ice sheet instabilities, but not marine ice cliff instabilities. The barystatic sea-level contribution is computed as the difference in volume of ice above flotation (van Calcar et al., 2025).

The ice-sheet model is coupled to an ELRA and a GIA model. The coupling method is discussed in detail in van Calcar et al. (2023) and van Calcar et al. (2024). An overview of the simulations with different Earth models is provided in Tab. 1. The bedrock deformation is computed based on the change in grounded ice thickness above flotation, which is computed by the ice-sheet model. In turn, the bedrock topography in the ice-sheet model is updated by the bedrock deformation provided by the ELRA or GIA model.

Besides the stabilising effect of bedrock deformation on ice-sheet evolution, there is also a sea surface height component, and together these comprise the sea-level feedback. The reduced gravitational pull from the ice sheet causes a local sea-level drop of up to 8 meters by the year 2500, particularly near regions of major ice loss in the West Antarctic Ice Sheet (van Calcar et al., 2025). However, this 8-meter drop in local sea level is small compared to the effect of bedrock deformation from ice mass changes, which results in up to 150 meters of uplift by 2500. Additionally, gravitational changes due to Earth deformation affect sea level. This additional stabilising feedback from the spatially and temporally varying sea surface height reduces barystatic sea-level projections in 2500 by 5 percent compared to simulations where sea level is fixed at present-day. Previous studies have likewise shown that the deformational component of GIA dominate the sea-level feedback on ice-sheet evolution

(Kachuck et al., 2020; Coulon et al., 2021). In all simulations presented in this study, the gravitational effect on sea level is not taken into account and sea level is therefore kept fixed at present-day in both the GIA and ice-sheet models.

**Table 1: Different Earth structures used in the coupled ice sheet – Earth models. The 2D relaxation time in the ELRA model is described in detail in section 3. The 1D viscosity profiles correspond to uniform upper mantle viscosities of $10^{21}$, $10^{20}$ and $10^{19}$ Pa·s, respectively. 1DASE refers to an upper mantle viscosity as suggested by Barletta et al. (2018). All 1D viscosity profiles are shown in Fig. 1. The 3D-stronger and weaker structures are taken from van Calcar et al. (2024).**

| Model | Input | Earth structures |
| --- | --- | --- |
| **ELRA** | Uniform | - Relaxation time: 3000, 1500, 500, 450, 400, 350, 300, 250 & 200 yr<br>- Lithospheric thickness: ~100 & ~60 km |
| | 2D relaxation time | - Based on 3D-stronger & 3D-weaker<br>- 2 different fits between relaxation time and viscosity<br>- Lithospheric thickness: ~120 & ~60 km |
| **1D GIA** | 1D profiles | 1D21, 1D20, 1D19 & 1DASE |
| **3D GIA** | 3D rheologies | 3D-stronger & 3D-weaker |

## 2.1 1D and 3D GIA models

To compute the Earth's deformation, a global spherical finite element model based on Abaqus software is used (van der Wal et al., 2013; van der Wal et al., 2015; Blank et al., 2021, van Calcar et al., 2023). The model includes material compressibility but it does not solve the sea-level equation, and it does not account for rotational feedback or the migration of coastlines because these have a relatively minor effect on sea-level change compared with the effect of changes in grounded ice thickness (Milne et al., 1999; van Calcar et al., 2025). This model is used for two purposes: (1) To produce sea-level projections via coupling to the ice-sheet model (referred as configuration 1 of the GIA model), and (2) to calculate the relaxation time of the bedrock deformation as a response to schematic ice unloading experiments which are used to derive a relation between relaxation time and viscosity (referred as configuration 2 of the GIA model).

In the GIA model, deformation in the upper mantle is assumed to be governed by diffusion and dislocation creep in olivine (Hirth & Kohlstedt, 2003) as in earlier studies (van der Wal et al., 2013; van der Wal et al., 2015; Blank et al., 2021, van Calcar et al., 2023; van Calcar et al., 2025). We do not specify lithospheric thickness, but instead use seismic velocity perturbations to assign appropriate rheological properties in each element between 35 and 670 km depth. At shallower depths, the layer is defined to be purely elastic. At deeper depths, the lower mantle is assumed to be homogenous with a viscosity of $5 \cdot 10^{21}$ Pa·s.

The effective viscosity, $\eta_{\text{eff}}$, is a function of the von Mises stress, $q$, and hence it is an output of the model rather than a property that is assumed *a priori*:

$$\eta_{\text{eff}} = \frac{1}{3B_{\text{diff}} + 3B_{\text{disl}}q^{n-1}} \tag{1}$$

Here, $n$ is the stress exponent, and $B_{\text{diff}}$ and $B_{\text{disl}}$ are laterally varying creep parameters for diffusion and dislocation creep as shown in Eq. 2a and 2b (Hirth and Kohlstedt, 2003).

$$B_{diff} = A_{diff}d^{-3}f_{H_2O}^1 e^{-\frac{E+PV}{RT(x,y)}} \tag{2a}$$

$$B_{disl} = A_{disl}d^0 f_{H_2O}^{1.2} e^{-\frac{E+PV}{RT(x,y)}} \tag{2b}$$

$A$ is experimentally determined ($A_{diff} = 10^6$ MPa, $A_{disl} = 90$ MPa), $d$ is the grain size, $f_{H_2O}$ is the water content, $E$ is the activation energy which is taken to be $335 \cdot 10^3$ kJ/mol for diffusion creep and $480 \cdot 10^3$ kJ/mol for dislocation creep. $P$ is the depth dependent pressure (Kearey et al., 2009). $V$ is the activation volume which is taken to be $4 \cdot 10^{-6}$ m$^3$/mol for diffusion creep and $11 \cdot 10^{-6}$ m$^3$/mol for dislocation creep. $R$ is the gas constant, .$A$, $E$ and $V$ are different according to the values for wet and dry olivine and are given here for wet olivine. All parameters, except temperature, grain size and water content, are taken from Hirth and Kohlstedt (2003). In this study, melt content is neglected as it has a relatively small influence on viscosity in this formulation (van der Wal et al., 2015). $T(x,y)$ is the spatially varying mantle temperature, which is derived from a high-resolution seismic model (Lloyd et al., 2019) in combination with a global seismic model from Becker and Boschi (2002). The mantle temperature variations are determined by converting these global seismic velocity perturbations to temperature perturbations using derivatives from Karato (2008), and then converting these to absolute temperature assuming a standard mantle geotherm (Turcotte and Schubert, 2002).

The upper mantle viscosity can vary greatly depending on the grain size and water content used. To obtain a 3D rheology, two different combinations of grain size and water content are chosen such that the average viscosity values across the Amundsen Sea Embayment and the Weddell Sea Embayment are the same as those constrained by GIA observations (Ivins et al., 2023), resulting in a relatively weaker 3D structure (labelled 3D-weaker) and a relatively stronger 3D Earth structure (labelled 3D-stronger) respectively (van Calcar et al., 2025). The 3D-weaker structure contains a water content of 400 ppm and a grain size of 2.5 mm, and the 3D-stronger structure contains a water content of 200 ppm and a grain size of 4.5 mm. Since the viscosity is constrained by observations, both structures are considered realistic and not just an upper or lower limit. Background stress that contributes to the variable $q$ in Eq. 1 is ignored here. Including background stress from the long-term GIA signal would lower viscosity (Blank et al., 2021), which will be compensated by grain size and water content parameters to still match the viscosity constraints.

For the coupling to the ice model, the GIA model is used with 10 vertical layers (0-35 km, 35-100 km, 100-150 km, 150-300 km, 300-420 km, 420-550 km, 550-670 km, 670-1171 km, and 1171-2890 km, and 2890-6371 km). We label this model as Configuration 1 of the GIA model. A high resolution area is defined over Antarctica with a horizontal and vertical grid

resolution of 30 km wide and deep between the surface and 670 km depth. The sensitivity test for the effect of resolution over a glacial cycle presented in van Calcar et al. (2023) shows that using a horizontal resolution of 15 km by 15 km instead of 30 km by 30 km decreases the total deformation by 0.01% (2 cm) over 1000 years and increases the computation time of the GIA model by approximately 30%. The uncertainty could be larger for elastic effects with a smaller spatial wavelength and deformation on shorter timescales. However, the uncertainty is significantly smaller than the uncertainty in adopted Earth structure (Wan et al., 2022; van Calcar et al., 2023). The spatial resolution outside the high-resolution area is 200 km wide and deep.

Different methods can be used to simulate the response due to 1D Earth structures (Peltier, 1974; Wu, 1998) but here we use the same GIA model to simulate a 1D Earth structure and a 3D Earth structure to avoid introducing differences that arise due to model formulation.

The relatively high computation time of the GIA model limits the number of cases we can investigate. Four 1D Earth structures are applied in the GIA model: one commonly used structure with an upper mantle viscosity of $10^{21}$ Pa·s, two structures with an upper mantle viscosity of $10^{20}$ and $10^{19}$ Pa·s, respectively, to represent the average viscosity under West Antarctica, and one with an upper mantle viscosity between $5 \cdot 10^{18}$ and $3 \cdot 10^{19}$ Pa·s that could represent the Amundsen Sea embayment (Barletta et al., 2018). These structures are hereafter referred to as 1D21, 1D20, 1D19 and 1DASE, respectively, and their 1D viscosity variations with depth are shown in Fig. 1.

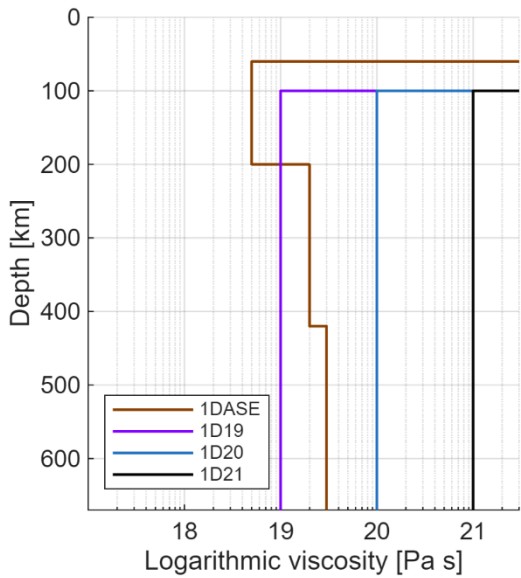

**Figure 1: Upper mantle viscosity of the 1D Earth structure profiles.**

To derive a relation between relaxation time and viscosity, as discussed in section 3, we used uplift rates from a schematic experiment using a 3D GIA model. In this case, the spatial resolution is 2 degrees at the surface and includes 8 vertical layers following van der Wal et al. (2013) and van der Wal et al. (2015) (0-35 km, 35-70 km, 70-120 km, 120-170 km, 170-230 km, 230-400 km, 400-670 km, and 670-2890 km), which we label as Configuration 2 of the GIA model. This resolution allowed us to run many schematic experiments. We used a global seismic model from Schaeffer & Lebedev (2013) combined with the regional seismic model of Heeszel et al. (2016) derived from Rayleigh wave array analysis over Antarctica to create a global seismic model. Uncertainties in the regional seismic model are used to ensure a smooth transition between values defined by the regional and global seismic models. The mantle temperature is then obtained following the same procedure as described for Configuration 1. The dislocation and diffusion parameters are then computed using the mantle temperature, stresses, and a suite of globally-uniform values for grain size (1, 4, and 10 mm) and water content (0 or 1000 ppm $H_2O$). Below 400 km, uniform creep parameters are adopted, which yield mantle viscosities of ~$10^{21}$-$10^{23}$ Pa s. The 3D GIA model is coupled to a

code that solves the sea-level equation (Farrell & Clark, 1976) as implemented by Wang and Wu (2006). This configuration is uncoupled from the ice dynamic model and the applied ice loading is further described in section 3.

## 2.2 ELRA

In ice-sheet modelling, the bedrock response is often simplified to the ELRA approximation as described in detail in Le Meur & Huybrechts (1996). In this approach, bedrock deformation is obtained by a convolution of the deformation to a point load with the actual load and is dependent on the flexural rigidity and the relaxation time. The flexural rigidity ($D$) determines, together with the density of the asthenosphere and the gravity acceleration at the surface, the radius of relative stiffness ($L_r$) as shown in Eq. 3.

$$L_r = \left(\frac{D}{\rho_a g}\right)^{\frac{1}{4}} \tag{3}$$

First, the equilibrium deflection ($w$) at a normalized distance ($x$) from a point load ($q$) is computed using

$$w(x) = \frac{q L_r^2}{2\pi D} \cdot X(x), \tag{4}$$

where $X$ is the zeroth order Kelvin function of $x$. The normalized distance is defined as the real distance ($r$) from the point load divided by the radius of relative stiffness. The total deflection at each point is the sum of the deflection at all neighboring 265 points within a distance of six times the radius of relative stiffness.

Second, the bedrock deflection can be computed using

$$\frac{db_{i,j}}{dt} = \frac{w_{i,j} - b_{i,j}}{\tau}, \tag{5}$$

where $\frac{db}{dt}$ is the bedrock elevation change over time, $b$ the current bedrock elevation, $\tau$ the relaxation time and $i, j$ the grid 270 coordinates.

Besides the commonly used relaxation time of 3000 years and flexural rigidity of $10^{25}$ N·m, we also applied a relaxation time of 1500, 500, 450, 400, 350, 300, 250, 200 and 50 years in combination with a flexural rigidity of $1.92 \cdot 10^{24}$ N·m. The flexural rigidities roughly correspond to lithospheric thicknesses of 100 km and 60 km, of which the latter is close to the estimated 275 lithospheric thickness of West Antarctica (e.g. An et al. 2015; Pappa et al. 2019). The lithospheric thickness is estimated to be much larger in East Antarctica, but the bedrock deformation in response to ice load change is relatively insensitive to variations in lithospheric thickness compared to the relaxation time (Coulon et al., 2021). The lithospheric thickness can be derived from the definition of flexural rigidity:

$$D = \frac{E h^3}{12(1 - v^2)}, \tag{6}$$

with the Young's modulus ($E$) set to 100 GPa and the Poisson's ratio ($v$) set to 0.25. The lithospheric thickness is defined by $h$.

To include laterally varying relaxation times (derived in section 3) in the ELRA model, we made the relaxation time in Eq.5 a function of the 2D grid coordinates, such that $\tau$ becomes $\tau_{i,j}$. A laterally varying flexural rigidity is also possible to implement in the ELRA model but this is more complex and the effect on bedrock deformation is limited (Coulon et al, 2021; Zhao et al., 2017; Mitrovica et al., 2011). We therefore used uniform flexural rigidity values of $1.536 \cdot 10^{25}$, $1 \cdot 10^{25}$, $4.5511 \cdot 10^{24}$, and $1.92 \cdot 10^{24}$ N·m, corresponding roughly to lithospheric thicknesses of 120, 100, 80 and 60 km (Eq. 6), in agreement with estimates for lithospheric thickness across West and East Antarctica (Lloyd et al., 2019).

## 3 Deriving 2D relaxation time maps from 3D viscosity profiles

Here, using the 3D GIA model in Configuration 2, we determine relaxation times empirically by analysing the solid Earth deformation triggered by the removal of schematic surface loads with the aim to derive a relationship between relaxation time and viscosity that can be used for any viscosity map without a priori constraints on where ice loss is exactly taking place. The surface loads are chosen to reflect large and small areas of ice mass change for different regions in West Antarctica. The small area is chosen to cover the main area of mass loss close to the present-day grounding line. The large area is chosen to cover the full basin of the Embayment, or the Peninsula. By applying schematic ice loads in various locations, the resulting empirical relation between mantle viscosity and relaxation time accounts for a wide spectrum of mantle conditions such that the relation is valid over a large viscosity range, including mantle viscosities similar to those found around the grounding line in East Antarctica. The resolution of the 2-degree finite element mesh that is used in this configuration of the 3D GIA model is relatively coarse, and therefore determines the exact shape of each area of loading (Fig. 2a). The uniform thickness of each load is taken to be 500 m to approximate stress changes comparable to those expected in realistic ice loss scenarios. To reduce computational costs, only the wavelength of the ice load is varied, and not the ice thickness, as the normal mode theory shows that wavelength is most influential on relaxation time derived from deformation. Each load is placed on the Earth until equilibrium is reached, and then instantaneously removed.

A total of 40 simulations are conducted, using a grain size of 1, 4 and 10 mm, a water content of 0 and 1000 ppm and a small, medium and large of the region of loading (as shown in Fig. 2a). For each simulation, the resulting displacement over time for each surface load/Earth model combination is computed, yielding a displacement curve. Each simulation contains 20 timesteps, of which the first time step is 15 years, increasing by a factor of 1.5 until the largest time step of 33.3 kyr. From the displacement curve, the uplift rate through time is calculated by time differentiation. The relaxation time is computed as half the time it takes for solid Earth rebound rates to decrease by $1/e^2$ following instantaneous unloading (Table 1 in Supplementary materials). Averaging over two relaxation times reflects more accurately the fact that viscosities at different depths will control the deformation at different stages of the relaxation. The difference in relaxation time between the large and small region of loading is on average 12% (31 years), with one outlier of 45% (47 years) in the Amundsen Sea Embayment where a large area of ice mass loss (indicated by pink and red in Fig. 2a) leads to a significantly lower relaxation time than a smaller area of ice mass loss (indicated by pink in Fig. 2a). This large difference only occurs for a water content of 1000 ppm and a grainsize of 1 cm.

Typically, depth averaged viscosities are computed by taking the average of the logarithmic viscosity values in a certain layer or area (e.g. Paulson et al., 2005; Whitehouse et al., 2006; Bagge et al., 2021). The filled symbols in Fig. 2b show the characteristic relaxation time of each region plotted against the average mantle viscosity, calculated as the volume-weighted mean viscosity of all elements between 120 and 400 km depth beneath each unloaded region indicated in Fig. 2a. However, the region in the mantle that primarily governs the Earth's response is determined by how strongly the Earth's deformation

under the ice load is influenced by viscosity at different depths, which in turn depends on the viscosity profile itself (Peltier, 1976; Wu 2006). The sensitivity to the viscosity profile can be taken into account by computing the vertically averaged viscosities weighted by the local strain rate (Christensen, 1984). Such a procedure would result in average viscosity values that are determined more by low viscosity values in sub-surface Antarctica (because low viscosity regions will experience the highest strain rates). To take that into account, the computed relaxation times are compared to not only the average mantle

viscosity value for each region, but also the lowest mantle viscosity derived from the seismic model, which is shown by the open symbols in Fig. 2b. A linear fit through the resulting log-log graph provides a relation between relaxation time in years, τ, and viscosity in Pa·s for the average viscosity (solid line in Fig. 2) and the lower bound viscosity (dashed line in Fig. 2). The linear fit is determined by Eq. 7:

$$\tau = a \cdot 10^{-b} \eta_{eff}{}^{c} \tag{7}$$

where $a$ is 2.3, $b$ is 5, and $c$ is 0.35 in the case when the average viscosity is used, and $a$ is 3.9, $b$ is 2 and $c$ is 0.20 in the case when the lower bound viscosity is used. Both relations will be used to create 2D relaxation time maps to identify which one is best approximating the sea-level rise projections resulting from the coupled 3D GIA – ice-sheet model. When the 2D relaxation time maps are used in an ELRA model, the relaxation time should be smoothly varying because otherwise discontinuities in deformation arise for adjacent points. Either a high-resolution viscosity profile should be used, because this will likely not

contain large sharp changes in viscosity, or the relaxation time map should be smoothed, as applied in this study.

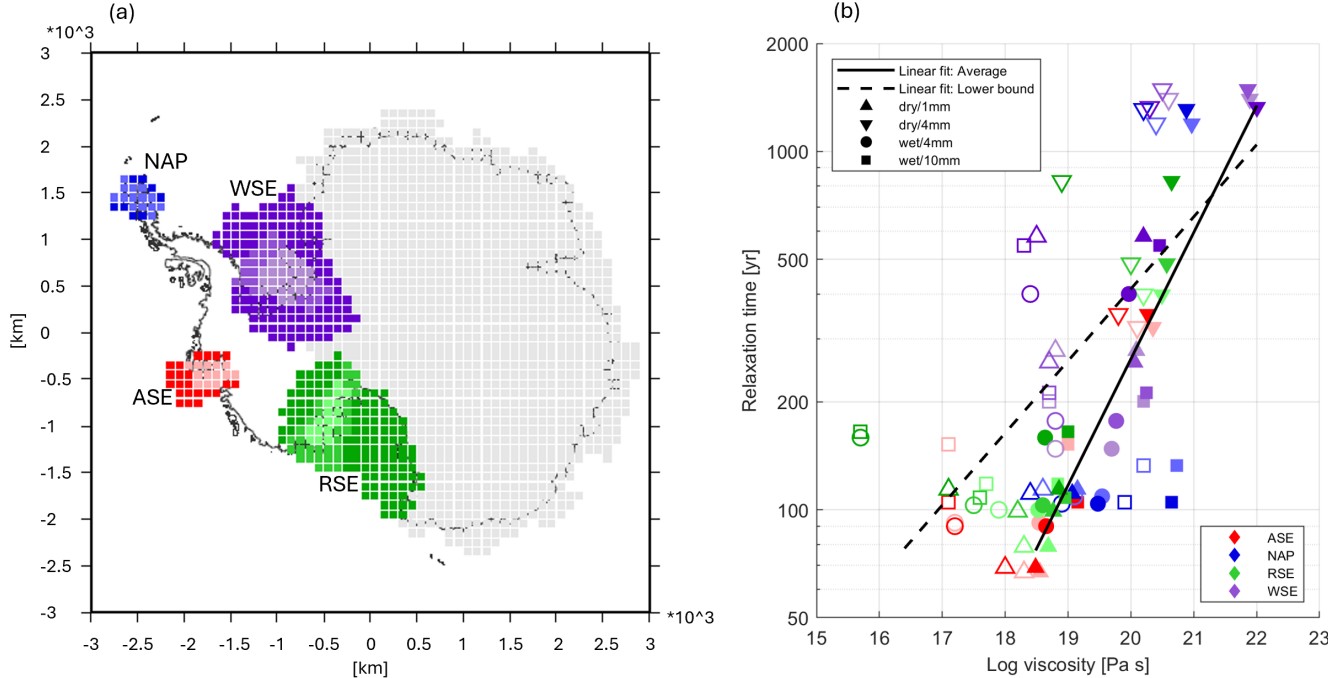

**Figure 2: The relationship between mean upper mantle viscosity and relaxation time across West Antarctica. (a) Regions from which ice is instantaneously unloaded in order to determine regional relaxation times. Within each of the four main regions, there is a large and a small version of the region, coloured in dark and light colours respectively. The small region overlaps the large region. NAP refers to Northern Antarctic Peninsula, WSE refers to Weddell Sea Embayment, ASE refers to Amundsen Sea Embayment, and RSE refers to Ross Sea Embayment. (b) Log-log plot of relaxation time against mean upper mantle viscosity and the lower bound viscosity. The colours are identical to (a). The symbols reflect the parameters in the 3D GIA model used in each experiment (see Supplementary Table 1). The filled symbols reflect the average viscosity and the open symbols reflect the lower bound viscosity.**

Using our empirically derived relationships between viscosity and relaxation time (Eq. 7), we derive laterally variable relaxation time maps based on the 3D-weaker and 3D-stronger Earth models described in Section 2.1. $\eta_{\text{eff}}$ is taken to be the viscosity of the 3D Earth structure vertically averaged between 120 and 400 km depth (Fig. 3a,d). For ice thickness changes over a timescale of centuries, the highest sensitivity will be in this relatively shallow layer (Barletta et al., 2018). This results in two relaxation time maps based on the 3D-weaker rheology, hereafter referred to as 2D-weaker Average and 2D-weaker Lower bound (Fig. 3 b-c), and two relaxation time maps based on the 3D-stronger rheology, hereafter referred to as 2D-stronger Average and 2D-stronger Lower bound (Fig. 3 e-f). Finally, the minimum relaxation time is set to 67 years, equal to the minimum relaxation time found in the experiments used to derive Eq. 7.

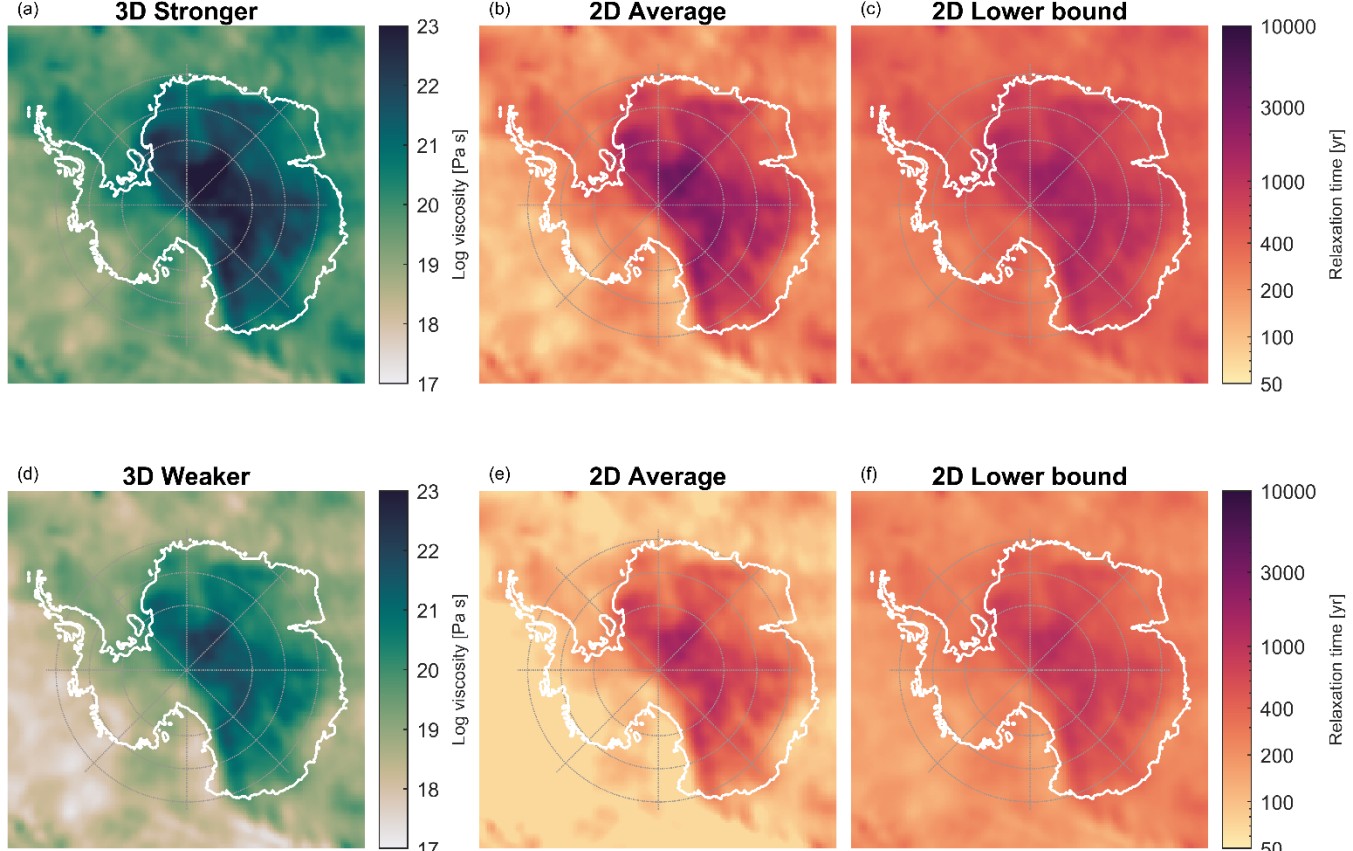

Figure 3: Panels a and d show the vertically averaged mantle viscosity between 120 and 400 km depth based on van Calcar et al. (2024). Panels b and e show the relaxation time maps computed using Eq. 7 with the parameters for the average viscosity fit, and 3D-stronger and 3D-weaker, respectively. Panels c and f show the relaxation time maps computed using Eq. 7 with the parameters for the lower bound viscosity fit, and 3D-stronger and 3D-weaker, respectively.

## 4 Projections using different approaches to bedrock deformation

Sea-level rise over the next 500 years is projected using two different climate models, each under a high and a low emissions scenario. Projections using the ice-sheet model coupled with simple Earth models that adopt a uniform relaxation time, a laterally variable relaxation time, and a 1D Earth structure are compared to the average sea-level rise obtained using the coupled ice-sheet – 3D GIA model (configuration 1) with 3D-weaker and 3D-stronger Earth structures. The average barystatic sea-level rise computed by the ice-sheet model coupled to the 3D GIA model using the two different 3D Earth structures is referred to as 3D-Average.

Depending on the emission scenario and climate model, we project ~3-7.5 m of barystatic sea-level rise with significant retreat in various basins due to marine ice sheet instability. This Antarctic sea-level contribution is significantly higher compared to other studies using the same forcing, which likely stems from differences in key model components, particularly the initialisation, the basal melt scheme, and the melt parametrization at the grounding line (Coulon et al., 2024; Klose et al.,

2024), which can lead to large variations in ice sheet evolution and corresponding sea-level rise (Seroussi et al., 2024). Other

climate models provide significantly different thermal forcings in terms of magnitude and spatial distribution, which might not be captured by the two climate models used in this study. However, the goal of the present study is to isolate and quantify the relative impact of different Earth structures on Antarctic ice sheet evolution, rather than sampling the full parameter space of IMAU-ICE or to conduct a full intercomparison of model sensitivities. To estimate the uncertainty associated with the magnitude of retreat, we include a scenario where the West Antarctic Ice Sheet collapses, meaning that most of the grounded

ice has been lost (SSP5-8.5), and a scenario where the Thwaites and Pine Island glaciers retreat significantly whereas the rest of the West Antarctic Ice Sheet is relatively stable (SSP1-2.6). Both scenarios include significant ice mass loss in Wilkes basin in East Antarctica. Together, these simulations capture both rapidly retreating and relatively stable drainage basins across different Antarctic regions.

## 4.1 ELRA model with uniform relaxation time

For all scenarios and climate models, retreat and thinning of the ice sheet occurs in the West Antarctic Ice Sheet and Wilkes Basin. The bedrock deformation depends indirectly on the climate model because varying ocean warming causes ice retreat in different regions, and the mantle viscosity differs in each region. The bedrock deformation depends on the emission scenario as well, since a larger region of ice mass loss will trigger deformation deeper in the mantle where viscosity, and hence relaxation times, will be different to values at shallower depths (Peltier, 1976). The sea-level rise resulting from the coupled

ice sheet – GIA model using a 3D Earth structure therefore differs from a uniform relaxation time, and this difference in turn varies for different emission scenarios and climate models.

To assess the performance of ELRA with a uniform relaxation time, the resulting sea-level rise is compared to the sea-level rise averaged from the output of the two models that employ 3D Earth structures (3D-Average). The widely used uniform relaxation time of 3000 years (hereafter referred to as ELRA3000) overestimates the contribution from the AIS to sea-level

rise by 0.44-0.70 m (8-20%) in 2500 compared to the 3D-Average value, with the precise value depending on the emission scenario and the applied climate model (Fig. 4b,c,e,f). First, differences occur because the ELRA model approximates bedrock adjustment as a local viscoelastic response with a single relaxation timescale, while GIA models resolve the full, gravitationally self-consistent, depth-dependent viscoelastic deformation of a layered Earth. Second, differences occur due to the chosen Earth structure in the models. ELRA3000 overestimates sea-level rise because this relaxation time is much longer than the relaxation

time associated with the low viscosity values found in the 3D Earth structures (Fig. 3), especially when retreat occurs in the Amundsen Sea Embayment (as predicted by climate model IPSL) where the mantle viscosity is relatively low. We therefore search for a better choice of relaxation time, as formulated in research question 1: What is the best parameter choice for a coupled ice sheet – ELRA model using uniform relaxation time to approximate the ice sheet evolution resulting from the reference model?

We investigated the effect of a uniform relaxation time of 200 and 500 years to increase the stabilisation effect of GIA on the ice sheet retreat compared to a uniform relaxation time of 3000 years. The optimal choice of relaxation time is defined as the

ELRA simulation with the smallest root mean square error (RMSE) compared to the 3D Average over the full time series, for both climate models and both emission scenarios. The RMSE is shown in supplementary Tab. 1 for each simulation. We find 300 years, with an uncertainty range of 25 years, as approximating closest to the 3D results in combination with a flexural rigidity corresponding to 100 km lithospheric thickness (Fig. 4).

For SSP1-2.6-IPSL, the difference in sea-level rise between using a relaxation time of 300 yr (hereafter referred to as ELRA300) and 3D-stronger is negligible until 2400, but increases afterwards, reaching a maximum of 17 cm in 2500 (Fig. 4c), which is 5% of the total of 3.6 m of sea-level rise using 3D-stronger (Fig. 4a). The ice is approximately 50 m thicker within the Amundsen Sea Embayment using ELRA300 (Fig. 5a) due to faster uplift in this region compared to 3D-stronger. On timescales of 400 years and longer, it is not only the local low viscosity, but also the surrounding higher viscosities, which impact bedrock deformation in the 3D model. The rate of uplift predicted by the 3D GIA model therefore slows down on these longer timescales whereas the relaxation time in ELRA is constant over time and corresponds only to the low viscosities of the 3D model. As a consequence, the amount of bedrock uplift is about 75 m greater in ELRA300 than 3D-stronger between 2400 and 2500. The impact of the difference in bedrock elevation on ice mass loss and grounding line position is negligible.

Contrary to this, the viscosity of 3D-weaker is much lower and the uplift predicted by ELRA300 is too slow compared to 3D-weaker over the full simulation time. The bedrock elevation of ELRA300 is tens of meters lower than 3D-weaker in 2300, causing faster retreat to be predicted by ELRA300 until 2500. The grounding line is similar between different Earth models for most of the AIS because bedrock deformation only has an effect in regions where there is mass loss. For SSP1-2.6-IPSL, significant ice mass loss in the West Antarctic Ice Sheet only occurs in the Amundsen Sea Embayment, where the grounding line in ELRA300 is 150 km greater than 3D-weaker by 2500 (Fig. 5).

ELRA300 also performs well when evaluated on the contribution of individual drainage basins to barystatic sea-level change, for both fast and slow retreating basins. For example, the drainage basin in Queen Maud Land in East Antarctica contributes significantly to barystatic sea-level change, however the impact of GIA is neglectable as the grounding line position is insensitive to bedrock deformation in this ice-sheet model (basin 6 in Supplementary Fig. 2). Therefore, the choice of relaxation time becomes arbitrary. Ice loss in the Wilkes basin in East Antarctica also contributes significantly to the barystatic sea-level rise but GIA has a large effect in this region because of the relatively low mantle viscosity (basin 14 in Supplementary Fig. 2). ELRA300 provides a very good fit for this basin. In West Antarctica, the contribution differs per basin, but the effect of GIA is significant in almost all basins due the relatively low mantle viscosity at the present-day grounding line of the West Antarctic Ice Sheet (basins 1, 18, 18, 21 and 22 in Supplementary Fig. 2). Here, ELRA300 somewhat underestimates the effect of GIA but still provides a stabilising effect. For the high emission scenario, ELRA300 underestimates sea-level rise by 0.4 m (6%) in 2500 compared with 3D-Average (Fig. 4a). When there is a larger region of ice mass loss, as is the case in the high emission scenario compared with the low emission scenario, the bedrock deformation is more sensitive to the rheology of deeper parts of the mantle, where the viscosity can be up to 3 orders of magnitude greater than at shallower depths. This causes the same effect as in the low emission scenario – a slowdown of the uplift projected by the 3D model on longer timescales – but the effect is even stronger. The relaxation time of ELRA300 is therefore too short compared to 3D-stronger and 3D-weaker

on the long-term, leading to faster uplift and a higher bedrock elevation by 150 m in 2500 (Supplementary Fig. 3). However, around 2300, uplift in the 3D model has not slowed down much and is faster than the uplift of ELRA300. Furthermore, the elastic response of the upper mantle is not taken into account in the ELRA model, which could lead to an underestimation of uplift compared to the viscoelastic mantle response in the 3D model. Therefore, at this moment in time, the ice is about 750
meters thicker in 3D-weaker compared with ELRA300 and the grounding line has retreated about 100 km less in the Amundsen Sea Embayment (Supplementary Fig. 4). The slowdown of bedrock uplift is less strong when retreat is concentrated in the Weddell Sea Embayment (using climate model CESM) due to less vertical variation in mantle viscosity in this region (Fig. 4e,f and Supplementary Fig. 5).

In the ELRA model, the elastic response of the lithosphere is computed using the flexural rigidity of the lithosphere. The
lithospheric beneath the West Antarctic Ice Sheet can be as thin as tens of kilometers (Lloyd et al., 2019). We therefore test the impact of using a flexural rigidity of $1.92 \cdot 10^{24}$ km·m$^2$ /s$^2$, which roughly correspond to a lithospheric thickness of 60 km. The combination of a lower flexural rigidity and higher relaxation time yields a similar result to the combination of a higher flexural rigidity and somewhat lower relaxation time. Therefore, decreasing the lithospheric thickness does not improve the fit of ELRA to the 3D Average (Supplementary Fig. 6).

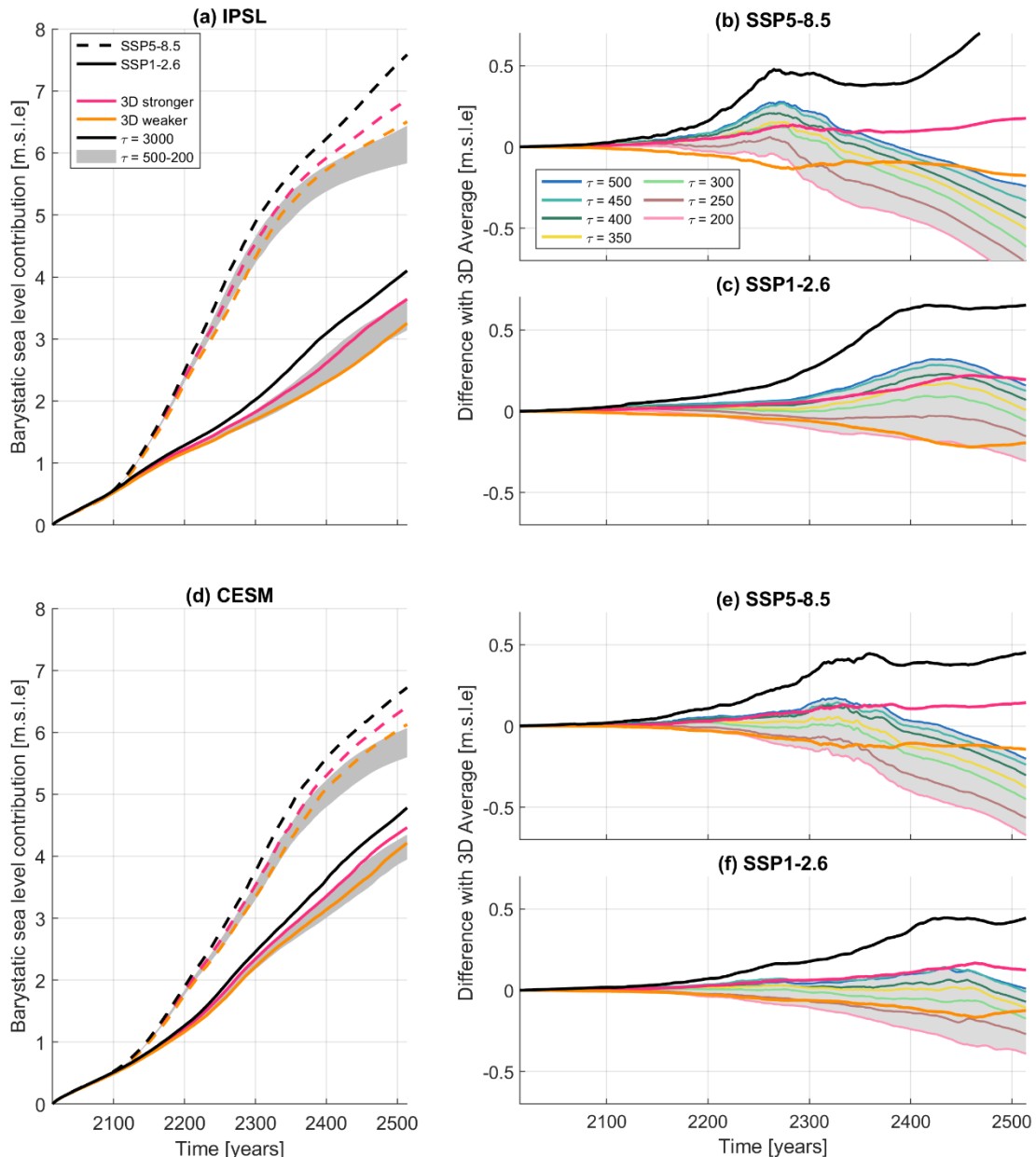

**Figure 4. The AIS contribution to barystatic sea-level rise using the 3D GIA model and ELRA for a high and a low emission scenario and two different climate models, IPSL-CM6A-LR (panel a) and CESM2-WACCM (panel d). Two different Earth structures are applied in the 3D GIA model, a stronger Earth structure and a weaker Earth structure. The relaxation time of ELRA is varied between 200 and 500 years, and a reference run of 3000 years is used. The flexural rigidity of $10^{25}$ N·m roughly corresponds to a lithospheric thickness of 100 km. Panels b, c, e, and f show the difference in barystatic sea-level contribution between ELRA with different relaxation times and the average sea-level contribution of the two 3D GIA simulations.**

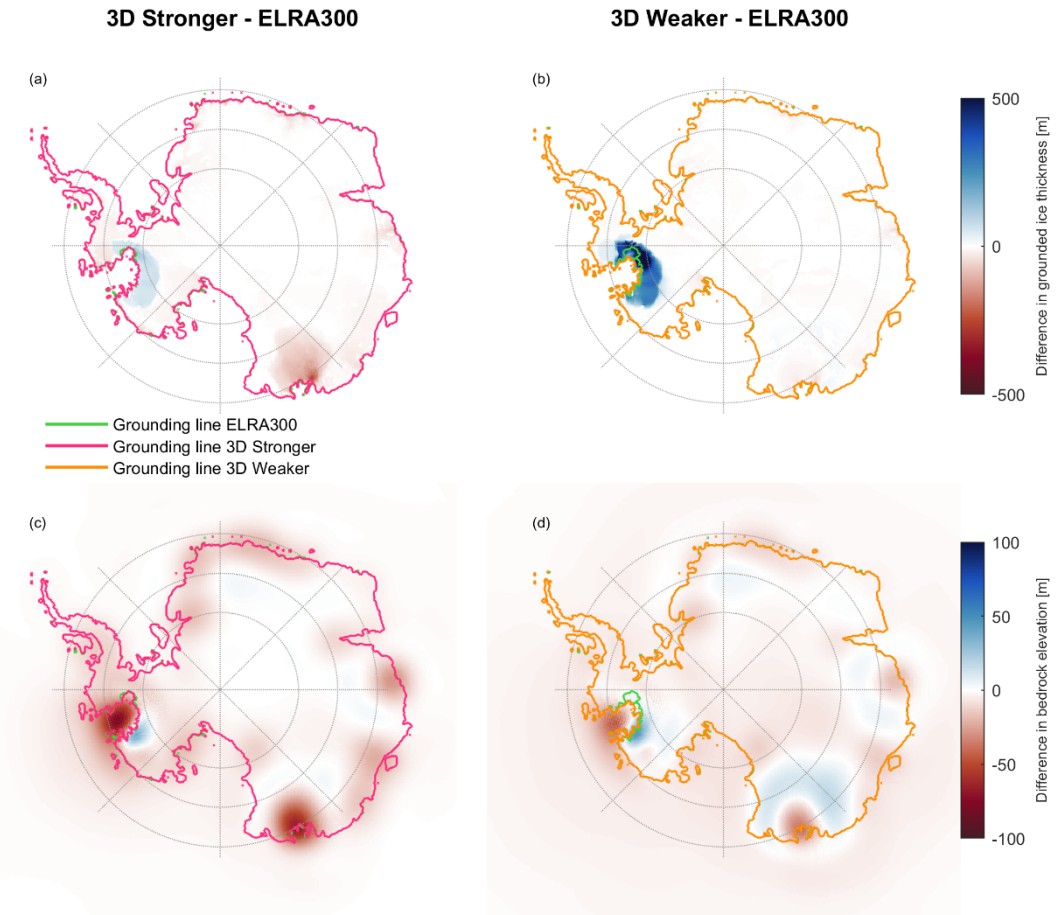

**3D Stronger - ELRA300**  **3D Weaker - ELRA300**

**Figure 5: Difference in grounded ice thickness above flotation (panel a and b) and bedrock elevation (panel c and d) in 2500 between ELRA with a relaxation time of 300 years (referred to as ELRA300) and the two 3D Earth structures. Panels a and c correspond to 3D-stronger and panels b and d to 3D-weaker. The climate model IPSL is applied for the low emission scenario SSP1-2.6.**

**4.2 ELRA model with 2D laterally variable relaxation time**

Previous studies have shown that a laterally varying Earth structure is needed to accurately simulate AIS evolution (Gomez et al., 2024; van Calcar et al., 2025). As these 3D GIA simulations are very costly, they prohibit large ensemble simulations. We therefore assess the performance of a 2D relaxation time, which is straight forward to implement in an ELRA model, to answer research question 2: What is the best parameter choice for a coupled ice sheet – ELRA model using laterally varying

relaxation time (LVELRA) to approximate the ice sheet evolution resulting from the reference model?

We combined 4 different laterally varying relaxation time maps with different uniform flexural rigidities to investigate how well the computationally efficient ELRA model can replicate the results of the 3D models. As there is no *a priori* reason to select the average or lower bound viscosity equations, or a particular flexural rigidity, we investigate which of the resulting

ice sheet evolutions using the 2D maps correspond best to ice sheet evolution using the 3D-Average, and whether the improvement is significant compared to the performance of ELRA300. The different relaxation time maps, in combination with different lithospheric thicknesses, result in a large range of sea-level rise projections (Fig. 6).

The 2D-stronger map, when combined with a flexural rigidity that corresponds to a lithospheric thickness of 120 km and derived from the average viscosity (Eq. 7), has the smallest RMSE compared to the 3D average, considering both climate models and emission scenarios (Supplementary Tab. 2), and will be considered in the following. For the high emission scenario, the sea-level rise is about 30-40 cm closer to 3D-Average at 2500 using 2D-stronger compared to using ELRA300 (Fig. 6a-b). The advantage of using 2D-stronger over ELRA300 is particularly significant in the Amundsen Sea Embayment projections for scenarios longer than 400 years because the difference between 3D-Average and ELRA300 increases strongly after 2300, whereas the difference between 2D-stronger and 3D-Average is constant over time (Fig. 6a).

On the one hand, the bedrock uplift in the Amundsen Sea Embayment is overestimated by about 250 meters by 2500 when using 2D-stronger compared with using the 3D GIA model in SSP5-8.5-IPSL (Supplementary Fig. 7). However, this uplift occurs mainly in the last 100 years. Furthermore, the effect on grounding line retreat is small because the grounding line is already retreating rapidly and the negative feedback from bedrock uplift is not strong enough to slow the rate of retreat. For another ice sheet simulation with different melt approximation or melt parametrization, the sensitivity to a similar uplift might be relatively larger. On the other hand, the uplift is underestimated by up to 60 meters using 2D-stronger compared to using the 3D GIA model in Wilkes basin in East Antarctica. The relaxation time in this area in 2D-stronger is too long to sustain the fast uplift of the 3D GIA model, and the ice mass loss is relatively sensitive to bedrock uplift.

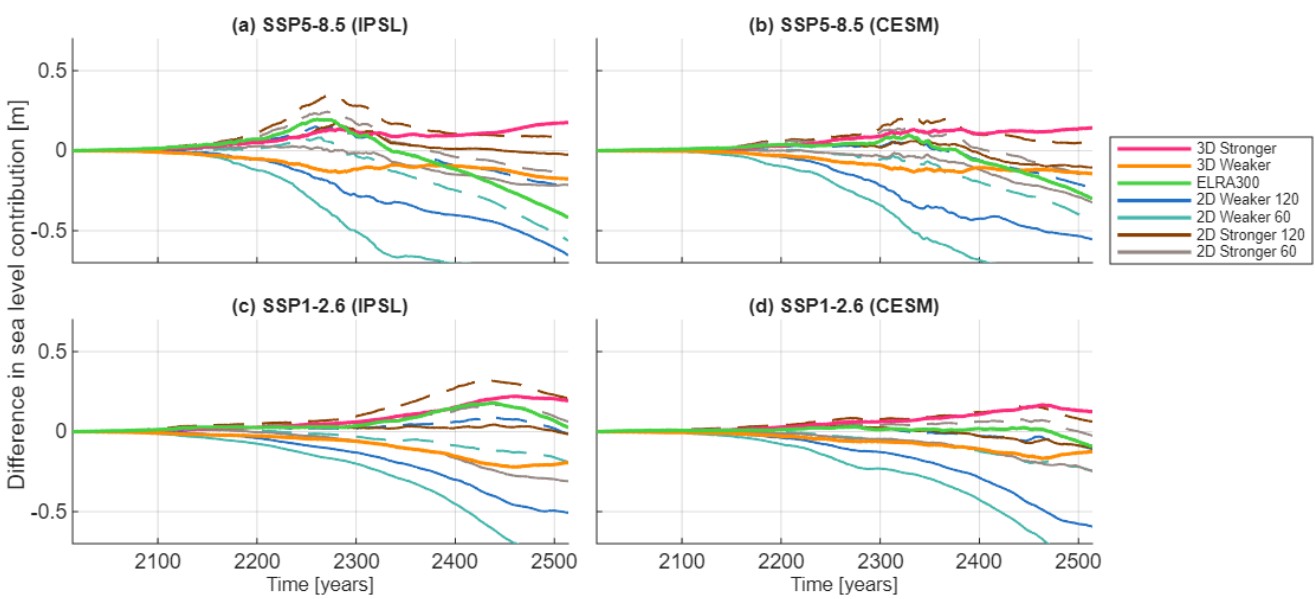

**Figure 6: The difference in AIS contribution to barystatic sea-level rise between the average sea-level contribution of the two 3D GIA simulations and the contribution using different 2D relaxation time maps. Results are shown for a high and a low emission scenario and two different climate models, IPSL-CM6A-LR and CESM2-WACCM. The solid lines refer to the relaxation time**

calculated from the average viscosity and the dashed lines refer to relaxation times calculated from the lower bound viscosity (Eq. 7 with corresponding parameters). The numbers 120 and 60 in the legend refer to the lithospheric thickness.

### 4.3 1D GIA

To conduct projections, some ice-sheet models are coupled with a 1D GIA model (Gomez et al., 2015) or with a bed deformation model using a lithosphere underlain by a viscous half-space (Golledge et al., 2015; Konrad et al., 2015; Kachuck et al., 2020; Rodehacke et al., 2020; Klose et al., 2024). The 1D GIA model and bed deformation models are more realistic than ELRA because it takes into account the radial depth variation of viscosity, which implies a variable relaxation time as the size of the load determines which part of the radial viscosity profile controls the response. As these models can also be

considered intermediate in terms of computation time compared to ELRA and 3D GIA, we study whether 1D Earth structures offer an improved accuracy compared to ELRA models to answer research question 3: What is the best parameter choice for a coupled ice sheet – 1D GIA model to approximate the ice sheet evolution resulting from the reference model?

A widely used mantle viscosity for a 1D Earth model is $10^{21}$ Pa·s (hereafter referred to as 1D21) (Gomez et al., 2015; Konrad

et al., 2015; Rodehacke et al., 2020; Golledge et al., 2019). Figure 7 shows that 1D21 overestimates the Antarctic sea-level contribution by 0.4-0.6 m (6-17%), depending on the emission scenario and climate model, because the structure is too stiff in West Antarctica compared to the 3D structures. The viscosity profiles 1DASE and 1D19 produce results similar to each other and to the 3D-Average model for the low emission scenario but, like ELRA300, they still underestimate the sea-level contribution by 0.3 m (4%) in 2500 for the high emission scenario. The viscosity profile 1D19 has the smallest RMSE

compared to the 3D Average (Supplementary Tab. 3).

The largest improvement of 1D19 compared to ELRA300 and 2D-stronger is in the bedrock uplift. The bedrock elevation of 1D19 in 2500 differs by a maximum of 80 meters from the results of the 3D GIA modelling in the high emission scenario, which is significantly smaller than the difference of 250 m when 2D-stronger is compared with the 3D GIA model output (Supplementary Fig. 8). This improved agreement is likely explained by the more complete representation of Earth rheology

in the 1D GIA model compared to the ELRA approach. While ELRA prescribes a simplified elastic lithosphere and a purely local, exponential relaxation toward isostatic equilibrium, the 1D model captures the full viscoelastic response of the Earth, including both elastic and time-dependent viscous deformation. Although it does not account for lateral variations in Earth structure, the 1D model with low viscosity still resolves the mantle's flow in response to loading, bringing it closer to the behavior captured in 3D GIA models. To replicate not only the sea-level contribution from 3D-Average, but also the geometry

of the bed, it can therefore be recommended to use a 1D GIA model with an upper mantle viscosity of $10^{19}$ Pa·s instead of ELRA with a uniform relaxation time of 300 years or the 2D-stronger relaxation time map. Especially for long-term projections under a high emission scenario, the 1D GIA model is preferred over an ELRA model with a uniform relaxation time. However, considering both scenarios and climate models, the 2D-stronger relaxation time map contains the smallest total RMSE over

both scenarios and climate models of only 0.17 m compared to a total RMSE of 0.25 m or 0.35 m for 1D19 and ELRA300,
respectively (Supplementary Tab. 1, 2 and 3).

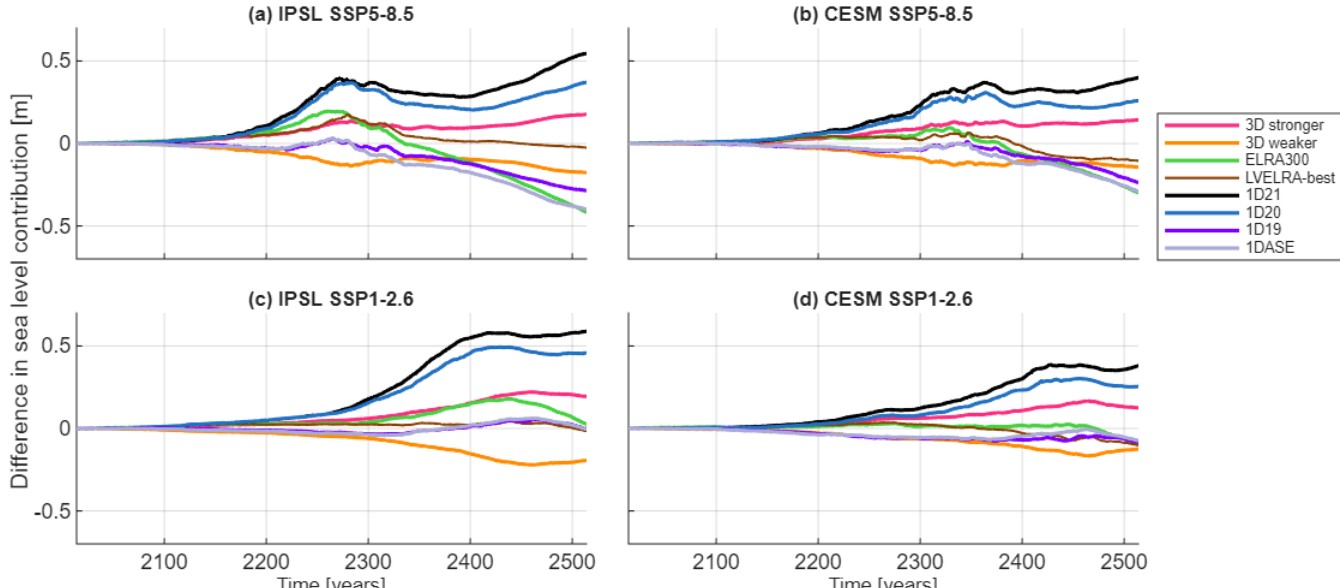

**Figure 7: The difference in AIS contribution to barystatic sea-level rise between the average sea-level contribution of the two 3D GIA simulations and the contribution using different 1D Earth structures. Also the best fitting ELRA with uniform relaxation time**
**(ELRA300), and laterally varying relaxation time (LVELRA-best) are shown. LVELRA-best refers to the best fitting relaxation time map which is the 2D-stronger map combined with a flexural rigidity that corresponds to a lithospheric thickness of 120 km and derived from the average viscosity. Results are shown for a high and a low emission scenario and two different climate models, IPSL-CM6A-LR and CESM2-WACCM. Note that the 1D19 and 1DASE mostly overlap for IPSL SSP1-2.6.**

## 5 Conclusions and outlook

Using forcing from two climate models under low and high emission scenarios, we investigated the accuracy of common implementations of bedrock displacement in an ice-sheet model by comparison with a coupled ice sheet-3D GIA model. The ELRA model with a commonly used uniform relaxation time of 3000 years combined with a uniform flexural rigidity overestimates sea-level rise by up to 0.7 m (20%) compared with the average barystatic sea-level rise predicted using a model that includes 3D Earth structures. A 1D GIA model with an upper mantle viscosity of $10^{21}$ Pa·s overestimates sea-level rise by
up to 0.6 m (17%). To replicate the sea-level rise of the average of the 3D GIA models (3D Average) better, a relaxation time or mantle viscosity corresponding to the area of ice mass loss should be chosen. We investigated the degree to which different bedrock models, Earth structures and parametrisations can replicate the bedrock uplift predicted by models that include 3D Earth structures.

Research question 1 was: What is the best parameter choice for a coupled ice sheet – ELRA model using uniform relaxation
time to approximate the ice sheet evolution resulting from the reference model? If the ELRA model with a uniform relaxation

time were to be used, we recommend using a uniform relaxation time of 275-325 years with a lithospheric thickness of 100 km to replicate the sea-level rise predicted by a model that includes 3D Earth structure. Using this relaxation time results in a sea-level rise that differs from 3D-Average by 0.03-0.4 m (0.8-6%), dependent on the emission scenario and the climate model.

Note that using this relaxation time does lead to an increasing underestimation of sea-level rise from 2400 onwards due to the
evolving location and area of ice mass loss which leads to deformation in different regions and influenced by different depths of the Earth's mantle. Even though the sea-level rise can be similar between ELRA300 and 3D-Average, the ice thickness can locally differ by up to 750 meters, the grounding position may differ by 100 km, and the bedrock elevation may differ by 150 meters between ELRA300 and the different 3D Earth structures. It is therefore recommended to vary the uniform relaxation time between 300 and 500 years to approximate the uncertainty from the 3D Earth structure.

The reduction in sea-level rise when using the optimal choice of uniform relaxation time compared to rigid Earth is independent of the total sea-level rise in 2500, which means that using the optimal relaxation time causes a larger relative reduction in sea-level change when the total sea-level rise is lower, for example, in projections from other ice-sheet models less sensitive to climate forcing. On millennial timescales, significant ice mass loss might occur in the East Antarctic Ice Sheet (Coulon et al., 2024). While low mantle viscosities of $5 \cdot 10^{18}$ Pa·s might exist in some regions around the present-day grounding line, the
viscosity increases up to 6 orders of magnitude inland (Fig. 3a,d). If the majority of the sea-level contribution would originate from the East Antarctic Ice Sheet, a larger relaxation time might be necessary. We stress that the relatively high root mean square error of ELRA with a uniform relaxation time can be significantly reduced by using LVELRA and 1D GIA models, which are the preferred models.

A spatially varying relaxation time can easily be included in ELRA by directly using a 2D array instead of a single value. We
derived an empirical relation between upper mantle viscosity and relaxation time and computed 2D maps of relaxation times to answer research question 2: What is the best parameter choice for a coupled ice sheet – ELRA model using laterally varying relaxation time (LVELRA) to approximate the ice sheet evolution resulting from the reference model? Applying the 2D-stronger map, derived using the relation between average viscosity for a strong 3D rheology and relaxation time, and a lithospheric thickness of 120 km, results in a sea-level rise projection that differs from the 3D-Average value by only 10 cm
in 2500. This difference doesn't increase on the long term in contrast to ELRA and it can thus be recommended to use ELRA with spatially varying relaxation time for long term simulations. Still, the bedrock elevation in 2D-stronger is hundreds of meters too high by 2500 compared to the 3D model under a high emission scenario.

For models that are able to use a 1D GIA model, we answer research question 3: What is the best parameter choice for a coupled ice sheet – 1D GIA model to approximate the ice sheet evolution resulting from the reference model? The use of an
upper mantle viscosity of $10^{19}$ Pa·s results in sea-level rise projections that only differ from 3D-Average by a maximum of 0.3 m. The bedrock elevation in 1D19 differs from 3D-Average by a maximum of 80 meters, thus this model provides the closest resemblance to the 3D Earth structures in terms of geometry, better than the ELRA and LVELRA models. However, the improvement should be traded off against a large increase in computation time. Our recommended values for the relaxation time and 1D viscosity will provide a better approximation of sea-level rise than the currently used standard values but should

be taken as guidelines and not as the true relaxation time or viscosity of the Earth's mantle. The simplified Earth models are all compared to the same coupled ice sheet - 3D GIA model and this model did not include the effect of a local sea-level drop on ice sheet retreat. Including the feedback of the sea-level drop on the ice sheet dynamics reduces the sea-level rise by 5% compared to using a fixed sea level (van Calcar et al., 2025). Furthermore, using the suggested upper mantle viscosity would lead to an overestimation of the response to changes in global ocean loading and to changes in ice loading in East Antarctica

over millennial timescales.

Finally, the sea-level projections are relatively high compared to literature (Seroussi et al., 2024). A different calibration of the ice-sheet model, or a completely different ice-sheet model could lead to lower projections of sea-level contribution. We include a scenario leading to a collapse of the West Antarctic Ice Sheet in 2500 (Supplementary Fig. 3, 7 and 8), and a scenario which does not lead to collapse (Fig. 5). The difference in grounding-line retreat between these scenarios means the ice sheet is

sensitive to a somewhat different part of the mantle, which leads to a small difference in preferred relaxation time. If a low emission pathway or a more muted dynamical response, for example a situation in which MISI is weak or does not progress substantially, were to lead to only limited grounding line retreat compared with our simulations, the influence of solid Earth deformation in that region would likely be minor, and the choice of Earth structure would have little effect on the results. Hence, the preferred LVELRA and 1D GIA models are also expected to remain applicable.

The laterally varying relaxation time is dependent on the 3D viscosity structure so different 2D relaxation time maps could be produced using the provided relation between relaxation time and viscosity. This allows other modellers to create their own relaxation time maps based on their preferred 3D viscosity profiles, for example based on different seismic models, a different time period such as the deglaciation since the last glacial maximum, or for other regions such as Greenland.

### Code and data availability

The supplementary data, i.e. Table 1 and the laterally varying relaxation time maps, are publicly available with DOI 10.4121/a7215d4c-767f-49f1-a8bb-da40d0d2b01d. The data produced for this publication is available via DOI 10.4121/b5548aaa-4c05-45f7-b0ce-775b83f13e5d. The source code of IMAU-ICE is included in this DOI and can be found on Github: https://github.com/IMAU-paleo/IMAU-ICE. The GIA model code and coupling script has been made publicly available by van Calcar et al. (2023) with DOI 10.4121/19765816.v2.

### 610 Author contributions

The conceptualisation was done by CvC, WvdW and RvdW. CvC, PW, and WvdW conducted model development and the experiments, and performed the data analysis with input from RvdW. All authors contributed to the writing of the manuscript.

## Ethics declarations

There are no competing interests.


## Acknowledgements

The authors would like to thank Ann Kristin Klose and Violaine Coulon for preparing and providing the forcing data of the climate models. We thank Grace Nield, Matt King, Terry Wilson and Doug Wiens for discussions regarding the computation of relaxation times, and Dirk Oude Egbrink for his master thesis work which provided the first insights into the effect of a laterally varying relaxation time. We thank the reviewers for their thorough and constructive feedback, which has significantly contributed to improving the clarity of the manuscript. Work for this publication was performed in the framework of PROTECT, which received funding from the European Union's Horizon 2020 Research and Innovation Programme under grant agreement No 869304. This is PROTECT publication number xxx (defined upon acceptance). The study was also supported by the project 3D Earth funded by ESA as a Support to Science Element (STSE), and by the UK Natural Environment Research Council as an Independent Research Fellowship (NE/K009958/1).

## Additional information

Correspondence and requests for materials should be addressed to Caroline van Calcar (c.j.vancalcar@tudelft.nl).

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
