# Peer review of "Approximating 3D bedrock deformation in an Antarctic ice-sheet model for projections"

_EGUsphere, 2024_

## Referee Comment (RC1)

**Introduction**

The authors made very clear that the target of the present article is to recommend:

1. a uniform relaxation time scale to use in ELRA,
2. a laterally-variable relaxation time scale to use in ELRA,
3. a radial viscosity profile to use in a 1D GIA model.

In a context where these GIA models are coupled to an ice-sheet model, these recommendations aim to reduce the discrepancies in sea level contribution from the Antarctic ice sheet compared to the use of a 3D GIA model (with 2 different plausible Earth structures, which is a well-appreciated detail of the work). This is specifically done within the frame of possible climate forcings until 2500, as computed by 2 GCMs (IPSL and CESM). The potential impact of the paper is therefore also very clear: offer a low-hanging fruit to ice-sheet modellers that want to improve their representation of GIA simply by adjusting the parameters that are used in their already-implemented GIA model. This effectively represents 1-10 seconds of work and is therefore very appealing. Additionally, the manuscript reads well and it is unambiguous that it required a lot of work, including many expensive simulations. After this praise, I unsurprisingly think that this is a very valuable endeavour and am generally in favour of seeing this article published. Nonetheless, I think that there are a series of important points that need to be addressed before this. I will now proceed to listing them with the following scheme:

- Primary comments: these are key points that need to be addressed in a very comprehensive and thorough manner before a possible publication.

- Secondary comments: these are points that are either curiosities of mine and/or easy improvements that can be made to the manuscript.

- Technical comments: these are related to phrasing, format, etc.

- Additional references that come up along the way are gathered at the end of the document.

I want to emphasise that the length of the upcoming points is motivated by the fact that I am eager to improve the quality of this article as much as I can, since it will likely be highly relied upon in future within the field of ice-sheet modelling.

I will assume that the authors are familiar with common abbreviations from ice-sheet and GIA modelling.

Disclaimer: I only read the editor's initial decision statement after writing down my comments.

**Primary comments**

1. The authors spend very little time explaining the details of the ice-sheet model that are of true importance for this work. For instance, mentioning the basal friction law (l. 110) is here of secondary importance compared to questions like:

    a. Is the potential of a marine ice-cliff instability included in the runs?

    b. Does IMAU-ICE accept a heterogeneous, time-varying sea level? For that matter: is the coupling to the 3D GIA model merely deformational or does it also include the effects of rotation and gravitation?

    c. What type of melt parametrization is applied at the grounding line (Leguy et al. 2014)? Is any flux imposed? Using a resolution of 16 km is in my opinion sufficient to go ahead with the present publication but it needs to be accompanied by such information on the subgrid parameterization.

2. In the present work, SSP8.5 gives an SL contribution of 3.5-4.5 metres by 2300 (depending on the GCM and the 3D Earth structure). This makes IMAU-ICE more sensitive than any other model included in Seroussi et al. (2024, c.f. Fig. 4, especially panels a and b), which were forced, among others, by the 2300 projections of CESM which offers a straightforward comparison to the present work. In Seroussi et al. (2024), the ensemble of SL contributions under SSP8.5 spans -/+ 0.2 m of SL rise by 2100, which is a substantial difference to the value of >0.5m obtained in the present work. Adding to this, Coulon et al. (2024) and Klose et al. (2024) present SL contributions of the AIS that are nominally less than 3 m by 2300, although you are using the same climate forcing.

    This is somewhat surprising, because IMAU-ICE is the only one of these models that is coupled to a 3D GIA model, which has stabilising effects on the grounding line and should therefore lead to a lesser sensitivity. I think this needs to be mentioned and, in the best case, explained comprehensively. Answering the points I raise in 1.a.-1.c might be a step in this direction.

3. I believe that the aforementioned high sensitivity of IMAU-ICE might slightly undermine the message of the article: if the positive feedbacks that are internal to (marine) ice sheets are particularly strong in IMAU-ICE, the negative feedback of GIA might be underestimated in comparison. I believe that the simulations you present show a few symptoms of this:

    a. Looking at Fig. 4, the error with respect to the 3D runs is almost the same for ELRA3000 and ELRA200 - although ELRA3000 is far from representing the reality in WAIS (Barletta et al. 2018, among many others). In essence, this means that the difference to the average 3D model is relatively insensitive to the choice of τ. Just so that you get me right: your optimal choice of τ would be the same with an ice-sheet model of lesser sensitivity and, in that sense, it does not change your final recommendation… but the impact of this optimal

choice would be higher. I think you should consider this, since it might imply a larger reduction of the relative error in SL contribution thanks to your recommendations and could make your message stronger.

b. The large uplift discrepancy of 250 m that you mention at l. 369 has almost no impact on the result. Explaining the little impact of this difference on SL contribution by invoking the fact that it arises relatively late in the simulation is a partially albeit not fully satisfactory explanation to me. I believe the (potentially too) high nonlinearities that are internal to the ice sheet model contribute to it.

4. This large discrepancy of 250 metres makes me wonder how the change of the ocean mask and the associated change in load is treated in your implementation of ELRA. When ice melts in zones grounded below sea level, it is replaced by the ocean which largely compensates for the reduction in surface load due to ice melt. Is this accounted for here? If not, this might be an explanation for such high error values of ELRA – which arise even for a 2D relaxation time field! This point illustrates a deficiency of the article, which does not mention at all what is the coupling scheme between IMAU-ICE and the various GIA models. This can however be easily fixed with a couple of additional lines in the manuscript.

5. I know very little about IPSL but CESM is known as a quite high-sensitivity model. I think this is a rather good thing since it leads to ice-sheet configurations that show large differences to present-day, thus exploring more diverse AIS configurations. I think this could be highlighted in a discussion section, especially since it supports the overall message of the paper. I would very much appreciate more information about the GCM outputs in the supplemental material: e.g. their present-day bias compared to RACMO, MAR or ERA5, and their warming pattern (atmospheric and oceanic) by 2300.

6. In Fig. 4, IPSL gives a barystatic sea level contribution of 3 to 4 metres by 2500 under SSP2.6, which roughly corresponds to the total sea-level equivalent volume of the WAIS (i.e. only taking into account ice above floatation + corrections). In the corresponding maps shown in Fig. 5, the WAIS is however far from a total collapse, since the Ross and Filchner-Ronne grounding lines are essentially unaffected compared to present-day. I suspect two possible explanations for this discrepancy:

a. The ice sheet has experienced a thinning throughout its domain, including EAIS (otherwise such values are impossible).

b. The authors are showing the ice volume that was lost (including ice grounded below sea level) and NOT the actual contribution to barystatic sea-level. If this is the case, this might of course partly resolve the doubts I raise in 2. and 3.

7. Questions around the spatial resolution of models can easily become generic. I hope this does not apply to the one I will try to raise here. The "high-resolution" configuration of your 3D GIA model has a lateral resolution of 200 km, which is refined to 35 km in the WAIS. I believe this is more than sufficient if one wants to

study mechanisms in a qualitative way, e.g. "how does rheology affect the grounding line retreat?". However, the goal of the present study is quite different, namely to make a recommendation of parameter choice to minimise the error of simpler GIA models compared to the averaged 3D results. This is a much more quantitative assessment, which essentially places your 3D GIA model as a "golden standard". The recent work of Gomez et al. (2024), which uses a resolution of 3 km for the GIA model and 5 km for the WAIS (somewhat coarser outside), places a quite high target to what can be called a "golden standard" nowadays. I am not arguing that such a resolution is needed for the present study. However, if you want to keep the resolution as it is you need to show a convergence of some meaningful metric over (lower) resolutions. Alternatively, I'd be happy with a 16km-100km combination for the lateral resolution of your 3D GIA model or with any argument that shows that your resolution is high enough. I know this type of comment is the authors' nightmare because it implies repeating runs that can become substantially longer. However, I hope that the authors understand that this expectation comes from the potential high impact of the paper.

8. Along the same idea, l. 149-150: "the GIA model is used with 9 vertical layers (0-35 km, 35-100 km, 100-150 km, 150-300 km, 300-420 km, 420-550 km, 550-670 km, 670-1171 km, and 1171-2890 km, and 2890-6371 km)." It seems like you used the same set-up in van Calcar et al. (2023) – which is fair in that case since you are not making direct parameter recommendations. Unless I am getting something wrong, this sounds very coarse to me, for instance compared with what was used already some time ago in Gomez et al. (2018), where the vertical layers have a thickness of 6–50 km (only down to the CMB). Another example with a different model can be found in Albrecht et al. (2024), where "$\Delta z$ = 5 km down to 420 km depth, followed below by $\Delta z$ = 10 km down to 670 km and $\Delta z$ = 40 to 60 km down to the core–mantle boundary": The use of a much coarser resolution here presents two problems:

   a. A potentially inaccurate representation of the mechanics. For instance, even for an idealised constant viscosity over depth, you might obtain significant errors compared to the actual solution. In particular, does a layer from 2890 down to 6371 km depth make any sense in that context?

   b. An inaccurate representation of the parameter field. If you have fast variations of viscosities over depths, as it is the case in ASE, averaging over such thick layers might lead to an additional error when resolving the mechanics.

   Just as for the lateral resolution, I wish to see at least some convergence/refinement arguments to fully approve a paper that recommends parameters to the community.

9. The explored range of ice-sheet configurations is essentially limited to a collapse of the WAIS (and parts of EAIS basins?) compared to present-day. If a different configuration of the ice sheet is being studied, the results shown here might be leading to large errors. For instance, performing the same task with simulations in the distant future would lead to a higher optimal uniform relaxation time (or 1D viscosity profile) since the bulk of contribution to sea level would come from EAIS, where the viscosities are even higher. This is illustrated in many of your plots, where it is clear

that the error has a pronounced drift beyond 2500, giving an advantage to slower responding models. This does not undermine your message at all but needs to be discussed.

10. Swierczek-Jereczek et al. (2024) present FastIsostasy, a regional GIA model that (1) is computationally as efficient as ELRA, (2) directly takes laterally-variable viscosity and lithospheric thickness as input, (3) displays different relaxation time depending on the wavelength of the load, (4) includes a regional approximation of gravitational feedbacks on sea level and (5) displays errors that are low compared to a 3D GIA model, which was tested with idealised as well as realistic ice histories. Citing this paper was omitted, in spite of its direct relevance for the aim of the present paper. It however seems difficult to circumvent because it changes the emphasis of the message that is conveyed in the present article:

   a. **Computationally efficient ways to represent lateral variability with low error compared to a 3D GIA model already exist** which contrasts with the statement at l. 45-47. Besides FastIsostasy, 3D regional models like those presented in Weederstijn et al. (2023) and Nield et al. (2018) are computationally tractable when considering simulations only until 2500. **The main advantage of the present work is to avoid coupling to a different GIA model, which can sometimes require quite a lot of programming effort.** I insist that this is the main contribution of the article, which is very useful but should be stated more precisely.

   b. If provided with a new seismic data set, the process of fitting the relaxation time scales (summarised in Fig. 2 and in Supplemental Tab. 1) might need to be repeated. You are in the best position to know how time consuming this is and the ice-sheet community will be grateful that you made this effort. However, it should be emphasised that repeating this effort can be avoided by using approaches like FastIsostasy.

   c. You mention that the impact of a laterally variable lithospheric thickness has only a marginal effect on the bedrock deformation (l. 212-213). I think this depends on the ice-sheet configurations that are explored, especially for future projections where the West-East gradient might become more relevant. Of course, including a laterally variable LT is more complex to implement… but it turns out it was already done in FastIsostasy without noticeable increase in the computational cost. Since the software is open source (in fortran and julia), **the only practical obstacle to including this is the coupling effort.** I think it should be discussed that the latter is not more tedious than exploring different combinations of 2D relaxation times and constant lithospheric thicknesses (e.g. your work summarised in Fig. 6).

   d. As you mention at l. 223-224, ELRA displays exactly the same relaxation time, regardless of the extent of the mass anomaly. Because of this, it would be particularly important to discuss the geometry of the schematic load that you apply to the 3D model to derive the relaxation time. What horizontal extent does the load have? Did you vary this to see how it impacted the final

result? From Tab. 1 in the Supplement I gather you did that (small, large and LGM), but this is not mentioned explicitly although it makes the paper stronger!

As a side note, FastIsostasy partially fixes this since it does not present a flat relaxation spectrum.

11. You recommend a radial viscosity profile for the 1D GIA based on the behaviour of the AIS compared to the outcome with a 3D GIA model. The caveat of this is that this lower viscosity is then applied globally whereas we know that most regions have higher viscosities. In other words, your error metric is local, although your model is global... Imagine we select a random point at the coast of the Netherlands: the sea level rises because of AIS melting and floods the grid cell which subsides because of the new load. This response happens too fast with your viscosity recommendation! Again, this is not critical but needs to be discussed at least a bit.

12. The points I raised so far make me think that a discussion section between the results and the conclusions is absolutely necessary!

13. I acknowledge the intention of the authors to provide all the data that resulted from the study in the supplemental material. This is mostly raw, ~100Gb data, which has a value if anyone wants to reproduce the analysis, but which is also anything but visual and hence of little use to the average reader. I encourage the authors to provide supplemental material that is of importance for the present study while being directly digestible, such as maps of the initial state of the ice sheet with an error plot with respect to to present-day, as well as maps of the ice thickness anomaly at 2500 compared to present-day for SSP2.6 and SSP8.5 (for at least one of the 3D runs). For instance, the doubt I raise in 5. could be answered directly by such a plot.

**Secondary comments**

1. The word "bedrock" is used throughout the manuscript with a very "GIA perspective" on things. Bedrock response to ice-sheet forcing includes, for instance, erosion and sedimentation - which is obviously not your current object of study. I suggest you change this to something less ambiguous throughout the manuscript like "bedrock (vertical) deformation". In particular I think this would make the title more specific. Something like "Approximating the coupled bedrock displacement in Antarctic ice sheet projections". Since the main result of your work is the parameter recommendation, you could even include this idea in the title.

2. L. 23-24: You mention a difference in SL rise that "deviates less than 40cm". Without knowing what is the total contribution to SL rise, this number is not very meaningful. In particular, it might sound like a huge amount (which it is in terms of policy advising!) if not cited as relative value.

3. L. 29: Citing the IPCC is of course pretty reliable but it is not the most up-to-date information. I believe you should at least add Seroussi et al. (2024), Coulon et al.

(2024) and Klose et al. (2024) here.

4. L. 30 "atmospheric and oceanic feedbacks". I would replace "feedbacks" by "processes" since it is more general. For instance, the warm water intrusion observed over the last decades in ASE is an oceanic process, not a feedback.

5. L. 43-45: the context of the numbers you are mentioning could be more precise. Over the next centuries = until 2500?

6. L. 46: I believe you can at least partially do that with the model presented in Swierczek-Jereczek et al. (2024).

7. L. 61: You are omitting Gomez et al. (2024) which contrasts with your statement.

8. L. 82 - 88: you repeatedly use "approximate the ice sheet evolution resulting from a coupled ice sheet - GIA model using a 3D Earth structure" in the three questions. I think it would read better if you define this before as your "baseline" and refer to it as such later.

   I believe the three questions you pose are erroneously formulated. This appears particularly in your conclusions: "Research question 1 was: How well can a uniform relaxation time approximate the ice sheet evolution resulting from 3D Earth structures? We recommend to use a uniform relaxation time of 300 years with a lithospheric thickness of 100 km to replicate the sea level rise predicted by a model that includes 3D Earth structure." The answer and the question don't match! You ask "how well?" and you answer "the best choice is". Your question is actually: what is the best parameter choice for the near future? Of course, this comes with an error analysis ("how well?"), but the final outcome you want to convey to the community is the improved choice of relaxation time for ELRA and viscosity profile for 1D GIA models.

9. L. 82 and following: do you have any reference that describes the GRD effects as implemented in your 3D GIA model? I feel like the description of dislocation and diffusion creep parameterization is actually less important than this – although admittedly much more concise to write down.

10. L. 90: you introduce IMAU-ICE but previous work of yours mentions ANICE and the code provided in the supplemental material also uses this name. Is there any difference between both apart from the name?

11. L. 95: "and one based on a constraint in the Weddell Sea Embayment and Palmer land in the Antarctic Peninsula." I don't really understand why you use the Weddell Sea Embayment and Palmer land as a constraint, since they are not particularly relevant for future sea level rise. Because we have better GNSS constraints in this region?

12. L. 145-147: do you have a reference where it is shown that both things are equivalent? As an analogy from ice-sheet modelling to support my question: tuning

the friction law or tuning the friction coefficient field can have similar impacts but are actually quite different conceptually and may lead to different sensitivities.

13. L. 182: I understand this means that the rotational feedback and the shoreline migration are not taken into account? This should be stated explicitly.

14. L. 186: ELRA does not "depend" on a point load. The displacement is just obtained by a convolution of the response to a point load with the actual load.

15. L. 203: I suspect using a uniform 60 or 100 km lithospheric thickness for any experiment that implies a large-scale retreat in East Antarctica would give a pretty erroneous displacement pattern, since the lithosphere is much thicker in this region. This should be mentioned at some point in the discussion to prevent ice-sheet modellers from using it in such cases.

16. L. 301 and following: I feel like recommending a parameter value until 2400 and a different one if running until 2500 is a very odd thing to do – especially because it depends on your ice sheet model + climate forcing and is therefore not particularly robust. I think recommending a single nominal value for 2500 accompanied by an uncertainty range is a much more useful information for ice-sheet modellers, who can then choose to sample the uncertainties or not. This would be nicely illustrated if you provide some integrated error metric (as box or scatter plot) on the right y-axis of Figure 4 b, c, e, f, Figure 6 and Figure 7.

**Technical comments**

1. Some of your section titles have points after the number e.g. "5. Conclusions and outlook" and some not e.g. "4 Projections using different approaches to bedrock response". Also their indentation is not consistent.

2. The table and figure captions are all in bold fonts, which I believe is a formatting error.

3. I think that throughout the Copernicus journals, abbreviated names for figures, tables and equations should be used when placed in the middle of a sentence but they should be spelled out if at the beginning or at the end of a sentence. If I am not mistaken in this regard, you should correct l. 188, l. 234, l. 240, l. 297 (maybe I am missing some).

4. L. 19: I reckon that when spelling out ELRA a comma, a dash or a slash should separate "elastic lithosphere" from "relaxed asthenosphere". Apply also later in the manuscript.

5. I believe "ELRA" can be written without specifying "the ELRA model", since models are generally referred to by their sole name once introduced (e.g. "CESM shows high

equilibrium climate sensitivity" and not "the CESM model shows…").

6. L. 26: you use "AIS" without introducing the acronym… and then you inconsistently use "Antarctic ice sheet" and "AIS" throughout the manuscript.

7. L. 32: giving a numeric value for the uncertainties arising from comparison projects could support your argument very nicely here (e.g. Seroussi et al., 2024, Fig. 15).

8. L. 42: "specific region where the loss occurs" is a bit bit vague. I think it would be clearer if replaced by "the solid Earth properties of the region where the ice loss occurs"

9. L. 52: I would add some reference to the statement – at the very least LeMeur and Huybrechts (1996).

10. L. 56: I would delete "(the elastic modulus)", since "stiffness" is clear enough.

11. L. 69: "However, using…" should be reformulated to make clear that it refers to the work presented in Pollard et al. (2017).

12. L. 103-105: this is a bit tedious to read. I suggest the following rephrasing: "We use the projections of two climate models, CESM and IPSL, under a low emission scenario (SSP1.2-6) and a high emission scenario (SSP5.8.5). In CESM, the warming mainly occurs in the Weddell sea, whereas in IPSL, the warming mainly occurs in the Amundsen Sea."

13. L. 108: I would replace "ice dynamical" by "thermomechanically coupled ice model", which is more accurate.

14. L. 110: I guess that you use the nonlocal formulation of the melt law proposed in Favier et al. (2019)? I think this should be written down explicitly to not confuse it with the local formulation.

15. Equation 1 and 2 (and rest of the manuscript): it is unclear to me which logic is adopted by the authors when it comes to italic vs. normal font for mathematical symbols and indices. For instance incoherent notation of D and $L_r$ between l. 187-188 and Equation 3 and 4. Also Equation 6 and following text…

16. Equation 2: I think T(x, y) makes much more sense than the subscript notation you use, since the temperature is a function of the space coordinates. Mentioning what T(x,y) is is redundant between l. 133 and l. 136 (but as you wish… sometimes nice to repeat).

17. L. 131-134: you give numerical values for some parameters and for others not. Why not for all?

18. L. 169: this very short sentence could be fused with the former one.

19. L. 174: you introduce the two configurations. One is introduced with quotation marks and the other without. This should be coherent. Moreover, writing out "low resolution configuration" and "high-resolution configuration" is almost just as long as "configuration 1 / 2" and it has the advantage that the reader does not need to remember which is which.

20. Figure 1: pa → Pa in the x label

21. Equation 5: The subscript notation "$w_{i,j}$", "$b_{i,j}$" makes more sense than using parentheses here since it refers to matrix indices, as stated in the final part of the sentence.

22. L. 231-232: I don't get what the sentence "For each simulation, the displacement over time for each surface load/Earth model combination is computed for multiple time steps" should convey. That you don't have a single time step as output? Isn't that a bit obvious?

23. Figure 2 is great. You probably can leave out the distracting axis ticks and labels of the Antarctic map, since it is clear what is represented but I leave that to your choice.

24. L. 237: I believe "log of the viscosity" should be replaced by "logarithm of the viscosity" or "logarithmic viscosity".

25. L. 268-270: this sentence reads pretty bad. Why invoke the ice sheet model here if the only thing you say is that it evolves the ice thickness in time?

26. L. 280-284: the two sentences begin with exactly the same structure, which reads a bit odd.

27. L. 284: a small technicality but… isn't the barystatic contribution to sea level actually calculated within your GIA model based on the ice thickness?

28. L. 295: 0.44-07 m does not correspond to the percentage of 8-20%. Also, I would suggest giving the error separately for each scenario since it changes the value quite significantly.

29. I would suggest giving references, at some point, about IPSL and CESM and potentially about any publication describing the simulations you are using. I reckon the refs can be found in the Coulon and Klose (et al., 2024).

30. L. 301-304: you begin with the conclusion on your results. I would first begin with the detailed analysis you make at l. 307-330 and then conclude (what you actually do partly already and hence 301-304 is redundant).

31. L. 304: I could not find Supplemental Fig. 1.

32. Figure 4: "km·m2 /s2" is a weird unit to use when you can simply use "N m" as done before in the manuscript.

33. L. 366: the IPSL projections are not ASE projections. They are just projections, with more warming in ASE. I think this is not formulated clearly here.

34. L. 307, 368: I would just write "IPSL-SSP2.6" and "IPSL-SSP8.5" instead of lengthy formulations like "In the high emission scenario driven by the IPSL climate model" (which is wrong anyway because it is the high emission scenario that is used as input for IPSL).

35. L. 408: this sentence reads pretty bad. How about "Using forcing fields from 2 different GCMs under low and high emission scenarios, we investigated the accuracy of common implementations of bedrock displacement when coupled to an ice sheet model." Or something similar…

36. L. 432: 2D stronger 120 looks more like 20 cm error by 2500 as far as I can tell from Fig. 6, and not 10 cm as you state.

37. L. 454 and following: I wonder if "doi" should be replaced with "DOI" since it's an acronym.

I hope these comments will make the paper stronger and I wish the best of luck to the authors for the rest of the publication process.

Sincerely,

Jan Swierczek-Jereczek

**Additional references**

1. Leguy, G. R., Asay-Davis, X. S., and Lipscomb, W. H.: Parameterization of basal friction near grounding lines in a one-dimensional ice sheet model, The Cryosphere, 8, 1239–1259, https://doi.org/10.5194/tc-8-1239-2014, 2014.

2. Seroussi, H., Pelle, T., Lipscomb et al.: Evolution of theAntarctic Ice Sheet over the next three centuries from an ISMIP6 model ensemble, Earth's Future, 12, e2024EF004561, https://doi.org/10.1029/2024EF004561, 2024.

3. Klose, A. , Coulon, V., Pattyn, F., and Winkelmann, R.: The long-term sea-level commitment from Antarctica, The Cryosphere, 18, 4463–4492, https://doi.org/10.5194/tc-18-4463-2024, 2024.

---

## Referee Comment (RC2)

**General comments**

Capturing gravitational, deformational and rotational (GRD) effects of the solid Earth in response to ice-sheet evolution has been known to be important for ice-sheet model simulations for the paleo and future timescales. Of these, bedrock response (i.e., deformational effects) play the most dominant role on ice-sheet evolution. Bedrock deformation can be computed with different models of complexities and computational expense, and coupled ice sheet – 3D GIA models remain to be the state-of-the-art.

However, not many ice-sheet models have been coupled to bedrock deformation that incorporate 3D (radially and laterally varying) Earth structure. Many other models incorporate more simplified bedrock models, such as ELRA, LVELRA, or 1D (radially varying Earth structure) GIA models. Thus, it could be beneficial to provide suitable parameter values to those ice sheet models that incorporate simpler and cheaper bedrock models that would best resemble ice-sheet contribution to sea level projected by state-of-the-art coupled ice sheet – 3D GIA models.

In their manuscript "Approximating ice sheet – bedrock interaction in Antarctic ice sheet projections", van Calcar et al. attempts to provide suitable solid-Earth related parameters to the structurally different type of bedrock deformation models – ELRA, LVELRA, 1D GIA model. The authors simulate Antarctic Ice Sheet for 500 years into future with a coupled Antarctic Ice Sheet – 3D GIA model as a benchmark for the other three bedrock models, using two different climate forcing from two different climate models. ELRA models incorporate parameters such as flexural rigidity and a relaxation time and GIA models incorporate parameters such as lithosphere thickness and mantle viscosity, and the authors first empirically derive 2D maps of relaxation times informed by the 3D Earth structure. They then provide parameters for ELRA, LVELRA and 1D GIA models that ice-sheet modelers can use in their projection of Antarctic Ice Sheet for the time period.

This manuscript undoubtedly contains a great amount of work with creativity. It also has in mind a meaningful contribution to the field of future ice-sheet modeling – to provide constrained parameter values to the ice-sheet modelers who do not have access to the most sophisticated, state-of-the-art 3D GIA model. This contribution can also potentially be very useful for the next round of Ice Sheet Model Intercomparison community effort, ISMIP7. Therefore, I think the topic of the work deserves eventual publication in *The Cryosphere*. However, before the acceptance of this manuscript, I recommend major revision in text and analysis with regards to few points outlined below.

1. The main issue I have with the current form of the manuscript is the limited exploration and discussion regarding the sensitivity of the derived parameter values to different climate forcing from different climate models. It is known that even for the same high-emission scenario (e.g., RCP8.5 & SSP5-8.5), different climate models provide drastically different thermal forcing and surface mass balance, resulting in a wide range of uncertainties in Antarctic Ice Sheet mass loss and region of mass loss (e.g., Seroussi et al., 2024). To this argument, I suspect the reason why the authors derive ELRA relaxation time of 300 years for pre-2300 and 500 years for post-2300 has to do with change in the location of ice mass loss (West vs. East Antarctica). Thus, I am uncertain how applicable these results would be when ice-sheet models apply climate forcings from models other then CESM and IPSL used in this study. I'm not necessarily asking for a wide range of ensemble simulations, but there at least needs to be extensive

discussions around this topic, which might potentially require some more simulation/analysis addressing this point.

2. The authors create a bound of benchmark sea-level rise values based on the 1) difference between 3D-Average and 3D-weaker and 3D-stronger GIA model results and seems to then hand-wavily pick lines that sit between these bounds for the longer period. I am not sure if this is a good metric and a way of choosing recommended values. For instance, how is relaxation time 200 years in Fig. 4c not a good choice compared to relaxation time 300 in Fig. 4c when the difference between 3D-Average and relaxation time 200 years is as small as the difference between 3D-Average and 3D-weaker? A way around could be to provide a range of recommended values rather than a single value for each bedrock model for different time periods.

3. Section 3: How are the choice of four regions in West Antarctica made, and can this choice be well-justified? Given that ice in North Antarctic Peninsula and Ross Sea Embayment seem particularly insensitive to bedrock response, wouldn't it be more accurate to use just ASE or ASE and WSE to derive the 2D map for LVELRA where most grounded ice mass loss occurs in realistic simulations? Also, how is the 500 m-thickness for ideal loading chosen? Please provide some discussion around how your results would be sensitive to the choice of this ice thickness. Also how is East Antarctica treated where the idealized unloading test is not performed? Can you improve your 2D ELRA map in Victoria Land and Mertz basins in East Antarctica so the difference between the 3D GIA results and 2D ELRA improve?

4. The manuscript will benefit significantly from more extensive literature review, giving credits to those deserve, and being more appropriate/consistent in the citation scheme. Detailed comments are main in the following section.

5. The manuscript could benefit from more careful proofreading for the next round of submission.

I realize these comments may seem a lot to address, but I hope they are helpful in improving the manuscript and getting it published in the journal. I am wishing the best to the authors.

**Specific comments: addressing individual scientific questions/issues.**

1. L25-27: The authors attribute the differences across different bedrock models only to the spatial variations in viscosity. But some of the differences arise due to structural differences in theory in each bedrock model. This needs to be clarified in the motivation and discussion.

2. The "negative" or "stabilizing" feedback on ice mass loss has first been introduced in the context of "sea level", which includes both bedrock uplift and sea surface height drop. I would recommend the authors to introduce "sea-level feedback" as a general concept, and then explicitly say that they choose to omit the sea surface height component in their investigation; I would like to confirm this, by the way - is it actually true that you only feed the bedrock changes back onto the ice model in the case of using 1D and 3D GIA models, not sea-level change?. If so, you should refer to previous studies that explored the separate impact of deformational and gravitational effects on ice evolution to mention that deformational effects are the dominant portion of sea-level feedback. (e.g. Gomez et al. 2015, Han et al. 2021, Coulon et al. 2021).

3. Provide in detail how sea-level change is calculated. Does it correct for changes grounded ice thickness (or volume) due to bedrock deformation captured in different bedrock models?

4. Some statements are introduced as if they are general "facts" when they are not. For example, in Abstract Line 11, there is a sentence as follows: *"... accounting for the impact of bed deformation on ice dynamics can reduce predictions of future sea level rise by up to 40% in comparison with scenarios that assume a rigid Earth."* This is not an established fact, but rather depend on strength of climate forcing, sensitivity of the ice-sheet model to climate forcing and bedrock topography change. Therefore, instead of citing a number from a specific study, being general would be more helpful.

5. L14: *"Because modelling the response for a varying viscosity is complex, sea level projections often exclude the Earth's response,..."* This is a very convenient and even a naive way of explaining why ice-sheet models that project the future have been using fixed bedrock, let alone spatially varying viscosity. Until relatively recently, bedrock deformation (or GIA) used to be considered not very important for the "short", centennial time scale ice-sheet evolution, which future projections focus on. This very same comment applies to L49 as well.

6. L15: *… or apply a globally constant relaxation time or viscosity."* It may not seem clear to many people what the differences you mean here by using the words "constant relaxation time" or "viscosity". You can make this clearer by specifically referring to the use of simplified bedrock adjustment models such as ELRA or 1D GIA models.

7. L24: *"... that deviates less than 40cm from the average of the 3D GIA models… can be further reduced to 10 cm".* The specific numbers 40cm & 10cm do not mean much. It would be helpful to express the errors in a relative number, such as percentage.

8. L34: "*... requiring thousands of simulations to produce robust projections of potential sea level rise over the coming centuries.*" Seroussi et al. (2020) goes up until 2100 only. Seroussi et al. (2024) goes up until 2300. The ice model ensemble simulations in Seroussi et al. 2020 & 2024 are on the order of hundreds, not thousands. Maybe just say "ensemble simulations".

9. L34-35: I would recommend citing only the seminal papers that first showed stabilizing sea-level feedback (e.g., Gomez et al., 2010; 2012). Otherwise, you are missing references that also confirm the effects in addition to the ones being cited.

10. L47-48: *"However, it is currently unfeasible to include a realistic Earth structure in a large ensemble of sea level projections due to the long computation time involved (val Calcar et al., 2023)"*: This statement could be potentially out of context and misleading. The previous work the first author they cite here simulates the Antarctic Ice Sheet over the last glacial cycle, which is for hundred thousands of years. Here we are talking about the next few hundred years of projections, which would be feasible. Also, it could be appropriate to acknowledge other studies that showed ways to overcome computational infeasibility for long simulations, e.g., de Boer et al., 2014, Han et al., 2022, which are for 1D GIA models and thus could be debatable whether they are "realistic" models or not. Also, there is a study that emulates and thus provides large ensemble for 3D GIA model (Love et al., 2024)

11. L50-51: ELRA models are not only computationally cheap, but easier to implement because you don't need to "couple" a whole new model that is complex by itself. Also, I believe 1D GIA models are cheap enough to produce large ensembles. The claim that 1D GIA models are too expensive to produce ensembles for centennial time scale runs, needs evidence.

12. L79: The authors should cite the work of Jan Swierczek-Jereczek who developed the FastIsostasy model that captures lateral variations in lithosphere thickness and mantle viscosity. It is a critical reference that needs to be acknowledged.

13. L101: it is confusing to now refer ELRA and LVELRA to as "GIA models".

14. L153: what is the value of the lower mantle viscosity?

15. L174: What is Configuration 1 like? How specifically is it different from Configuration 2 rather than the former is "variable resolution". Variable resolution in what?

16. L179: "... procedure as described for configuration 1.". Nothing has been described for configuration 1. More detail would be helpful.

17. L182-183: So, both bedrock deformation and sea surface height changes are calculated. Are you then only passing bedrock deformation back to the ice model? Whichever way, this needs to be made clear.

18. L213: Mitrovica et al. 2011 shows laterally varying flexural rigidity has negligible effects on bedrock deformation?  Also, is the citation order following the year of references or the alphabetical order of authors' last names?

19. L225: How are the "large" and "small" areas chosen based on what metric?

20. L226: "*... such that the resulting empirical relationship between mantle viscosity and relaxation time accounts for the different mantle conditions…*". What does it mean by the relationship accounts for the "different mantle conditions"? Isn't the mantle condition the same but the surface load size is being varied? Clarify the sentence.

21. L228: Do you mean "each surface load" as in surface load on each grid cell or of each basin or each small/large area? I presume the latter but clarify. And what does it mean by "surface load is controlled"?

22. justification on why this thickness was chosen. Given that the marine-based portion of the West Antarctic Ice Sheet goes up to 2-km thick, and that most of the ice melting will happen in Amundsen Sea Embayment, 500m seems quite arbitrary and out of place. It would be also good to mention how sensitive the viscosity-relaxation time relationship would be to the choice of ice thickness.

23. L223-224: "This implies that the relaxation time depends on the size of the ice sheet and that a single relaxation time cannot be derived". Clarify this sentence, as the current version is confusing as to what the difference is between "the relaxation time" and "a single relaxation time". Something like, "a single relaxation time" that best represents those integrated over all modes."

24. L241: how is "the Earth" different from "the mantle" in this context? Does it make sense to say "the region of the mantle….depends on the sensitivity of the Earth…"? The current sentence sounds like the mantle is sensitive to its own viscosity, which is confusing.

25. L250&251: Explicitly mention that these equations are derived based on your fit shown in Fig. 2b. Also, I think it'd be more helpful to provide a general equation and then say you get different values for the average-viscosity case and the lower-bound viscosity case.

26. L255: So which method do you take? High-resolution viscosity profile or smoothing the 2D maps?

27. L232: Specifically, how many timesteps of how many years?

28. L224: Table 1 is missing from Supplemental Material.

29. L231: Provide more information for the 40 simulations. Within what range of the mentioned parameters are varied?

30. L265: Do you mean Supplemental Material not Extended Data? Also, Table 1 is missing.

31. L267: "... 3D-weaker and 3D-stronger Earth models described in section 2.1". Please provide more detailed description on these 3D models in Section 2.1.

32. L301-303: It seems like these recommendations would work well for the low-emission scenario, but for the high-emission scenarios these choices seem hand-wavy. For example, how is tau = 300 years any better than 350 in Fig. 4b? More importantly, there needs to be a discussion as to how these recommended values can change when applying climate forcings from different climate models. In a similar context, there needs to be a discussion as to why the recommended values change depending on the period (before 2300 vs. after 2300) in relation to the region of ice melting. It looks like the dominant mass loss happens in West Antarctica until 2300, so relaxation time that best represents bedrock response in West Antarctica will match better, and after 2300 ice in East Antarctica also goes away, and therefore higher relaxation time that reflects slower bedrock response, resembling East Antarctic response, will work better.

33. L314: It would be helpful to use a relative number (e.g., %) rather than absolute values. This comment applies throughout the manuscript.

34. Fig. 5 & Supp. Figs. 2,4,5: It would be helpful to see some comments regarding the strikingly similar grounding line contours across all GIA models. Also, it would be helpful to show these plots for year 2300 where there will be actual differences in both ice thickness and grounding line extent in West Antarctica.

35. It looks like most of the differences after 2300 across simulations come from Victoria Land and Mertz in East Antarctica. I'm curious why 2D ELRA models perform equally poorly compared to the 1D GIA and 1D ELRA models? This goes back to the question asked earlier on what kind of treatment is done for deriving relaxation time based on

viscosity in East Antarctica given the ideal unloading tests were performed only in West Antarctica.

**Technical comments:**

1. "Approximating ice sheet-bedrock interaction in Antarctic ice sheet projections". I think the title can be improved by being more specific.

2. Antarctic ice sheet => Antarctic Ice Sheet (throughout the manuscript)

3. L20: conducted => conduct. (or make the tense consistent throughout)

4. L23: sea level rise => sea-level rise

5. L56: "*A GIA model can include the bedrock response to changes in ice loading…*". Remove "can". GIA models by default should include bedrock response.

6. Define GIA earlier in the text. Right now, it's only introduced in L59 after the word has been used many times interchangeably with bedrock response.

7. L84&86: "GIA model using a 3D Earth structure" -> 3D GIA model

8. L103: "table 1" => Table 1

9. Table 1: explain the values and units used for the different Earth models. What maximum and minimum values do 3D-stronger and 3D-weaker models have? It would be helpful to describe those rather than referring readers to the other paper, van Calcar et al. (2024).

10. In Figure 1, you have a legend 1DASE, but in the table you have 1D18

11. L117: What does it mean by "realistic" sea-level projections?

12. L159: Earth structures => Earth structure

13. L169: Fig 1 => Fig. 1

14. Figure 1 legend: 1D Earth structures => 1D Earth structure profile?

15. L214: use consistent significant figures.

16. L217: I think it would be beneficial to have a more specific title for this section, something like, "Deriving 2D relaxation time maps from 3D viscosity profiles"

17. L223: "This implies that the relaxation time …..cannot be derived". This sentence feels redundant, as it should be very clear from the sentence before. But this is a minor comment.

18. L229: "The uniform thickness of each load is taken to be 500m". There needs to be

19. L285: ELRA model with uniform relaxation time?

20. L286-288: This sentence can be clarified. Strictly speaking, bedrock response is dependent on ice loading and viscosity. You could probably mention climate forcing then affects ice loading changes.

21. L290: This whole paragraph is quite confusing and don't seem necessary. It is clear the coupled ice sheet – 3D GIA model results differ from ELRA models (or 1D GIA models), and the difference in the results will vary with climate forcings (which are in turn different for different climate models), which is also clear.

22. L279: Unindent

23. L350: ELRA model with 2D laterally varying relaxation time?

24. Figure 6: Some lines go out of bound in the y-axis.

25. L380: The references cited here seem quite arbitrary and incorrect (Whitehouse et al., 2019 is a review paper, not an ice-sheet model.)

26. section => Section

27. configuration => Configuration

28. L408: accuracy in the metric of ice volume change.

29. L409: for two different emission scenarios from two different climate models.

**References**
1. deBoer et al., ice-sheet–sea-level model: algorithm and applications, GMD, 7, 2141–2156
2. Love et al., 2024., GMD. A fast surrogate model for 3D Earth glacial isostatic adjustment using Tensorflow (v2.8.0) artificial neural networks
3. Seroussi et al., 2024 ISMIP6 Antarctica 2300
4. Han et al., 2021. Modeling northern hemispheric ice sheet dynamics, sea level change and solid earth deformation through the last glacial cycle
5. Han et al., 2022. Capturing the interactions between ice sheets, sea level and the solid Earth on a range of timescales: a new "time window" algorithm
6. Jan Swierczek-Jereczek.,2024. GMD. FastIsostasy v1.0 – a regional, accelerated 2D glacial isostatic adjustment (GIA) model accounting for the lateral variability of the solid Earth
7. Gomez et al., 2015. Sea-level feedback lowers projections of future Antarctic Ice-Sheet mass loss
8. Gomez et al,. 2012. Evolution of a coupled marine ice sheet- sea level model
9. Gomez et al., 2010. Sea level as a stabilizing factor for marine-ice-sheet grounding lines.

---

## Author Comment (AC1)

**Introduction**

The authors made very clear that the target of the present article is to recommend:

1. a uniform relaxation time scale to use in ELRA,
2. a laterally-variable relaxation time scale to use in ELRA,
3. a radial viscosity profile to use in a 1D GIA model.

In a context where these GIA models are coupled to an ice-sheet model, these recommendations aim to reduce the discrepancies in sea level contribution from the Antarctic ice sheet compared to the use of a 3D GIA model (with 2 different plausible Earth structures, which is a well-appreciated detail of the work). This is specifically done within the frame of possible climate forcings until 2500, as computed by 2 GCMs (IPSL and CESM). The potential impact of the paper is therefore also very clear: offer a low-hanging fruit to ice-sheet modellers that want to improve their representation of GIA simply by adjusting the parameters that are used in their already-implemented GIA model. This effectively represents 1-10 seconds of work and is therefore very appealing. Additionally, the manuscript reads well and it is unambiguous that it required a lot of work, including many expensive simulations. After this praise, I unsurprisingly think that this is a very valuable endeavour and am generally in favour of seeing this article published. Nonetheless, I think that there are a series of important points that need to be addressed before this. I will now proceed to listing them with the following scheme:

- Primary comments: these are key points that need to be addressed in a very comprehensive and thorough manner before a possible publication.
- Secondary comments: these are points that are either curiosities of mine and/or easy improvements that can be made to the manuscript.
- Technical comments: these are related to phrasing, format, etc.
- Additional references that come up along the way are gathered at the end of the document.

I want to emphasise that the length of the upcoming points is motivated by the fact that I am eager to improve the quality of this article as much as I can, since it will likely be highly relied upon in future within the field of ice-sheet modelling. I will assume that the authors are familiar with common abbreviations from ice-sheet and GIA modelling.

We sincerely thank the reviewer for the thoughtful and constructive feedback. We greatly appreciate the time and effort invested in providing such detailed comments, and we are encouraged by the reviewer's recognition of the potential relevance of our study for the ice-sheet modelling community.

Disclaimer: I only read the editor's initial decision statement after writing down my comments.

**Primary comments**

1. The authors spend very little time explaining the details of the ice-sheet model that are of true importance for this work. For instance, mentioning the basal friction law (l. 110) is here of secondary importance compared to questions like:
   a. Is the potential of a marine ice-cliff instability included in the runs?
      We did not include the possibility of ice-cliff instability in the ice sheet model and will mention this in the manuscript.
   b. Does IMAU-ICE accept a heterogeneous, time-varying sea level? For that matter: is the coupling to the 3D GIA model merely deformational or does it also include the effects of rotation and gravitation?
      IMAU-ICE does include a heterogeneous, time-varying sea level, which is shown to be a much smaller effect than the effect of bedrock elevation changes in the preprint van Calcar et al. (2024). For the simulations presented in this manuscript, the sea level is assumed to equal the present day sea level as considered in Bedmachine version 3 (Morlighem et al., 2020), and constant in time. Furthermore, van Calcar et al. (2024) show that the effect of self-gravitation in the 3D GIA model is small whereas the computation time doubles by including this effect so this effect is also excluded in the simulations of this manuscript. This is mentioned in the introduction but we will move it to the method section for readability. Rotational feedback is not included. This will be mentioned in the method section.
   c. What type of melt parametrization is applied at the grounding line (Leguy et al. 2014)? Is any flux imposed? Using a resolution of 16 km is in my opinion sufficient to go ahead with the present publication but it needs to be accompanied by such information on the subgrid parameterization.
      We applied the flotation condition melt parameterization. The grounding-line behavior follows from the physics and numerics of the model without explicitly forcing a flux. We will mention this in the method section.
2. In the present work, SSP8.5 gives an SL contribution of 3.5-4.5 metres by 2300 (depending on the GCM and the 3D Earth structure). This makes IMAU-ICE more sensitive than any other model included in Seroussi et al. (2024, c.f. Fig. 4, especially panels a and b), which were forced, among others, by the 2300 projections of CESM which offers a straightforward comparison to the present work. In Seroussi et al. (2024), the ensemble of SL contributions under SSP8.5 spans -/+ 0.2 m of SL rise by 2100, which is a substantial difference to the value of >0.5m obtained in the present work. Adding to this, Coulon et al. (2024) and Klose et al. (2024) present SL contributions of the AIS that are nominally less than 3 m by 2300, although you are using the same climate forcing. This is somewhat surprising, because IMAU-ICE is the only one of these models that is coupled to a 3D GIA model, which has stabilising effects on the grounding line and should therefore lead to a lesser sensitivity. I think this needs to be mentioned and, in the best case, explained comprehensively. Answering the points I raise in 1.a.-1.c might be a step in this direction.
   We acknowledge that the sea-level contribution from IMAU-ICE under SSP8.5 is higher than reported in Coulon et al. (2024), and Klose et al. (2024), despite using

the same climate forcing. We agree that this is a noteworthy point. The higher sea-level contribution in our simulations likely stems from differences in key model components, particularly the initialization strategy, the basal melt scheme, and the melt parametrization at the grounding line. Varying parameters that control these aspects of the model lead to large variations in ice sheet evolution and corresponding sea-level rise. We emphasize that the goal of the present study is not to sample the full parameter space of IMAU-ICE or to conduct a full intercomparison of model sensitivities, but rather to isolate and quantify the relative impact of 3D Earth structure on Antarctic ice sheet evolution. That said, we agree that further investigation of how our model setup compares to others in the literature—including the role of subglacial and grounding-line parameterizations—is an important direction for future work, and we will make this point more clearly in the conclusion.

3. I believe that the aforementioned high sensitivity of IMAU-ICE might slightly undermine the message of the article: if the positive feedbacks that are internal to (marine) ice sheets are particularly strong in IMAU-ICE, the negative feedback of GIA might be underestimated in comparison. I believe that the simulations you present show a few symptoms of this:

   a. Looking at Fig. 4, the error with respect to the 3D runs is almost the same for ELRA3000 and ELRA200 - although ELRA3000 is far from representing the reality in WAIS (Barletta et al. 2018, among many others). In essence, this means that the difference to the average 3D model is relatively insensitive to the choice of $\tau$. Just so that you get me right: your optimal choice of $\tau$ would be the same with an ice-sheet model of lesser sensitivity and, in that sense, it does not change your final recommendation… but the impact of this optimal choice would be higher. I think you should consider this, since it might imply a larger reduction of the relative error in SL contribution thanks to your recommendations and could make your message stronger.

   We find the difference between ELRA3000 and the 3D model is significantly larger than that between ELRA200 and the average 3D model for most of the emission scenarios and climate forcings. However, we agree that with a smaller ice loss signal, the effect of GIA and hence the effect of difference in tau would be relatively larger, and it is a good suggestion to discuss this possibility as it indeed makes our recommendations more impactful. We will do so in the conclusions.

   b. The large uplift discrepancy of 250 m that you mention at l. 369 has almost no impact on the result. Explaining the little impact of this difference on SL contribution by invoking the fact that it arises relatively late in the simulation is a partially albeit not fully satisfactory explanation to me. I believe the (potentially too) high nonlinearities that are internal to the ice sheet model contribute to it.

   Indeed, it is not only the fact that the uplift arises relatively late, but also the fact that the ice sheet is in a state of collapse. Whether this is due to larger sensitivity (non-linearity) or and of the other differences with published estimates ("different

initialization method, basal melt approximation, and melt parametrization at the grounding line) is difficult to say, but we will extend the description as follows: "However, this uplift occurs mainly in the last 100 years. Furthermore, the effect on grounding line retreat is small because the grounding line is already retreating rapidly and the West Antarctic ice sheet in our simulation is in a phase of collapse. For another ice sheet simulation with different melt approximation or melt parametrization, the sensitivity to a similar uplift might be relatively larger."

4. This large discrepancy of 250 metres makes me wonder how the change of the ocean mask and the associated change in load is treated in your implementation of ELRA. When ice melts in zones grounded below sea level, it is replaced by the ocean which largely compensates for the reduction in surface load due to ice melt. Is this accounted for here? If not, this might be an explanation for such high error values of ELRA – which arise even for a 2D relaxation time field! This point illustrates a deficiency of the article, which does not mention at all what is the coupling scheme between IMAU-ICE and the various GIA models. This can however be easily fixed with a couple of additional lines in the manuscript.

The replacement of ice grounded below sea level by the ocean is accounted for in both the implementation of ELRA and the GIA models. This is discussed in van Calcar et al. (2024) but we will add additional lines in the manuscript to make the clear here as suggested.

5. I know very little about IPSL but CESM is known as a quite high-sensitivity model. I think this is a rather good thing since it leads to ice-sheet configurations that show large differences to present-day, thus exploring more diverse AIS configurations. I think this could be highlighted in a discussion section, especially since it supports the overall message of the paper. I would very much appreciate more information about the GCM outputs in the supplemental material: e.g. their present-day bias compared to RACMO, MAR or ERA5, and their warming pattern (atmospheric and oceanic) by 2300.

We will add more information the manuscript about the GCMs. What is most relevant for the Antarctic simulations is the ocean warming. The average ocean temperature anomaly is colder for CESM than IPSL in the high emission scenario, but warmer in the low emission scenario (Extended Data Fig. 7 in van Calcar et al., 2024). In the supplemental materials of this manuscript, we will include a figure showing the average ocean temperature anomaly by 2300.

6. In Fig. 4, IPSL gives a barystatic sea level contribution of 3 to 4 metres by 2500 under SSP2.6, which roughly corresponds to the total sea-level equivalent volume of the WAIS (i.e. only taking into account ice above floatation + corrections). In the corresponding maps shown in Fig. 5, the WAIS is however far from a total collapse, since the Ross and Filchner-Ronne grounding lines are essentially unaffected compared to present-day. I suspect two possible explanations for this discrepancy:
   a. The ice sheet has experienced a thinning throughout its domain, including EAIS (otherwise such values are impossible).

b. The authors are showing the ice volume that was lost (including ice grounded below sea level) and NOT the actual contribution to barystatic sea-level. If this is the case, this might of course partly resolve the doubts I raise in 2. and 3.

The ice volume that we show does not include ice grounded below sea level. The explanation corresponds to option a: The 3 to 4 meters sea level rise not only includes a large retreat of the Amundsen Sea Embayment, but also significant retreat in the Wilkes basin and some retreat of the Ross Ice Shelf. Furthermore, there is thinning of the WAIS. This is shown in Fig. 5 in the manuscript and will be discussed in the results.

7. Questions around the spatial resolution of models can easily become generic. I hope this does not apply to the one I will try to raise here. The "high-resolution" configuration of your 3D GIA model has a lateral resolution of 200 km, which is refined to 35 km in the WAIS. I believe this is more than sufficient if one wants to study mechanisms in a qualitative way, e.g. "how does rheology affect the grounding line retreat?". However, the goal of the present study is quite different, namely to make a recommendation of parameter choice to minimise the error of simpler GIA models compared to the averaged 3D results. This is a much more quantitative assessment, which essentially places your 3D GIA model as a "golden standard". The recent work of Gomez et al. (2024), which uses a resolution of 3 km for the GIA model and 5 km for the WAIS (somewhat coarser outside), places a quite high target to what can be called a "golden standard" nowadays. I am not arguing that such a resolution is needed for the present study. However, if you want to keep the resolution as it is you need to show a convergence of some meaningful metric over (lower) resolutions. Alternatively, I'd be happy with a 16km-100km combination for the lateral resolution of your 3D GIA model or with any argument that shows that your resolution is high enough. I know this type of comment is the authors' nightmare because it implies repeating runs that can become substantially longer. However, I hope that the authors understand that this expectation comes from the potential high impact of the paper.

We acknowledge that a higher resolution of the GIA model can have an impact and we appreciate the thoughtful comment. We performed a sensitivity test using the GIA model loaded with a parabolic ice cap using a resolution of 70, 55, 30 and 15 kilometers, and found that using a horizontal resolution of 15 km by 15 km instead of 30 km by 30 km decreases the deformation by 0.01% (2 cm) over 1000 years and increases the computation time of the GIA FE model by approximately 30% (Fig. S3 in the supplemental material of van Calcar et al., 2023). The reason for the relatively low sensitivity is that we think the high resolution is much more important for elastic effects that have a smaller spatial wavelength. We will mention the conclusion of this test in the current manuscript.

Although the effect of different resolution might be somewhat larger on shorter timescales, we think it is smaller than the uncertainty in the adopted mantle viscosity structure (Wan et al., 2022, van Calcar et al., 2023). Therefore the "gold standard" we use, has in fact a large uncertainty due to the unknown mantle viscosity and a smaller uncertainty due to resolution. Therefore we think it is justified to stress the former by

exploring the sensitivity to different 3D Earth structures. . We will explain this in the manuscript.

8. Along the same idea, l. 149-150: "the GIA model is used with 9 vertical layers (0-35 km, 35-100 km, 100-150 km, 150-300 km, 300-420 km, 420-550 km, 550-670 km, 670-1171 km, and 1171-2890 km, and 2890-6371 km)." It seems like you used the same set-up in van Calcar et al. (2023) – which is fair in that case since you are not making direct parameter recommendations. Unless I am getting something wrong, this sounds very coarse to me, for instance compared with what was used already some time ago in Gomez et al. (2018), where the vertical layers have a thickness of 6–50 km (only down to the CMB). Another example with a different model can be found in Albrecht et al. (2024), where "$\Delta z$ = 5 km down to 420 km depth, followed below by $\Delta z$ = 10 km down to 670 km and $\Delta z$ = 40 to 60 km down to the core–mantle boundary": The use of a much coarser resolution here presents two problems:

   a. A potentially inaccurate representation of the mechanics. For instance, even for an idealised constant viscosity over depth, you might obtain significant errors compared to the actual solution. In particular, does a layer from 2890 down to 6371 km depth make any sense in that context?

   b. An inaccurate representation of the parameter field. If you have fast variations of viscosities over depths, as it is the case in ASE, averaging over such thick layers might lead to an additional error when resolving the mechanics.

   Just as for the lateral resolution, I wish to see at least some convergence/refinement arguments to fully approve a paper that recommends parameters to the community.

   The thickness of the layers mentioned by the reviewer holds for the Earth model parameters assigned to the elements within those layers, and not to the mesh and resolution of the FE model. All the elements in the model are approximately square and thus in the Antarctic region they are around 30 x 30 x 30 km for the lithosphere down to the lower mantle. The sensitivity test mentioned in comment 7 is performed on this grid. We will clarify this in the text. The thickness of the elements is larger than in other studies. However, as noted before, this is much more important for the elastic response than for the viscous response

9. The explored range of ice-sheet configurations is essentially limited to a collapse of the WAIS (and parts of EAIS basins?) compared to present-day. If a different configuration of the ice sheet is being studied, the results shown here might be leading to large errors. For instance, performing the same task with simulations in the distant future would lead to a higher optimal uniform relaxation time (or 1D viscosity profile) since the bulk of contribution to sea level would come from EAIS, where the viscosities are even higher. This is illustrated in many of your plots, where it is clear that the error has a pronounced drift beyond 2500, giving an advantage to slower responding models. This does not undermine your message at all but needs to be discussed.
   We agree and will discuss the case of EAIS retreat in the results. Thank you for this suggestion

10. Swierczek-Jereczek et al. (2024) present FastIsostasy, a regional GIA model that (1) is computationally as efficient as ELRA, (2) directly takes laterally-variable viscosity and lithospheric thickness as input, (3) displays different relaxation time depending on the wavelength of the load, (4) includes a regional approximation of gravitational feedbacks on sea level and (5) displays errors that are low compared to a 3D GIA model, which was tested with idealised as well as realistic ice histories. Citing this paper was omitted, in spite of its direct relevance for the aim of the present paper. It however seems difficult to circumvent because it changes the emphasis of the message that is conveyed in the present article:

    a. Computationally efficient ways to represent lateral variability with low error compared to a 3D GIA model already exist which contrasts with the statement at l. 45-47. Besides FastIsostasy, 3D regional models like those presented in Weederstijn et al. (2023) and Nield et al. (2018) are computationally tractable when considering simulations only until 2500. The main advantage of the present work is to avoid coupling to a different GIA model, which can sometimes require quite a lot of programming effort. I insist that this is the main contribution of the article, which is very useful but should be stated more precisely.

We agree that the statement about existing models with lateral variability needs more nuance and context. However, the applicability of the models cited by the reviewer for our purpose is not straight-forward: (i) for ensemble studies the computation time can still be limiting and (ii) because of assumptions in these models they have inaccuracies and in terms of accuracies could resemble more the ELRA with 2D relation time that is presented in our paper, or the 3D model. More study would be required to establish that.

For example  FastIsostasy has great potential to be applied in ice sheet models to conduct sea level projections as it includes more realistic features compared to an ELRA model, as mentioned by the reviewer in his comments below (varying lithosphere, response time dependence on ice sheet geometry) but its performance for a prescribed ice history over a glacial cycle shows significant differences with the 3D GIA model that it's compared to. Furthermore,  Weerdesteijn et al. (2023) and Nield et al. (2018) present regional models which, if coupled with an ice sheet model would have relatively large errors for Antarctic wide ice sheet models. We will describe efforts to overcome computational infeasibility that can be used to compute bedrock deformation coupled to an ice sheet model for projections, by Han et al. (2022) and Swierczek-Jereczek et al. (2024), and regional models, and we will include the possible limitations of such models  in the manuscript to explain the statement made in lines 45-47. If provided with a new seismic data set, the process of fitting the relaxation time scales (summarised in Fig. 2 and in Supplemental Tab. 1) might need to be repeated. You are in the best position to know how time consuming this is and the ice-sheet community will be grateful that you made this effort. However, it should be emphasised that repeating this effort can be avoided by using approaches like FastIsostasy.

Indeed, our idea is that the fitting is done with a large enough sample of 3D viscosities that it is robust. Therefore, when provided with a new seismic data set, the fit we provided could directly be used to compute a new relaxation time map for the new seismic assuming it is converted to viscosity in a straightforward way. The possibility of using FastIsostasy will be described as mentioned in the reply to the previous comments. It will be interesting to study how FastIsostasy would perform in terms of accuracy and computation time.

b. You mention that the impact of a laterally variable lithospheric thickness has only a marginal effect on the bedrock deformation (l. 212-213). I think this depends on the ice-sheet configurations that are explored, especially for future projections where the West-East gradient might become more relevant. Of course, including a laterally variable LT is more complex to implement… but it turns out it was already done in FastIsostasy without noticeable increase in the computational cost. Since the software is open source (in fortran and julia), the only practical obstacle to including this is the coupling effort. I think it should be discussed that the latter is not more tedious than exploring different combinations of 2D relaxation times and constant lithospheric thicknesses (e.g. your work summarised in Fig. 6).

Our premise is to present a suggestion for in implementation that ice sheet modellers could implement with ELRA that is contained in several models. As such, ice sheet modelers don't need to explore combinations of 2D relaxation times and an lithospheric thickness themselves as this manuscript presents the optimal choice for tau that results in the closest match to a model that is more realistic in several aspects, including the effect of lithosphere thickness variations. We think laterally varying flexural rigidity, which has little effect on deformation (Coulon et al., 2021), does not justify the extra complexity. With FastIsostasy a varying lithosphere thickness could also be explored and we will mention this possibility in the manuscript.

c. As you mention at l. 223-224, ELRA displays exactly the same relaxation time, regardless of the extent of the mass anomaly. Because of this, it would be particularly important to discuss the geometry of the schematic load that you apply to the 3D model to derive the relaxation time. What horizontal extent does the load have? Did you vary this to see how it impacted the final result? From Tab. 1 in the Supplement I gather you did that (small, large and LGM), but this is not mentioned explicitly although it makes the paper stronger! As a side note, FastIsostasy partially fixes this since it does not present a flat relaxation spectrum. The geometry of the surface loads, and the different sizes used, are discussed in lines 225-229. The horizontal extent is shown in Fig. 2a and is indeed varied in size. We add a discussion on the effect of horizontal effect on the derived relaxation time as it can be deduced from our results The fact that FastIsostasy does simulate a response time that depends on the geometry of the load will be mentioned in the discussed of FastIsostasy as mentioned before. We note that also ELRA could be modified to include a relationship between the wavelength of the load and the relation time.

11. You recommend a radial viscosity profile for the 1D GIA based on the behaviour of the AIS compared to the outcome with a 3D GIA model. The caveat of this is that this lower viscosity is then applied globally whereas we know that most regions have higher viscosities. In other words, your error metric is local, although your model is global... Imagine we select a random point at the coast of the Netherlands: the sea level rises because of AIS melting and floods the grid cell which subsides because of the new load. This response happens too fast with your viscosity recommendation! Again, this is not critical but needs to be discussed at least a bit. We agree with the argumentation that the lower viscosity would not work well to simulate ocean loading and will include this in the manuscript.

12. The points I raised so far make me think that a discussion section between the results and the conclusions is absolutely necessary!
Since we have many results of different types of simulations (ELRA, LVELRA, and 1D), it becomes confusing to first show the results of all these simulations, but to place them in context much later in the text in a separate discussion section. To keep overview, we discuss the interpretation of the results directly after showing the results for each type of simulations. We will extend the discussion, please see our response to the previous comments on where we will add to the discussion.

13. I acknowledge the intention of the authors to provide all the data that resulted from the study in the supplemental material. This is mostly raw, ~100Gb data, which has a value if anyone wants to reproduce the analysis, but which is also anything but visual and hence of little use to the average reader. I encourage the authors to provide supplemental material that is of importance for the present study while being directly digestible, such as maps of the initial state of the ice sheet with an error plot with respect to to present-day, as well as maps of the ice thickness anomaly at 2500 compared to present-day for SSP2.6 and SSP8.5 (for at least one of the 3D runs). For instance, the doubt I raise in 5. could be answered directly by such a plot.
The suggested maps are published in the preprint van Calcar et al. (2024). We will refer to it explicitly in the method section of this manuscript.

**Secondary comments**

1. The word "bedrock" is used throughout the manuscript with a very "GIA perspective" on things. Bedrock response to ice-sheet forcing includes, for instance, erosion and sedimentation - which is obviously not your current object of study. I suggest you change this to something less ambiguous throughout the manuscript like "bedrock (vertical) deformation". In particular I think this would make the title more specific. Something like "Approximating the coupled bedrock displacement in Antarctic ice sheet projections". Since the main result of your work is the parameter recommendation, you could even include this idea in the title.

We will change the title to: "Approximating bedrock deformation of a 3D Earth in an ice sheet model for Antarctic sea-level projections". We will also change "bedrock response" to "bedrock deformation" throughout the manuscript.

2. L. 23-24: You mention a difference in SL rise that "deviates less than 40cm". Without knowing what is the total contribution to SL rise, this number is not very meaningful. In particular, it might sound like a huge amount (which it is in terms of policy advising!) if not cited as relative value. We will include relative values in the manuscript.
3. L. 29: Citing the IPCC is of course pretty reliable but it is not the most up-to-date information. I believe you should at least add Seroussi et al. (2024), Coulon et al. (2024) and Klose et al. (2024) here. We will include those references.
4. L. 30 "atmospheric and oceanic feedbacks". I would replace "feedbacks" by "processes" since it is more general. For instance, the warm water intrusion observed over the last decades in ASE is an oceanic process, not a feedback. We will adjust this.
5. L. 43-45: the context of the numbers you are mentioning could be more precise. Over the next centuries = until 2500? We will adjust this.
6. L. 46: I believe you can at least partially do that with the model presented in Swierczek-Jereczek et al. (2024). Please see our response to the comment 10 above.
7. L. 61: You are omitting Gomez et al. (2024) which contrasts with your statement. The sentence refer to ice sheet models including a 1D GIA model, in context to the sentence before. We will change the sentence to: Current existing ice sheet projections including a 1D GIA model use a homogeneous…
8. L. 82 - 88: you repeatedly use "approximate the ice sheet evolution resulting from a coupled ice sheet - GIA model using a 3D Earth structure" in the three questions. I think it would read better if you define this before as your "baseline" and refer to it as such later. I believe the three questions you pose are erroneously formulated. This appears particularly in your conclusions: "Research question 1 was: How well can a uniform relaxation time approximate the ice sheet evolution resulting from 3D Earth structures? We recommend to use a uniform relaxation time of 300 years with a lithospheric thickness of 100 km to replicate the sea level rise predicted by a model that includes 3D Earth structure." The answer and the question don't match! You ask "how well?" and you answer "the best choice is". Your question is actually: what is the best parameter choice for the near future? Of course, this comes with an error analysis ("how well?"), but the final outcome you want to convey to the community is the improved choice of relaxation time for ELRA and viscosity profile for 1D GIA models.
We will adjust the research questions as suggested to match the answer given in the conclusion.
9. L. 82 and following: do you have any reference that describes the GRD effects as implemented in your 3D GIA model? I feel like the description of dislocation and diffusion creep parameterization is actually less important than this – although admittedly much more concise to write down.
We will clarify that the model including the GRD aspects is described in van der Wal et al., 2013; van der Wal et al., 2015; Blank et al., 2021, van Calcar et al., 2023; van Calcar

et al., 2024 but we will add a brief extra description following reviewer comments about the sea level equation and rotation.

10. L. 90: you introduce IMAU-ICE but previous work of yours mentions ANICE and the code provided in the supplemental material also uses this name. Is there any difference between both apart from the name? IMAU-ICE and ANICE are two different ice sheet models. ANICE is used in Van Calcar et al. (2023), and IMAU-ICE is used in van Calcar et al. (2024) and this manuscript. We cannot find a mention of ANICE in the supplemental materials, neither in any of the data behind the provided doi's. We did include the code of IMAU-ICE called IMAU-ICE.zip in https://data.4tu.nl/datasets/b5548aaa-4c05-45f7-b0ce-775b83f13e5d/1 and provided the github link of IMAU-ICE.

11. L. 95: "and one based on a constraint in the Weddell Sea Embayment and Palmer land in the Antarctic Peninsula." I don't really understand why you use the Weddell Sea Embayment and Palmer land as a constraint, since they are not particularly relevant for future sea level rise. Because we have better GNSS constraints in this region? The ice sheet retreat is actually mainly occurring from the Ronne ice shelf when the climate model CESM is used, making this region particularly relevant for ice sheet dynamics in projections. The fact that this is not the case for all published scenarios makes clear that the uncertainty in where ice loss takes place is larger than expected. Furthermore, there are very few constraints on mantle viscosity in Antarctica and indeed, we tune the grainsize and water content to where viscosity could be constrained based on the limited number of GNSS studies. We will include this in the method section and refer to van Calcar et al. (2024) where this is discussed in detail.

12. L. 145-147: do you have a reference where it is shown that both things are equivalent? As an analogy from ice-sheet modelling to support my question: tuning the friction law or tuning the friction coefficient field can have similar impacts but are actually quite different conceptually and may lead to different sensitivities. Indeed the background stress generally lowers viscosity which has similar impact as for example decreasing grain size. We will refer to Blank et al (2021) who included several simulations with background stress.

13. L. 182: I understand this means that the rotational feedback and the shoreline migration are not taken into account? This should be stated explicitly. Rotational feedback was not included. The change in grounded ice cover (or alternatively labelled 'meltwater influx') was taken into account. The time dependent coastlines caused solely by changing sea levels is a much smaller effect (Milne et al., 1999) and was not taken into account. We add this in the method section.

14. L. 186: ELRA does not "depend" on a point load. The displacement is just obtained by a convolution of the response to a point load with the actual load. We will adjust this.

15. 15.L. 203: I suspect using a uniform 60 or 100 km lithospheric thickness for any experiment that implies a large-scale retreat in East Antarctica would give a pretty erroneous displacement pattern, since the lithosphere is much thicker in this region. This should be mentioned at some point in the discussion to prevent ice-sheet modellers from using it in such cases. Coulon et al. (2021) explored a large range of flexural rigidity in projections using the ELRA model including large ice mass loss in the Aurora and

Wilkens basins in East Antartica. They found little sensitivity to lithospheric thickness and a strong sensitivity to the relaxation time. We will extend the discussion on the flexural rigidity in lines 212-126 in the manuscript.

16. L. 301 and following: I feel like recommending a parameter value until 2400 and a different one if running until 2500 is a very odd thing to do – especially because it depends on your ice sheet model + climate forcing and is therefore not particularly robust. I think recommending a single nominal value for 2500 accompanied by an uncertainty range is a much more useful information for ice-sheet modellers, who can then choose to sample the uncertainties or not. This would be nicely illustrated if you provide some integrated error metric (as box or scatter plot) on the right y-axis of Figure 4 b, c, e, f, Figure 6 and Figure 7. We agree that providing a range is more convenient for ice sheet modelers and will include in the text that a range of 275 to 375 years for the relaxation time is best suitable amongst both scenarios and climate models, and recommend 300 years as the optimal choice. Additionally, we will present the error metric.

**Technical comments**

1. Some of your section titles have points after the number e.g. "5. Conclusions and outlook" and some not e.g. "4 Projections using different approaches to bedrock response". Also their indentation is not consistent. We will fix this.

2. The table and figure captions are all in bold fonts, which I believe is a formatting error. We precisely followed the Copernicus template.

3. I think that throughout the Copernicus journals, abbreviated names for figures, tables and equations should be used when placed in the middle of a sentence but they should be spelled out if at the beginning or at the end of a sentence. If I am not mistaken in this regard, you should correct l. 188, l. 234, l. 240, l. 297 (maybe I am missing some). We will adjust this.

4. L. 19: I reckon that when spelling out ELRA a comma, a dash or a slash should separate "elastic lithosphere" from "relaxed asthenosphere". Apply also later in the manuscript. We will follow Le Meur and Huybrechts (1996) and include a comma.

5. I believe "ELRA" can be written without specifying "the ELRA model", since models are generally referred to by their sole name once introduced (e.g. "CESM shows high equilibrium climate sensitivity" and not "the CESM model shows…"). We will adjust this.

6. L. 26: you use "AIS" without introducing the acronym… and then you inconsistently use "Antarctic ice sheet" and "AIS" throughout the manuscript. We will adjust this.

7. L. 32: giving a numeric value for the uncertainties arising from comparison projects could support your argument very nicely here (e.g. Seroussi et al., 2024, Fig. 15). We will include this.

8. L. 42: "specific region where the loss occurs" is a bit bit vague. I think it would be clearer if replaced by "the solid Earth properties of the region where the ice loss occurs" We will adjust this.

9. L. 52: I would add some reference to the statement – at the very least LeMeur and Huybrechts (1996). We will add the references.

10.L. 56: I would delete "(the elastic modulus)", since "stiffness" is clear enough. We will adjust this.

11. L. 69: "However, using…" should be reformulated to make clear that it refers to the work presented in Pollard et al. (2017). We will adjust this.

12.L. 103-105: this is a bit tedious to read. I suggest the following rephrasing: "We use the projections of two climate models, CESM and IPSL, under a low emission scenario (SSP1.2-6) and a high emission scenario (SSP5.8.5). In CESM, the warming mainly occurs in the Weddell sea, whereas in IPSL, the warming mainly occurs in the Amundsen Sea." We will adjust this.

13.L. 108: I would replace "ice dynamical" by "thermomechanically coupled ice model", which is more accurate. We will adjust this.

14.L. 110: I guess that you use the nonlocal formulation of the melt law proposed in Favier et al. (2019)? I think this should be written down explicitly to not confuse it with the local formulation. We use the local formulation and will include it in the text.

15.Equation 1 and 2 (and rest of the manuscript): it is unclear to me which logic is adopted by the authors when it comes to italic vs. normal font for mathematical symbols and indices. For instance incoherent notation of D and L between l. 187-188 r and Equation 3 and 4. Also Equation 6 and following text… We will make it consistent.

16.Equation 2: I think $T(x, y)$ makes much more sense than the subscript notation you use, since the temperature is a function of the space coordinates. Mentioning what $T(x,y)$ is is redundant between l. 133 and l. 136 (but as you wish… sometimes nice to repeat). We will adjust it to $T(x,y)$.

17.L. 131-134: you give numerical values for some parameters and for others not. Why not for all? We will include all parameter values.

18.L. 169: this very short sentence could be fused with the former one. We will adjust this.

19.L. 174: you introduce the two configurations. One is introduced with quotation marks and the other without. This should be coherent. Moreover, writing out "low resolution configuration" and "high-resolution configuration" is almost just as long as "configuration 1/2" and it has the advantage that the reader does not need to remember which is which. We will adjust this.

20.Figure 1: pa → Pa in the x label We will adjust this.

21. Equation 5: The subscript notation "w ", "b " makes more sense than using i,j i,j parentheses here since it refers to matrix indices, as stated in the final part of the sentence. As suggested, we will replace equation 5 by the following:

$$\frac{db_{ij}}{dt} = \frac{w_{i,j} - b_{i,j}}{\tau}.$$

22. L. 231-232: I don't get what the sentence "For each simulation, the displacement over time for each surface load/Earth model combination is computed for multiple time steps" should convey. That you don't have a single time step as output? Isn't that a bit obvious? Yes, we will remove it.

23. Figure 2 is great. You probably can leave out the distracting axis ticks and labels of the Antarctic map, since it is clear what is represented but I leave that to your choice. We understand that the figure is more attractive without the axis ticks and labels but the ticks allow to interpret the spatial scale of the different regions so we prefer to keep it.

24. L. 237: I believe "log of the viscosity" should be replaced by "logarithm of the viscosity" or "logarithmic viscosity". We will adjust this.

25. L. 268-270: this sentence reads pretty bad. Why invoke the ice sheet model here if the only thing you say is that it evolves the ice thickness in time? We will change this to: For the ice thickness changes over a timescale of centuries, the highest sensitivity will be in this relatively shallow layer (Barletta et al., 2018).

26. L. 280-284: the two sentences begin with exactly the same structure, which reads a bit odd. We will adjust this.

27. L. 284: a small technicality but... isn't the barystatic contribution to sea level actually calculated within your GIA model based on the ice thickness? No, we computed the sea level contribution using ice thickness and bedrock deformation in the ice sheet model.

28. L. 295: 0.44-07 m does not correspond to the percentage of 8-20%. Also, I would suggest giving the error separately for each scenario since it changes the value quite significantly. We will adjust this.

29. I would suggest giving references, at some point, about IPSL and CESM and potentially about any publication describing the simulations you are using. I reckon the refs can be found in the Coulon and Klose (et al., 2024). We will provide the references.

30. L. 301-304: you begin with the conclusion on your results. I would first begin with the detailed analysis you make at l. 307-330 and then conclude (what you actually do partly already and hence 301-304 is redundant). We will adjust this.

31. L. 304: I could not find Supplemental Fig. 1. Supplemental Fig. 1 is included in the supplemental materials, which is published along with the preprint.

32.Figure 4: "km·m2 /s2" is a weird unit to use when you can simply use "N m" as done before in the manuscript. We will adjust this.

33.L. 366: the IPSL projections are not ASE projections. They are just projections, with more warming in ASE. I think this is not formulated clearly here. We will adjust this.

34.L. 307, 368: I would just write "IPSL-SSP2.6" and "IPSL-SSP8.5" instead of lengthy formulations like "In the high emission scenario driven by the IPSL climate model" (which is wrong anyway because it is the high emission scenario that is used as input for IPSL). We will adjust this.

35.L. 408: this sentence reads pretty bad. How about "Using forcing fields from 2 different GCMs under low and high emission scenarios, we investigated the accuracy of common implementations of bedrock displacement when coupled to an ice sheet model." Or something similar… We will adjust this.

36.L. 432: 2D stronger 120 looks more like 20 cm error by 2500 as far as I can tell from Fig. 6, and not 10 cm as you state. The error is maximum 10 cm (brown dashed lines).

37.L. 454 and following: I wonder if "doi" should be replaced with "DOI" since it's an acronym. We will adjust this.

I hope these comments will make the paper stronger and I wish the best of luck to the authors for the rest of the publication process.

We would like to thank you for the thorough review.

Sincerely,

Jan Swierczek-Jereczek

**Additional references**

1. Leguy, G. R., Asay-Davis, X. S., and Lipscomb, W. H.: Parameterization of basal friction near grounding lines in a one-dimensional ice sheet model, The Cryosphere, 8, 1239–1259, https://doi.org/10.5194/tc-8-1239-2014, 2014.

2. Seroussi, H., Pelle, T., Lipscomb et al.: Evolution of theAntarctic Ice Sheet over the next three centuries from an ISMIP6 model ensemble, Earth's Future, 12, e2024EF004561, https://doi.org/10.1029/2024EF004561, 2024.

3. Klose, A. , Coulon, V., Pattyn, F., and Winkelmann, R.: The long-term sea-level commitment from Antarctica, The Cryosphere, 18, 4463–4492, https://doi.org/10.5194/tc-18-4463-2024, 2024.

Additional references

Glenn A. Milne, Jerry X. Mitrovica, James L. Davis, Near-field hydro-isostasy: the implementation of a revised sea-level equation, Geophysical Journal International, Volume 139, Issue 2, November 1999, Pages 464–482, https://doi.org/10.1046/j.1365-246x.1999.00971.x

Wan, J. X. W., Gomez, N., Latychev, K., and Han, H. K.: Resolving glacial isostatic adjustment (GIA) in response to modern and future ice loss at marine grounding lines in West Antarctica, The Cryosphere, 16, 2203–2223, https://doi.org/10.5194/tc-16-2203-2022, 2022.

---

## Author Comment (AC2)

General comments

Capturing gravitational, deformational and rotational (GRD) effects of the solid Earth in response to ice-sheet evolution has been known to be important for ice-sheet model simulations for the paleo and future timescales. Of these, bedrock response (i.e., deformational effects) play the most dominant role on ice-sheet evolution. Bedrock deformation can be computed with different models of complexities and computational expense, and coupled ice sheet – 3D GIA models remain to be the state-of-the-art.

However, not many ice-sheet models have been coupled to bedrock deformation that incorporate 3D (radially and laterally varying) Earth structure. Many other models incorporate more simplified bedrock models, such as ELRA, LVELRA, or 1D (radially varying Earth structure) GIA models. Thus, it could be beneficial to provide suitable parameter values to those ice sheet models that incorporate simpler and cheaper bedrock models that would best resemble ice-sheet contribution to sea level projected by state-of-the-art coupled ice sheet – 3D GIA models.

In their manuscript "Approximating ice sheet – bedrock interaction in Antarctic ice sheet projections", van Calcar et al. attempts to provide suitable solid-Earth related parameters to the structurally different type of bedrock deformation models – ELRA, LVELRA, 1D GIA model. The authors simulate Antarctic Ice Sheet for 500 years into future with a coupled Antarctic Ice Sheet – 3D GIA model as a benchmark for the other three bedrock models, using two different climate forcing from two different climate models. ELRA models incorporate parameters such as flexural rigidity and a relaxation time and GIA models incorporate parameters such as lithosphere thickness and mantle viscosity, and the authors first empirically derive 2D maps of relaxation times informed by the 3D Earth structure. They then provide parameters for ELRA, LVELRA and 1D GIA models that ice-sheet modelers can use in their projection of Antarctic Ice Sheet for the time period.

This manuscript undoubtedly contains a great amount of work with creativity. It also has in mind a meaningful contribution to the field of future ice-sheet modeling – to provide constrained parameter values to the ice-sheet modelers who do not have access to the most sophisticated, state-of-the-art 3D GIA model. This contribution can also potentially be very useful for the next round of Ice Sheet Model Intercomparison community effort, ISMIP7. Therefore, I think the topic of the work deserves eventual publication in The Cryosphere. However, before the acceptance of this manuscript, I recommend major revision in text and analysis with regards to few points outlined below.

We would like the thank the reviewer for the valuable feedback and the recognition of the importance of our manuscript to the field of ice-sheet modeling.

1. The main issue I have with the current form of the manuscript is the limited exploration and discussion regarding the sensitivity of the derived parameter values to different climate forcing from different climate models. It is known that even for the same highemission scenario (e.g., RCP8.5 & SSP5-8.5), different climate models provide drastically different thermal forcing and surface mass balance, resulting in a wide range of uncertainties in Antarctic Ice Sheet mass loss and region of mass loss (e.g., Seroussi et al., 2024). To this argument, I suspect the reason why the authors derive ELRA relaxation time of 300 years for pre-2300 and 500 years for post-2300 has to do with change in the location of ice mass loss (West vs. East Antarctica). Thus, I am uncertain how applicable these results would be when ice-sheet models apply climate forcings from models other then CESM and IPSL used in this study. I'm not necessarily asking for a wide range of ensemble simulations, but there at least needs to be extensive discussions around this topic, which might potentially require some more simulation/analysis addressing this point.

We agree that there are more climate models available and used in the study by Serrousi et al. 2024, but our goal is to focus on a variety of GIA related aspects in the context of climate change. For this reason we have used two climate models which are widely varying in there patterns. They include warming in key regions such as the Amundsen Sea, the Ross Sea, and the Weddell Sea, covering the primary areas of ice loss, while differing significantly in warming intensity. Thus, our choice for these two models ensures that we capture different forcing magnitudes, spatial warming patterns, and long-term projections. We acknowledge that different climate models provide significantly different thermal forcings in terms of both magnitude and spatial distribution. which might not be captured by these two models. Therefore, we will extend the discussion of possible effects of unknown climate forcing to highlight this issue, but again repeat that our aim is not to come to the best possible projection, but to focus on the importance of GIA in the context of warming.

Regarding the changing relaxation times, we note that this is primarily driven by the evolving location of ice mass loss over time that will occur regardless of the climate models. For example the grounding line retreat in the Amundsen Sea Embayment initially occurs over regions with relatively low mantle viscosity. As retreat progresses further inland, the viscosity increases, leading to a change in bedrock response characteristics. This effect is thus not related to differences between East and West Antarctica, but to the retreat within one basin in which ice loss is expected to occur in the near future regardless of which emission scenario is used. Furthermore, the high emission scenario covers a larger area of ice mass loss, triggering deformation deeper in the mantle where the viscosity is different from the viscosity at shallower depths. This explains the change in relaxation time with timescales for the high-emission scenario which also holds for a different climate forcing used as long as comparable ice thickness changes are induced. This rationale is discussed in lines 311-314 and 321-323 of the manuscript, but we will clarify it further and include it in the conclusions.

2. The authors create a bound of benchmark sea-level rise values based on the 1) difference between 3D-Average and 3D-weaker and 3D-stronger GIA model results and seems to then hand-wavily pick lines that sit between these bounds for the longer period. I am not sure if this is a good metric and a way of choosing recommended

values. For instance, how is relaxation time 200 years in Fig. 4c not a good choice compared to relaxation time 300 in Fig. 4c when the difference between 3D-Average and relaxation time 200 years is as small as the difference between 3D-Average and 3D-weaker? A way around could be to provide a range of recommended values rather than a single value for each bedrock model for different time periods.

We acknowledge that the criteria for selecting the best-fitting relaxation time are not sufficiently clear. For each ELRA simulation, we will quantify deviations from the average response of the two 3D models across all scenarios and climate models to determine the best fit. We will clarify this methodology. We also agree that a range of recommended values is better. We find that a range of 275 to 375 years is suitable and will change the text to recommend 300 years as the optimal choice considering both scenarios and climate models.

3. Section 3: How are the choice of four regions in West Antarctica made, and can this choice be well-justified? Given that ice in North Antarctic Peninsula and Ross Sea Embayment seem particularly insensitive to bedrock response, wouldn't it be more accurate to use just ASE or ASE and WSE to derive the 2D map for LVELRA where most grounded ice mass loss occurs in realistic simulations? Also how is East Antarctica treated where the idealized unloading test is not performed?

The aim was to have a robust relationship between relaxation time and viscosity that is valid for any viscosity map without a priori constraints on where ice loss is exactly taking place. Therefore we aimed to include as much variation in size and location of the ice load and underlying viscosity as possible while balancing computational constraints. Although the four regions did not include a region in East Antarctica, the range in viscosities and relaxation times is very large and includes plausible viscosities in East Antarctica in locations where mass loss might occur. Having an extra loading scenario would add a few points to figure 2b but likely not change the linear relation between viscosity and time scale significantly. This is especially so because we observe reduced sensitivity to viscosity variations at higher values that are more likely in East Antarctica (Fig. 2b).

Can you improve your 2D ELRA map in Victoria Land and Mertz basins in East Antarctica so the difference between the 3D GIA results and 2D ELRA improve?

In those basins, the 3D GIA model results in greater uplift than the 2D ELRA model. The 2D ELRA map might therefore be improved for Victoria Land and Mertz basins by decreasing the relaxation time in those basin. The goal of this study was to provide one relationship between viscosity and relaxation time that works best for the whole of Antarctica, but studying region specific relaxation time alterations would be very interesting. We will add this to the discussion.

Also, how is the 500 m-thickness for ideal loading chosen? Please provide some discussion around how your results would be sensitive to the choice of this ice thickness.

Regarding the choice of 500 m ice thickness for the idealized unloading test, this thickness was selected to approximate stress changes comparable to those expected in realistic ice loss scenarios. The wavelength of the ice load is more influential than its thickness, as per normal mode theory, so we prioritized including sensitivity to loading wavelength by varying the size of loading areas. We will further clarify why wavelength of the load is more important than magnitude and the choice for 500 m in the manuscript.

4. The manuscript will benefit significantly from more extensive literature review, giving credits to those deserve, and being more appropriate/consistent in the citation scheme. Detailed comments are main in the following section.

We appreciate the reviewer's recommendations and will incorporate the suggested references where appropriate to ensure a comprehensive and representative citation scheme.

5. The manuscript could benefit from more careful proofreading for the next round of submission.

We acknowledge the need for careful proofreading and will implement de reviewers' detailed textual comments and ensure that the revised manuscript is polished and free of inconsistencies.

I realize these comments may seem a lot to address, but I hope they are helpful in improving the manuscript and getting it published in the journal. I am wishing the best to the authors.

We would like to thank the reviewer for providing valuable conceptual and detailed feedback.

Specific comments: addressing individual scientific questions/issues.

1. L25-27: The authors attribute the differences across different bedrock models only to the spatial variations in viscosity. But some of the differences arise due to structural differences in theory in each bedrock model. This needs to be clarified in the motivation and discussion. We will clarify this in the introduction and the discussion.

2. The "negative" or "stabilizing" feedback on ice mass loss has first been introduced in the context of "sea level", which includes both bedrock uplift and sea surface height drop. I would recommend the authors to introduce "sea-level feedback" as a general concept, and then explicitly say that they choose to omit the sea surface height component in their investigation; I would like to confirm this, by the way - is it actually true that you only feed the bedrock changes back onto the ice model in the case of using 1D and 3D GIA models, not sea-level change?. If so, you should refer to previous studies that explored the separate impact of deformational and gravitational effects on ice evolution to mention that deformational effects are the dominant portion of sea-level feedback. (e.g. Gomez et

al. 2015, Han et al. 2021, Coulon et al. 2021). We will introduce the term "sea-level feedback" and confirm that only bedrock changes are returned to the ice sheet model. We have performed a sensitivity test for the effect of sea-level changes in van Calcar et al. (preprint 2024) and will repeat the conclusion in this manuscript. Furthermore, we will refer to the suggested references.

3. Provide in detail how sea-level change is calculated. Does it correct for changes grounded ice thickness (or volume) due to bedrock deformation captured in different bedrock models? We will include a detailed description of the sea level change calculation which involves computing the volume above floatation and division by the ocean area.

4. Some statements are introduced as if they are general "facts" when they are not. For example, in Abstract Line 11, there is a sentence as follows: "... accounting for the impact of bed deformation on ice dynamics can reduce predictions of future sea level rise by up to 40% in comparison with scenarios that assume a rigid Earth."  This is not an established fact, but rather depend on strength of climate forcing, sensitivity of the ice-sheet model to climate forcing and bedrock topography change. Therefore, instead of citing a number from a specific study, being general would be more helpful.  We agree that the impact of the GIA feedback is highly variable. We stated the reduction can be up to 40%, implying a range. We will add to the text that the impact is highly variable depending on forcing and Earth structure, but we think that it is useful to keep the upper limit to emphasize the significance GIA might have in projections.

5. L14: "Because modelling the response for a varying viscosity is complex, sea level projections often exclude the Earth's response,..."  This is a very convenient and even a naive way of explaining why ice-sheet models that project the future have been using fixed bedrock, let alone spatially varying viscosity. Until relatively recently, bedrock deformation (or GIA) used to be considered not very important for the "short", centennial time scale ice-sheet evolution, which future projections focus on. This very same comment applies to L49 as well.  We agree and we will emphasize that GIA had until recently not yet been shown to be an important feedback on the time scale of hundreds of years, and therefore was not implemented with a 3D Earth structure.

6. L15: ... or apply a globally constant relaxation time or viscosity." It may not seem clear to many people what the differences you mean here by using the words "constant relaxation time" or "viscosity".  You can make this clearer by specifically referring to the use of simplified bedrock adjustment models such as ELRA or 1D GIA models. We understand the confusion and will move the mention of ELRA and 1D GIA models in lines 19-20 to line 15.

7. L24: "... that deviates less than 40cm from the average of the 3D GIA models… can be further reduced to 10 cm". The specific numbers 40cm & 10cm do not mean much. It would be helpful to express the errors in a relative number, such as percentage.  We will

provide the percentages throughout the whole document.

8. L34: "... requiring thousands of simulations to produce robust projections of potential sea level rise over the coming centuries." Seroussi et al. (2020) goes up until 2100 only. Seroussi et al. (2024) goes up until 2300. The ice model ensemble simulations in Seroussi et al. 2020 & 2024 are on the order of hundreds, not thousands. Maybe just say "ensemble simulations". We will change "thousands of simulations" to "ensemble simulations", and we will refer to the 2024 paper.

9. L34-35: I would recommend citing only the seminal papers that first showed stabilizing sea-level feedback (e.g., Gomez et al., 2010; 2012). Otherwise, you are missing references that also confirm the effects in addition to the ones being cited. We will refer only to the seminal papers for the stabilizing feedback.

10. L47-48: "However, it is currently unfeasible to include a realistic Earth structure in a large ensemble of sea level projections due to the long computation time involved (van Calcar et al., 2023)": This statement could be potentially out of context and misleading. The previous work the first author they cite here simulates the Antarctic Ice Sheet over the last glacial cycle, which is for hundred thousands of years. Here we are talking about the next few hundred years of projections, which would be feasible. Also, it could be appropriate to acknowledge other studies that showed ways to overcome computational infeasibility for long simulations, e.g., de Boer et al., 2014, Han et al., 2022, which are for 1D GIA models and thus could be debatable whether they are "realistic" models or not. Also, there is a study that emulates and thus provides large ensemble for 3D GIA model (Love et al., 2024). We will change the word "realistic" to "3D" in the manuscript to exclude any confusion about this term. We will clarify this sentence by stating that it is unfeasible to include a radially and laterally varying Earth structure in a large ensemble of sea level projections using a dynamic ice sheet model. Furthermore, we will discuss studies to overcome computational infeasibility that can be used to compute bedrock deformation coupled to an ice sheet model for projections, which are Han et al. (2022) and Swierczek-Jereczek et al. (2024).

11. L50-51: ELRA models are not only computationally cheap, but easier to implement because you don't need to "couple" a whole new model that is complex by itself. Also, I believe 1D GIA models are cheap enough to produce large ensembles. The claim that 1D GIA models are too expensive to produce ensembles for centennial time scale runs, needs evidence. We agree with the ease implementation as argument why ELRA models are commonly used and we will include this in the text. We also agree that 1D GIA model can be fast enough to be used in ensemble studies and our manuscript doesn't state that 1D GIA models are too expensive for centennial time runs. On the contrary, we studied which parameters can be used in the 1D coupled model to approximate a 3D coupled model because 1D GIA models are currently being used in ensemble studies. The manuscript already states the following in lines 384-385: "As 1D GIA models can also be considered intermediate in terms of computation time compared to ELRA and 3D GIA" and we will add

here that 1D models are thus suitable to be used in ensemble projections.

12. L79: The authors should cite the work of Jan Swierczek-Jereczek who developed the FastIsostasy model that captures lateral variations in lithosphere thickness and mantle viscosity. It is a critical reference that needs to be acknowledged. We acknowledge that there are different models to simulate bedrock deformation for the glacial cycle, of which FastIsostasy is one, but in this study we compare the currently used methods to include bedrock deformation in ice sheet models in projections, which are a rigid Earth, ELRA and 1D GIA, to the results of using a 3D GIA model. We will discuss this motivation more extensive in the introduction and place the various models in context of available methods by also discussing FastIsostasy . Please see our response to comment 10 for details on how we will discuss Swierczek-Jereczek et al. (2024).

13. L101: it is confusing to now refer ELRA and LVELRA to as "GIA models". We agree and will change it to "Earth model".

14. L153: what is the value of the lower mantle viscosity? We will add that the viscosity of the lower mantle is assumed to be $5*10^{21}$ Pa s.

15. L174: What is Configuration 1 like? How specifically is it different from Configuration 2 rather than the former is "variable resolution". Variable resolution in what? The variable resolution refers to the high resolution area over Antarctica. We understand the current wording in the manuscript is confusing. Configuration 1 is described in lines 149-155. We will add in line 155 that that configuration is referred to as configuration 1 and will remove this from line 174.

16. L179: "… procedure as described for configuration 1.". Nothing has been described for configuration 1. More detail would be helpful. The change suggested in point 15 of this review will clarify what is meant by configuration 1 and the details are described in lines 149-155.

17. L182-183: So, both bedrock deformation and sea surface height changes are calculated. Are you then only passing bedrock deformation back to the ice model? Whichever way, this needs to be made clear. This configuration of the 3D GIA model is used standalone with prescribed ice unloading. This was only mentioned later in the text and we will now clarify that here already.

18. L213: Mitrovica et al. 2011 shows laterally varying flexural rigidity has negligible effects on bedrock deformation? Also, is the citation order following the year of references or the alphabetical order of authors' last names? Mitrovica et al. 2011 state: "our predictions based on 3-D elastic earth models indicate that lateral variations in elastic Earth structure need not be included in future studies". We will make sure the in-text references are in chronological order consistently throughout the paper.

19. L225: How are the "large" and "small" areas chosen based on what metric? The small area is chosen to cover the main area of mass loss close to the present-day grounding line. The large area is chosen to cover the full basin of the embayment, and the full peninsula. We will clarify this in the text.

20. L226: "… such that the resulting empirical relationship between mantle viscosity and relaxation time accounts for the different mantle conditions…". What does it mean by the relationship accounts for the "different mantle conditions"? Isn't the mantle condition the same but the surface load size is being varied? Clarify the sentence. Because the GIA model has a laterally and radially varying structure the mantle conditions differ per region chosen. By having schematic ice loads in various locations, we sample the structure in different ways, obtaining a relationship that is supposedly valid over a large viscosity range. We will clarify this in the text.

21. L228: Do you mean "each surface load" as in surface load on each grid cell or of each basin or each small/large area? I presume the latter but clarify. And what does it mean by "surface load is controlled"? We will change this sentence to: "The resolution of the 2-degree finite element mesh that is used in this configuration of the 3D GIA model is relatively coarse and therefore determines the shape of each area of loading."

22. justification on why this thickness was chosen. Given that the marine-based portion of the West Antarctic Ice Sheet goes up to 2-km thick, and that most of the ice melting will happen in Amundsen Sea Embayment, 500m seems quite arbitrary and out of place. It would be also good to mention how sensitive the viscosity-relaxation time relationship would be to the choice of ice thickness. Please see our answer to general comment 3.

23. L223-224: "This implies that the relaxation time depends on the size of the ice sheet and that a single relaxation time cannot be derived". Clarify this sentence, as the current version is confusing as to what the difference is between "the relaxation time" and "a single relaxation time". Something like, "a single relaxation time" that best represents those integrated over all modes." To clarify, we will change this sentence to: This implies that the relaxation time that is experienced by a certain ice load depends on the size of the ice (un)loading. Thus, a single, uniform relaxation time fixed in time cannot be directly derived.

24. L241: how is "the Earth" different from "the mantle" in this context? Does it make sense to say "the region of the mantle….depends on the sensitivity of the Earth…"? The current sentence sounds like the mantle is sensitive to its own viscosity, which is confusing. To clarify, we will change this sentence to: "However, the actual region in the mantle that dominates the Earth's response depends on the sensitivity of the ice load to the viscosity in the sub-surface, which depends on the viscosity profile itself.

25. L250&251: Explicitly mention that these equations are derived based on your fit shown in Fig. 2b. Also, I think it'd be more helpful to provide a general equation and then say you get different values for the average-viscosity case and the lower-bound viscosity case. This is mentioned in lines 247-249. As suggested by the reviewer, we will combine equations 8 and 9 to the following general equation:

$$\tau = a \cdot 10^{-b} \eta_{\text{eff}}^{\,c},$$

where a is 2.3, b is 5, and c is 0.35 in case the average viscosity used. When the lower bound viscosity is used, a is 3.9, b is 2 and c is 0.20.

26. L255: So which method do you take? High-resolution viscosity profile or smoothing the 2D maps? We will add to the text that we apply smoothing to the 2D maps.

27. L232: Specifically, how many timesteps of how many years? We will add the following to the method section: "Each simulation contains 20 timesteps, of which the first time step is 15 years, increasing by a factor of 1.5 until the largest time step of 33.3 kyr."

28. L224: Table 1 is missing from Supplemental Material. We will add the table to the materials section.

29. L231: Provide more information for the 40 simulations. Within what range of the mentioned parameters are varied? We will change the sentence to: "A total of 40 simulations are conducted, using a grain size of 1, 4 and 10 mm, a water content of 0 and 1000 ppm and a small, medium and large of the region of loading (as shown in Fig. 2a)."

30. L265: Do you mean Supplemental Material not Extended Data? Also, Table 1 is missing. Yes, we mean supplemental materials and we will add the table to the materials.

31. L267: "... 3D-weaker and 3D-stronger Earth models described in section 2.1". Please provide more detailed description on these 3D models in Section 2.1. Lines 140-144 provide an explanation on the 3D Earth structures and we will add the following sentence: The 3D-weaker structure contains a water content of 400 ppm and a grain size of 2.5 mm, and the 3D-stronger structures contains a water content of 200 ppm and a grain size of 4.5 mm.

32. L301-303: It seems like these recommendations would work well for the low-emission scenario, but for the high-emission scenarios these choices seem hand-wavy. For example, how is tau = 300 years any better than 350 in Fig. 4b? More importantly, there needs to be a discussion as to how these recommended values can change when applying climate forcings from different climate models. In a similar context, there needs to be a discussion as to why the recommended values change depending on the period (before 2300 vs. after 2300) in relation to the region of ice melting. It looks like the dominant mass loss happens in West Antarctica until 2300, so relaxation time that best

represents bedrock response in West Antarctica will match better, and after 2300 ice in East Antarctica also goes away, and therefore higher relaxation time that reflects slower bedrock response, resembling East Antarctic response, will work better. Please see the response to general comment 1.

33. L314: It would be helpful to use a relative number (e.g., %) rather than absolute values. This comment applies throughout the manuscript. We will mention relative numbers throughout the manuscript.

34. Fig. 5 & Supp. Figs. 2,4,5: It would be helpful to see some comments regarding the strikingly similar grounding line contours across all GIA models. Also, it would be helpful to show these plots for year 2300 where there will be actual differences in both ice thickness and grounding line extent in West Antarctica. We will explain in the text that the grounding line is similar for most of Antarctica because most of the AIS is stable over time. GIA only has an effect where there is mass loss, which is, for example, mainly in the Amundsen Sea Embayment in the case of the IPSL climate model. Supplemental Figure 3 shows the difference in terms of grounding line position, ice thickness and bedrock elevation between the ELRA300 and the two 3D Earth structures for the year 2300 when using IPSL. We will add the same figure but for the CESM climate model.

35. It looks like most of the differences after 2300 across simulations come from Victoria Land and Mertz in East Antarctica. I'm curious why 2D ELRA models perform equally poorly compared to the 1D GIA and 1D ELRA models? This goes back to the question asked earlier on what kind of treatment is done for deriving relaxation time based on viscosity in East Antarctica given the ideal unloading tests were performed only in West Antarctica. As explained in the response to general comment 1, most of the difference in sea level change after 2300 between different Earth models does not come from East Antarctica but from West Antarctica. This is described in lines 307-311 and we will clarify that the most important effect occurs in West Antarctica. Furthermore, two of the 2D ELRA models perform much better than the uniform ELRA model for some cases, of which the best performing model is 2D Stronger 120 using the relation between the relaxation time and the average viscosity (brown dashed lines in Fig. 6). This is described in lines 364-370 stating "For the high emission scenario, the sea level rise is about 30-40 cm closer to 3D-Average at 2500 using 2D-stronger compared to using ELRA300 (Fig. 6a-b). The advantage of using 2D-stronger over ELRA300 is particularly great in the Amundsen Sea Embayment projections (IPSL) for scenarios longer than 400 years because the difference between 3D-Average and ELRA300 increases strongly after 2300, whereas the difference between 2D-stronger and 3D-Average is constant over time (Fig. 6a)."

Lines 393-396 describe that 1D19 performs better than the best fitting 2D ELRA model: "The largest improvement of 1D19 compared to ELRA300 and 2D-stronger is in the bedrock uplift. The bedrock elevation of 1D19 in 2500 differs by a maximum of 80 meters from the results of the 3D GIA modelling in the high emission scenario, which is significantly smaller

than the difference of 250 m when 2D-stronger is compared with the 3D GIA model output (Supplemental Fig. 5)."

We will the following explanation of why that is: "This improved agreement is explained by the more complete representation of Earth rheology in the 1D GIA model compared to the ELRA approach. While ELRA prescribes a simplified elastic lithosphere and a purely local, exponential relaxation toward isostatic equilibrium, the 1D model captures the full viscoelastic response of the Earth, including both elastic and time-dependent viscous deformation. Although it does not account for lateral variations in Earth structure, the 1D model still resolves the mantle's dynamic flow in response to loading, bringing it closer to the behavior captured in 3D GIA models."

Technical comments:
1. "Approximating ice sheet-bedrock interaction in Antarctic ice sheet projections". I think the title can be improved by being more specific. We will change the title to: "Approximating bedrock deformation of a 3D Earth in an ice sheet model for Antarctic sea-level projections".

2. Antarctic ice sheet => Antarctic Ice Sheet (throughout the manuscript) We will do so.

3. L20: conducted => conduct. (or make the tense consistent throughout) We will do so.

4. L23: sea level rise => sea-level rise We will do so.

5. L56: "A GIA model can include the bedrock response to changes in ice loading…". Remove "can". GIA models by default should include bedrock response.  We will do so.

6. Define GIA earlier in the text. Right now, it's only introduced in L59 after the word has been used many times interchangeably with bedrock response.  We will do so.

7. L84&86: "GIA model using a 3D Earth structure" -> 3D GIA model We will do so.

8. L103: "table 1" => Table 1 We will do so.

9. Table 1: explain the values and units used for the different Earth models. What maximum and minimum values do 3D-stronger and 3D-weaker models have? It would be helpful to describe those rather than referring readers to the other paper, van Calcar et al. (2024).  We will add in the caption that the years described for the uniform ELRA model are the relaxation times and add the different flexural rigidity that we tested. We will also describe in the caption that 1D19 refer to an upper mantle viscosity of $10^{19}$ pa s, etc. We think that it would be useful to show spatial plots of the viscosity at multiple depths from which the maximum and minimum values can be read. We will the add the figure below, which is taken from van Calcar et al. (2024), to the supplemental material.

[Figure]

10. In Figure 1, you have a legend 1DASE, but in the table you have 1D18. We will adjust this.

11. L117: What does it mean by "realistic" sea-level projections? We understand that is can be interpret differently and will remove the word "realistic" from the sentence.

12. L159: Earth structures => Earth structure We will adjust this.

13. L169: Fig 1 => Fig. 1 We will do so.

14. Figure 1 legend: 1D Earth structures => 1D Earth structure profile? We will do so.

15. L214: use consistent significant figures. We will do so.

16. L217: I think it would be beneficial to have a more specific title for this section, something like, "Deriving 2D relaxation time maps from 3D viscosity profiles" We will change the title.

17. L223: "This implies that the relaxation time …..cannot be derived". This sentence feels redundant, as it should be very clear from the sentence before. But this is a minor comment. Please find our response in comment 23.

18. L229: "The uniform thickness of each load is taken to be 500m". There needs to be . The sentence of the reviewer is not finished but we assume the reviewer refers to the same question as comment 3 in the general comments section. Please find our response there.

19. L285: ELRA model with uniform relaxation time? We will adjust the title of this subsection.

20. L286-288: This sentence can be clarified. Strictly speaking, bedrock response is dependent on ice loading and viscosity. You could probably mention climate forcing then affects ice loading changes. We will change this sentence to: "The bedrock response depends indirectly on the climate model because varying ocean warming cause different regions of ice retreat, and the mantle viscosity differs in each region, and the sensitivity to the Earth structure depends on the size load."

21. L290: This whole paragraph is quite confusing and don't seem necessary. It is clear the coupled ice sheet – 3D GIA model results differ from ELRA models (or 1D GIA models), and the difference in the results will vary with climate forcings (which are in turn different for different climate models), which is also clear. This paragraph explains the indirect dependency of the impact of the GIA – ice sheet feedback on climate forcing and emission scenario, which is due to lateral variations in the mantle and due to different size of regions of ice mass loss. This information is necessary to understand the results and we therefore think it is important to keep this paragraph in the text for readers outside of the GIA community.

22. L279: Unindent We will do so.

23. L350: ELRA model with 2D laterally varying relaxation time? We will adjust the subtitle.

24. Figure 6: Some lines go out of bound in the y-axis. To maintain consistency across all figures, we have chosen to use the same y-axis limits, allowing for a more direct comparison between them. As a result, some lines in Fig. 6 extend beyond these limits. However, our primary focus is on the lines closest to y = 0, as they provide the most relevant insights. The lines that exceed the y-axis bounds already show significant deviations from y = 0 early in the simulation, indicating that certain parameter combinations do not perform well. Since these deviations are evident at earlier time steps, the loss of visibility at later stages does not compromise the key interpretation. Expanding the y-axis range to display the full extent of all lines would reduce clarity in the more relevant portions of the figure. Therefore, we have opted to maintain the current limits to prioritize visibility where it is most informative.

25. L380: The references cited here seem quite arbitrary and incorrect (Whitehouse et al., 2019 is a review paper, not an ice-sheet model.) The list of references was indeed incomplete, we will add Klose et al. (2014), Golledge et al. (2015), Gomez et al. (2015), and Rodehacke et al. (2020) to the list, and remove Whitehouse et al. (2019). The reference to Klose et al. (2014) was also missing in the introduction so we will add it there as well.

26. section => Section We will adjust this.

27. configuration => Configuration We will adjust this.

28. L408: accuracy in the metric of ice volume change.  We will adjust this.

29. L409: for two different emission scenarios from two different climate models. The emission scenario is forced on the climate model and does not follow from the climate model. To improve clarity, we will add that we used two climate models and two scenarios and will change it to the following: "for two climate models under two emission scenario's".

References
1.  deBoer et al., ice-sheet–sea-level model: algorithm and applications, GMD, 7, 2141–2156
2.  Love et al., 2024., GMD. A fast surrogate model for 3D Earth glacial isostatic adjustment using Tensorflow (v2.8.0) artificial neural networks
3.  Seroussi et al., 2024 ISMIP6 Antarctica 2300
4.  Han et al., 2021. Modeling northern hemispheric ice sheet dynamics, sea level change and solid earth deformation through the last glacial cycle
5.  Han et al., 2022. Capturing the interactions between ice sheets, sea level and the solid Earth on a range of timescales: a new "time window" algorithm
6.  Jan Swierczek-Jereczek.,2024. GMD. FastIsostasy v1.0 – a regional, accelerated 2D glacial isostatic adjustment (GIA) model accounting for the lateral variability of the solid Earth
7.  Gomez et al., 2015. Sea-level feedback lowers projections of future Antarctic Ice-Sheet mass loss
8.  Gomez et al,. 2012. Evolution of a coupled marine ice sheet- sea level model
9.  Gomez et al., 2010. Sea level as a stabilizing factor for marine-ice-sheet grounding lines.

**References**

Klose, A. K., Coulon, V., Pattyn, F., and Winkelmann, R.: The long-term sea-level commitment from Antarctica, The Cryosphere, 18, 4463–4492, https://doi.org/10.5194/tc-18-4463-2024, 2024.

Golledge, N., Kowalewski, D., Naish, T., Levy, R.H., Fogwill, C.J., and Gasson, E.G.W.: The multi-millennial Antarctic commitment to future sea-level rise. Nature, 526, 421–425, https://doi.org/10.1038/nature15706, 2015.

Gomez, N., Pollard, D., and Holland, D.: Sea-level feedback lowers projections of future Antarctic Ice-Sheet mass loss, Nat. Commun., 6, 8798, https://doi.org/10.1038/ncomms9798, 2015.

Rodehacke, C.B., Pfeiffer, M., Semmler, T., Gurses, O., and Kleiner, T.: Future sea level contribution from Antarctica inferred from CMIP5 model forcing and its dependence on precipitation ansatz, Earth Syst. Dynam., 11(4), 1153–1194, https://doi.org/10.5194/esd-11-1153-2020, 2020.

---

## Referee Report (RR1)

**General comments**

The authors have satisfactorily included some comments and answered some questions that were made in the first round of review. This is appreciated, and I believe that no technical obstacle stands in the way of a publication. Nonetheless, I want to stress that some comments were not addressed, although the authors said they would do it in their answers to the first round of review. This is easy to fix, and I provide a list below. It is however symptomatic of a larger problem. The re-submitted manuscript has many flaws, including:

- Many papers cited in the main text are absent from the reference section.

- The uploaded track change document of the main manuscript is… not a track change document. This substantially complicates a second revision.

- The uploaded pdf of the supplements looks like a track change document, which shouldn't be the case.

- Many hyperlinks for figures and tables are missing.

- A random line break appears at l. 118. Double parenthesis at l. 41. Odd punctuation l. 180 and l. 300.

- The use of the hyphen is very inconsistent. For instance, "ice-sheet model" vs. "ice sheet model"; "sea level rise" vs. "sea-level rise". This was already pointed out in the last review round. It is not limited to these examples and should be checked thoroughly.

- The caption of Tab. 1 and 2 of the supplements reads "RMSE [m]" although the values from the table look like they are unitless.

Reading the document with better care, from the beginning until the end, would easily prevent such oversights, which I believe are not appropriate for a second round of revision. As a reviewer who should mostly focus on the scientific aspect of the paper, it is very frustrating to be forced to comment the form so much.

From the scientific side, I think the authors do not highlight a quite important message that results from their work. ELRA with homogeneous relaxation timescale never fits the transient GIA response, since there is always a drift in the error over time. Of course some parameter choices do better than others, but I feel like the final recommendation should be: "do not use ELRA for AIS projections (even with tau = 300 years!), since (1) a single relaxation timescale just can't fit things correctly as soon as the retreat occurs both in the west and the east, and (2) using LVELRA is equally simple while greatly reducing the error drift".

The authors don't mention, in the introduction, that the deformational responsehas an elastic and a viscous component. I believe this is misleading for the result interpretation because ELRA and LVELRA are tuned to fit the elastic + viscous displacement of the 3D GIA model, although their underlying relaxation equation is only adequate for the viscous response. This should be discussed more extensively, especially because the elastic response is important on such short projection horizons. As you mention in l. 501-504, this is the main reason why

the 1D GIA model gives a better uplift pattern than LVELRA. This points to an obvious limitation of the present work: estimating tau while accounting for the elastic response (simple models exist for that) would give better results and should be mentioned as an outlook.

I wish the best of luck to the authors for the last modifications.

Jan Swierczek-Jereczek

**Unaddressed comments of the previous round of revision**

- L. 90: You said you would mention that Bueler et al. (2007) and Swierczek-Jereczek et al. (2024) capture the dependence on the load wavelength... but you don't do so explicitly. The same applies for the evolving and heterogeneous sea level (that is coupled to deformation) in the case of Coulon et al. (2021) and Swierczek-Jereczek et al. (2024). Furthermore, I don't think you can consider that the errors over a full glacial cycle, presented in Swierczek-Jereczek et al. (2024), are "notable" since (1) the maximal error is lower than with a 1D GIA model and (2) you are not making any comparison to FastIsostasy in the present paper. As I stated in the previous review round, you have such an easy justification for your work, which does not need to invoke the specificities of other GIA models: "going from ELRA to LVELRA is extremely simple in terms of code adaptation". Why not just focus on this appealing aspect of your work to motivate it?

- L. 199: You don't specify that the layers refer to layers where the solid-Earth parameters can vary along the depth, and that this is not the vertical resolution.

- L. 255: Be consistent with your index notation (either with comma or without).

- L. 271: A field tau should either be tau_ij if discrete or tau(x, y) if continuous.

- You don't include the problematic use of a very low viscosity in your 1D GIA model, for instance when representing the global response to ocean load.

- The research questions mentioned in the introduction are inconsistently cited in the conclusion.

**Specific comments**

- L. 65-66, l. 484-485, l. 491-492: Golledge, Rodehacke, Klose and Kachuck use the Lingle-Clark-Bueler approach, which uses a single viscosity as parameter but is not a 1D GIA model. However, the sentences are confusing and should be rephrased.

- L. 132: "Basal melt at the ice shelf is computed using the local Favier quadratic method and the surface mass balance  is computed using a temperature and radiation parametrization (Favier et al., 2019; Berends et al., 2022)." → "Sub-shelf melting follows a quadratic local law (Favier et al., 2019) and the surface mass balance is computed using a temperature and radiation parametrization (Berends et al., 2022)."

- L. 135: Does "ice shelf instabilities" refer to marine ice sheet instability? It would make much more sense in this sentence.

- L. 142: "Besides the stabilising effect of bedrock deformation on ice-sheet evolution, there is also a sea surface height component, and together these comprise the sea-level feedback. The loss of gravitation from the ice sheet causes a local sea-level drop…". This reads quite bad (e.g. there is no "loss of gravitation" but rather a reduced gravitational pull), please rephrase.

- L. 202-204: You already say that above. Please remove redundancy.

- L. 242: "Earth properties are" → "viscous response is"

- L. 302: Isn't the uplift rate a direct output of your model, which computes it more accurately than what you do based on the displacement curve?

- L.360: PISM and Kori use parameterizations for sub-shelf and grounding-line melting that yield similar sensitivities to those used here. I don't think this explains the large sensitivity of IMAU-ICE that you observe.

- L. 427: I would mention earlier that ELRA300 minimizes the RMSE. This way, the reader understands much better why you spend so many lines on describing this specific parameter realization.

- Fig. 5 (and all other figures with the same format): the figure would be much more legible if you would use a different colour bar for displacement and ice thickness anomaly.

---

## Editor Decision (ED1)

Dear Dr. van Calcar, Dear Authors,

Thank you very much for your replies to the reviewer comments. I would now like to ask you to implement the envisaged changes and submit a revised manuscript which will undergo another round of review, hopefully by the same reviewers. I have added some additional remarks below on some of the reviewer comments which might help in drafting the revised manuscript.

Thank you very much and I am looking forward to receiving your revised manuscript.

Best regards,

Johannes Sutter

One comment on your reply to R1:

*b. The large uplift discrepancy of 250 m that you mention at l. 369 has almost no impact on the result. Explaining the little impact of this difference on SL contribution by invoking the fact that it arises relatively late in the simulation is a partially albeit not fully satisfactory explanation to me. I believe the (potentially too) high nonlinearities that are internal to the ice sheet model contribute to it.* Indeed, it is not only the fact that the uplift arises relatively late, but also the fact that the ice sheet is in a state of collapse. Whether this is due to larger sensitivity (non-linearity) or and of the other differences with published estimates ("differentinitialization method, basal melt approximation, and melt parametrization at the grounding line) is difficult to say, but we will extend the description as follows: "However, this uplift occurs mainly in the last 100 years. Furthermore, the effect on grounding line retreat is small because the grounding line is already retreating rapidly and the West Antarctic ice sheet in our simulation is in a phase of collapse. For another ice sheet simulation with different melt approximation or melt parametrization, the sensitivity to a similar uplift might be relatively larger."

I am not sure whether you are addressing the reviewer's comment with the addition: For another ice sheet simulation with different melt approximation or melt parametrization, the sensitivity to a similar uplift might be relatively larger." As the main question here is the non-linear response (ongoing MISI). Variations in melt rate approximations would probably not affect retreat much in case of an unforced self-sustained grl-retreat. R2 has a similar comment mentioning the similarities between grl positions in ASE. A model setup which is close to collapse but hasn't crossed a threshold yet might show a bigger imprint of bedrock response parameterizations.

Regarding the comment of R1:

*5. I know very little about IPSL but CESM is known as a quite high-sensitivity model. I think this is a rather good thing since it leads to ice-sheet configurations that show large differences to present-day, thus exploring more diverse AIS configurations. I think this*

*could be highlighted in a discussion section, especially since it supports the overall
message of the paper. I would very much appreciate more information about the GCM
outputs in the supplemental material: e.g. their present-day bias compared to RACMO,
MAR or ERA5, and their warming pattern (atmospheric and oceanic) by 2300.*

We will add more information the manuscript about the GCMs. What is most relevant
for the Antarctic simulations is the ocean warming. The average ocean temperature
anomaly is colder for CESM than IPSL in the high emission scenario, but warmer in the
low emission scenario (Extended Data Fig. 7 in van Calcar et al., 2024). In the
supplemental materials of this manuscript, we will include a figure showing the average
ocean temperature anomaly by 2300.

I would encourage you to also add a plot of SMB changes as CESM5-8.5 suggests extreme
surface melt starting around 2150 CE and ocean temperature increase is also quite large.

Additionally (R1):

*The authors are showing the ice volume that was lost (including ice grounded
below sea level) and NOT the actual contribution to barystatic sea-level. If this is
the case, this might of course partly resolve the doubts I raise in 2. and 3.*

The ice volume that we show does not include ice grounded below sea level. The
explanation corresponds to option a: The 3 to 4 meters sea level rise not only includes
a large retreat of the Amundsen Sea Embayment, but also significant retreat in the
Wilkes basin and some retreat of the Ross Ice Shelf. Furthermore, there is thinning of
the WAIS. This is shown in Fig. 5 in the manuscript and will be discussed in the results.

It seems that in the projections the ice sheet begins to lose mass from the onset which would
indicate an imprint of the model initialization on projected sea level. I am aware that this is not
the main focus of this manuscript but for context I would invite you to elaborate on the model
initialization and any persisting drift which might affect the results. I agree with the reviewer that
the substantial SLE contributions under the SSP1-2.6 scenario merit discussion including a plot
of the GCM forcing over time (ocean temp, smb, sat). This pertains also to R2's comment to
extend the discussion on scenario uncertainties and how these could affect your suggested
parametric range.

As per the comment by R1:

*Just as for the lateral resolution, I wish to see at least some convergence/refinement
arguments to fully approve a paper that recommends parameters to the community.*

I would encourage you to consider this point and provide a meaningful convergence test for a
selected experiment (i.e. SSP5-8.5).

In your response to the first comment of R2 you mention:

This effect is thus not related to differences between East and West Antarctica, but to the retreat
within one basin in which ice loss is expected to occur in the near future regardless of which
emission scenario is used …

Grl retreat in the Amundsen Sea sector is highly uncertain and the question whether we have crossed a threshold to MISI already is unresolved. Depending on the model, the initialization and the boundary condition you can either surmise stability or instability. I agree with the two reviewers that changing parameters depending on time scales might not be the ideal input for ice sheet modelers but providing a parameter range and associated uncertainties would be a very valuable contribution.

*34. Fig. 5 & Supp. Figs. 2,4,5: It would be helpful to see some comments regarding the strikingly similar grounding line contours across all GIA models. Also, it would be helpful to show these plots for year 2300 where there will be actual differences in both ice thickness and grounding line extent in West Antarctica.* We will explain in the text that the grounding line is similar for most of Antarctica because most of the AIS is stable over time. GIA only has an effect where there is mass loss, which is, for example, mainly in the Amundsen Sea Embayment in the case of the IPSL climate model. Supplemental Figure 3 shows the difference in terms of grounding line position, ice thickness and bedrock elevation between the ELRA300 and the two 3D Earth structures for the year 2300 when using IPSL. We will add the same figure but for the CESM climate model.

I think what the reviewer alludes to here is that the grounding line positions in the different cases are remarkably close to each other even for the retreating Pine Island and Thwaites glacier region. My suspicion is, that this is due to the scenario at hand in which MISI is already under way and the different modes of bedrock response cannot modulate the pace of retreat considerably. The impact of different bedrock response parameterisations is probably more pronounced in model setups close to the tipping point in which one considerably delays retreat or even prevents it and the other favours it due to a delayed bedrock rebound for example.

In response to comment of R2:

*24. L241: how is "the Earth" different from "the mantle" in this context? Does it make sense to say "the region of the mantle….depends on the sensitivity of the Earth…"? The current sentence sounds like the mantle is sensitive to its own viscosity, which is confusing.* To clarify, we will change this sentence to: "However, the actual region in the mantle that dominates the Earth's response depends on the sensitivity of the ice load to the viscosity in the sub-surface, which depends on the viscosity profile itself.

I think your modified sentence could cause confusion as it is unclear what you mean by "*depends on the sensitivity of the ice load*", do you mean "*depends on the sensitivity to ice load changes*"? Apologies if I misunderstand something here.

---

## Author Response (AR2)

**Editor comment:**

Dear Dr. van Calcar,

thanks again for your careful consideration of all reviewer comments in drafting your revised manuscript. I would have an additional note to the fact that the ice sheet model setup seems to represent a highly responsive ice sheet illustrated in the fact that even for the scenario SSP1-2.6 the West Antarctic Ice Sheet collapses. This means (as you have already mentioned in the manuscript) that WAIS is probably undergoing MISI from the get-go of your simulation. This narrows the applicability of your study somewhat which could be reflected in the title and abstract, e.g.: Approximating the moderating role of 3D bedrock deformation in scenarios of West Antarctic Ice Sheet collapse (i'm sure there's a formulation more elegant than this, it's just to illustrate the point). The alternative would be that you include an ice sheet model setup which is relatively stable/showing linear retreat for the strong mitigation scenario. One could imagine that the careful consideration of the bedrock response you exercise here could lead to a bifurcation in the sea level response for weak, intermediate or strong climate forcing scenarios instead of linearly modulating the extent of WAIS and EAIS sea level contributions. I think this could strongly influence your best estimates of relaxation times when calibrating the 1D with the 3D approaches. But this is just speculation. As i am aware that this is computationally expensive and time consuming (and might be a promising topic for a follow up study) the discussion of the fact that you only consider cases where the ice sheet is already collapsing needs to be robust.

With kind regards,
Johannes Sutter

**Response:**

Thank you very much for your careful consideration of our manuscript and for raising this important point. We would like to clarify that the West Antarctic Ice Sheet (WAIS) does not collapse in scenario SSP1-2.6. In this scenario, retreat in WAIS is limited to Pine Island and Thwaites glaciers, while the remainder of WAIS is still intact by 2500. This can be seen in Fig. 5 showing the grounding lines for both 3D Earth structures and the ELRA model with a 300-year relaxation time. A substantial part of the barystatic sea-level contribution in this scenario originates instead from East Antarctica, particularly the Wilkes Basin. This can be seen in the contributions per drainage basin (see figure below, to be added to the supplementary material and discussed in the text as explained below)

We agree that the ice-sheet model applied here is relatively responsive to climate forcing compared to other published models. We have addressed this point in lines 259–266 of the last version of the manuscript. The new figure shows that there are basins with both rapid retreat (e.g. basin 14) and slower retreat (e.g. basins 1 and 18), Thus, these variations across basins provide some insight in whether best relaxation times would change in case of a different ice

sheet response to the climate forcing. The 300 year relaxation time does quite well in all, with a somewhat underestimated GIA feedback in West Antarctica.

Furthermore, we can discuss potential variations in the relaxation time dependent on the sensitivity of the ice model to climate forcing in light of regional differences. The main reason for a different relaxation time is that the ice retreat may occur in a region with a different mantle viscosity. The difference in grounding line retreat in the current simulations between the high and the low emission scenario is very large (no collapse vs collapse) and this has only a small effect on the ideal relaxation time. It is likely that the difference in grounding line retreat between the current simulations of the low emission scenario and a less responsive ice sheet is significantly smaller than the difference between the low and the high emission scenario because the retreat of the low emission scenario only covers a small region of retreat. Within a small region, the Earths viscosity doesn't vary much. We would therefore expect the result for a less responsive ice sheet to stay close to an average relaxation time of 300 years, or an average 1D viscosity of $10^{20}$ Pa s. For this reason, we think that our findings can be applied to a wide range of scenarios and model simulations.

Please find below the detailed changes to the manuscript. We hope these clarifications address your comment.

[Figure]

*Fig. 1: Antarctic drainage systems 1–27 (according to Zwally et al. 2012), grouped into East Antarctica (yellow), West Antarctica (blue), and the Antarctic Peninsula (green) following the basin definition of Zwally et al. (2012). For each basin, the accumulated barystatic sea-level contribution in 2500 is shown for three Earth structures: Rigid (light colors, left bar), 3D weaker (dark colors, middle bar) and ELRA with a relaxation time of 300 years (middle light colors, right bar).*

We add the figure above as supplementary Fig. 2 and add the following to the manuscript

Line 369: "To include uncertainty due to unknown magnitude of retreat, we include a scenario where the West Antarctic Ice Sheet collapses (SSP5-8.5), and a scenario where the Twaithes and Pine Island glaciers retreat significantly whereas the rest of the West Antarctic Ice Sheet is relatively stable (SSP1-2.6). Both scenarios include significant ice mass loss in Wilkes basin in East Antarctica. Together, these simulations capture both rapidly retreating and relatively stable drainage basins across different Antarctic regions."

Line 413: "ELRA300 also performs well when evaluated on the contribution of individual drainage basins to barystatic sea level change, for both fast and slow retreating basins. For example, the drainage basin in Queen Maud Land in East Antarctica contributes significantly to barystatic sea level change, however the impact of GIA is neglectable as the grounding line position is insensitive to bedrock deformation in this ice sheet model (basin 6 in Supplementary Fig. 2). Therefore, the choice of relaxation time becomes arbitrary. Ice loss in the Wilkes basin in East Antarctica also contributes significantly to the barystatic sea level rise but GIA has a large effect in this region because of the relatively low mantle viscosity (basin 14 in Supplementary Fig. 2). ELRA300 provides a very good fit for this basin. In West Antarctica, the contribution differs per basin, but the effect of GIA is significant in almost all basins due the relatively low mantle viscosity at the present-day grounding line of the West Antarctic Ice Sheet (basins 1, 18, 18, 21 and 22 in Supplementary Fig. 2). Here, ELRA300 somewhat underestimates the effect of GIA but still provides a stabilising effect."

Line 583: "Finally, the sea level projections are relatively high compared to literature (Seroussi et al., 2024). A different calibration of the ice sheet model, or a completely different ice sheet model could lead to lower projections of sea level contribution. We include a scenario with fast retreat leading to a collapse of the West Antarctic Ice Sheet and a scenario with slower retreat without a collapse. The difference in grounding-line retreat between these scenarios means the ice sheet is sensitive to a somewhat different part of the mantle, which leads to a small difference in preferred relaxation time. If the low-emission scenario were to produce less grounding-line retreat than in our current simulations, the region over which ice mass loss occurs cannot be very different from the low emission case. Hence, the preferred ELRA and 1D GIA models are also expected to remain applicable."

References:

Zwally, H. J., Giovinetto, M. B., Beckley, M. A., & Saba, J. L. (2012). Antarctic and Greenland drainage systems. *GSFC Cryospheric Sciences Laboratory*, *265*.

---

## Author Response (AR3)

We would like to thank both the reviewers and the editor for their valuable feedback. Please find our response in blue.

**Editor report**

Dear Dr. van Calcar,

we have now received two reviews and both reviewers in principle suggest publication providing additional contextualization of your results. The revised manuscript will not be sent out to reviewers again but only undergo review by the editor.

Apart from the reviewer comments on the discussion of the model results there remain stylistic and formatting issues (see both reviewer's comments).

Currently, your manuscript and abstract do not clearly state that your experiments only consider ice sheet model projections where MISI is already under way. Thus, generalization of your conclusions to other modes of ice sheet response (e.g. linear trend, equilibrium) remains difficult. I think it is important for the reader to understand this from the get-go (including the abstract).

We agree that this can be stated more explicit in the text. We add to the abstract in lines 23-24: "Using a rigid Earth model, this results in ~3-7.5 m of barystatic sea level rise with significant retreat in various basins due to marine ice sheet instability."

In the beginning of the result section at line 432, we added: "Depending on the emission scenario and climate model, we project ~3-7.5 m of barystatic sea level rise with significant retreat in various basins due to marine ice sheet instability."

We add the following in the conclusion starting in line 730: "If a low emission pathway or a more muted dynamical response, for example a situation in which MISI is weak or does not progress substantially, were to lead to only limited grounding line retreat compared with our simulations, the influence of solid Earth deformation in that region would likely be minor, and the choice of Earth structure would have little effect on the results. Hence, the preferred ELRA and 1D GIA models are also expected to remain applicable.

While you added paragraphs (l367 - 371 & 571 - 578) discussing MISI I do not see "a scenario with slower retreat without a collapse" (l573). Figure 4 shows the experiments with ongoing MISI and the supplements don't include the no-collapse case either. Maybe a change in the wording alongside quantification of grounding line retreat paint a clearer picture compared to general terms such as "collapse" vs. "slower retreat".

We define a collapse as the loss of most grounded ice in west Antarctica, including Amundsen Sea Embayment, Ross Sea sector and Weddell Sea sector, following Bamber et al. (2009). In our simulations of the low emission scenario, retreat in WAIS is limited to Pine Island and Thwaites glaciers, while the remainder of WAIS is still intact by 2500. This can be seen in Fig. 5. We therefore state in lines 729-730 that the West Antarctic Ice Sheet (WAIS) does not collapse in scenario SSP1-2.6 in our simulations and refer to Fig. 5. Even though there is ongoing MISI, there is no collapse of WAIS. Figure 4 does show a significant sea level contribution for the low emission scenario. This contribution comes from significant retreat of Thwaites and Wilkes glaciers and not from a collapse of the WAIS.

We made multiple changes in the manuscript to be more explicit about this.

To be more explicit about what is meant by "collapse", we changed the sentence starting in line 444 and show in green what we added: "To estimate the uncertainty associated with the magnitude of retreat, we include a scenario where the West Antarctic Ice Sheet collapses, meaning that most of the grounded ice has been lost (SSP5-8.5), and a scenario where the Thwaites and Pine Island glaciers retreat significantly whereas the rest of the West Antarctic Ice Sheet is relatively stable (SSP1-2.6)."

We removed the word "collapse" from the results section in line 585 and change the sentence to: "Furthermore, the effect on grounding line retreat is small because the grounding line is already retreating rapidly and the negative feedback from bedrock uplift is not strong enough to slow the rate of retreat."

We removed "faster retreat" and "slower retreat" from the sentence starting in line 728 and only mention the collapse, now that it is more clearly defined.

Furthermore, we changed the sentence starting in line 493 to: "For SSP1-2.6-IPSL, significant ice mass loss in the West Antarctic Ice Sheet only occurs in the Amundsen Sea Embayment, where grounding line retreat in ELRA300 is 150 km greater than 3D-weaker by 2500 (Fig. 5)."

We add references to the figures in the conclusion section in lines 733-734: "We include a scenario with fast retreat leading to a collapse of the West Antarctic Ice Sheet in 2500 (Supplementary Fig. 3, 7 and 8), and a scenario with slower retreat which does not lead to collapse (Fig. 5)."

Sentences such as l575-577 "If the low-emission scenario were to produce less grounding-line retreat than in our current simulations, the region over which ice mass loss occurs cannot be very different from the low emission case" are unclear and not

sufficient to support your statement "Hence, the preferred ELRA and 1D GIA models are also expected to remain applicable."

*We agree that that sentence is unclear. Please see the response to your first comment where we changed these sentences to clarify this.*

All figures in the supplement show ice sheet configurations with substantial grounding line retreat of Thwaites and PIG and Figure 4 does not seem to include a scenario without ongoing MISI judging from the sea level contributions displayed. Please add the SSP1-2.6 experiments without collapse to the timeseries figures (at least Fig. 4). So far Fig. 5 seems to be the only figure showing results from SSP1-2.6 forcing albeit with ongoing MISI in WAIS and Wilkes.

*The SSP1-2.6 scenario is included in all the time series: Fig. 4 a and d (solid lines), Fig. 4c and f, Fig. 6c-d, Fig. 7c-d.*

*References:*

*Bamber et al. Reassessment of the Potential Sea-Level Rise from a Collapse of the West Antarctic Ice Sheet. Science, 324, 901-903 (2009).*

**Report 1**

The paper by van Calcar et al. discusses how simple assumptions made in the Earth model can be used for ice-sheet model projections for Antarctica. The paper is well written (for most sections), and the authors have taken into account the comments and questions raised by the two previous reviewers. The authors explain their choice of parameters in detail and also the limitations of their results in terms of the ice-sheet model assumptions.

I have only a few smaller, more technical, comments that would increase the readability.
1) The usage of a hyphen is not consistent in the manuscript.

*We have changed this throughout the manuscript.*
2) Line 63: Does that mean your GIA model does not solve the sea-level equation?

*Thank you for pointing this out. We used two different sets of settings for the GIA model, which we refer to as configuration 1 and 2. We did state the use for configuration 2 of the GIA model but not for configuration 1 of the GIA model. We now indicate this specifically in the method section and add a reference where the effect of solving the sea-level equation on sea-level projections is discussed in detail. We change the*

sentence in lines 205-208 to: "The model includes material compressibility but it does not solve the sea-level equation, and it does not account for rotational feedback or the migration of coastlines because these have a relatively minor effect on sea-level change compared with the effect of changes in grounded ice thickness (Milne et al., 1999; van Calcar et al., 2025)."

3) Line 68: Does the statement in this line mean that an upper mantle viscosity of 10^21 Pa s relates to a relaxation time of 3000 years? If so, it would be good to mention the viscosity already in the paragraph before, around line 56.

Yes, one could state that and we add in green in line 58:

Typically, ELRA is used with a uniform relaxation time of 3000 years and a flexural rigidity of $10^{25}$ kg m$^2$ s$^{-2}$, which roughly correspond to a mantle viscosity of $10^{21}$ Pa·s and a lithospheric thickness of 100 km, respectively.

4) Lines 94-95: Do both GIA models not include the sea-level equation? Your description in lines 102/103 sound like the sea-level equation is included.

We are using only one GIA model but with two different configurations. The configuration referred to in lines 94-95 in the previous manuscript does not solve the sea-level equation. We explain this now in the method section, following comment 2 of this review. The other configuration used to derive the LVELRA maps does solve the sea-level equation, which is mentioned in line 298. Since solving the sea-level equation in the GIA model has a relatively small effect on the resulting barystatic sea-level projections (van Calcar et al., 2025), we prefer to describe the solving of the sea-level equation in different configurations in the method section. To improve clarity in the introduction, we did add "barystatic" in front of "sea-level rise" in the lines 120, 133 and 135 to be explicit that this is the barystatic sea-level rise resulting from the ice-sheet model and not regional sea level from the GIA model.

5) Maybe good to mention LVELRA in point 2 as well, because this is the type of model you are using there, right?

We add LVELRA in brackets to point 2.

6) Lines 203/204: This is a repetition. You wrote the same thing in lines 168-170.

Thank you for pointing this out. We have removed the repetition and added the viscosity of the lower mantle in line 216.

7) Line 270ff: I'm not sure that I understand this right. I thought you used a 3D GIA model to derive a relation between relaxation times and viscosities that can be used to find laterally varying relaxation times. Or is this paragraph about the lithospheric thickness only? Please make this clearer.

This paragraph is about the adjustment of the ELRA model to be able to include a laterally varying relaxation time (which is then later derived from the 3D GIA model in section 3). To make this clear, we change line 331 to: "To include laterally varying

relaxation times (derived in section 3) in the ELRA model, we made the relaxation time in Eq.5 a function of the 2D grid coordinates, such that $\tau$ becomes $\tau(i,j)$."

8) Lines 277-280: This should come earlier, preferably in the Introduction.

We agree and move the whole paragraph to line 91 in the introduction.

9) Line 359: Which simulation now? Just ice-sheet model simulation or coupled with a 3D GIA model?

We add in lines 428-429 that this is the coupled model.

10) Section 4: Please mention in the beginning that your sub-sections focus on the comparison to the 3D coupled model. Sub-section head 4.1 gives the reader the impression that you talk about the results for just this model set-up, but you also discuss the difference to the 3D coupled model set-up.

We adjusted the first paragraph of section 4 to be more explicit about which model is used:

"Sea-level rise over the next 500 years is projected using two different climate models, each under a high and a low emissions scenario. Projections using the ice-sheet model coupled with simple Earth models that adopt a uniform relaxation time, a laterally variable relaxation time, and a 1D Earth structure are compared to the average sea-level rise obtained using the coupled ice-sheet – 3D GIA model (configuration 1) with 3D-weaker and 3D-stronger Earth structures. The average barystatic sea-level rise computed by the ice-sheet model coupled to the 3D GIA model using the two different 3D Earth structures is referred to as 3D-Average."

11) Section 4.1 is a bit un-structured. You shortly discuss the results for ELRA3000 but then move to ELRA300 and discuss these results in detail. Later on, you mention why you do this, because ELRA300 is the best approximation for the 3D coupled model set-up. This information needs to come much earlier. You did this much better in the sub-sections 4.2 and 4.3.

We have moved the paragraph about the optimal choice of relaxation time straight after mentioning the research question in the beginning of the section.

12) Figure 7: Why not adding the LVELRA best case here as well?

We agree that this would be more consistent and added LVELRA best case to the figure and adjusted the caption accordingly.

**Report 2**

**General comments**

The authors have satisfactorily included some comments and answered some questions that were made in the first round of review. This is appreciated, and I believe that no technical obstacle stands in the way of a publication. Nonetheless, I want to stress that some comments were not addressed, although the authors said they would do it in their answers to the first round of review. This is easy to fix, and I provide a list below. It is however symptomatic of a larger problem. The re-submitted manuscript has many flaws, including:

- Many papers cited in the main text are absent from the reference section.
  We have checked all the references and added missing references of Weerdesteijn et al. (2023), Nield et al. (2018) and Swierczek-Jereczek et al. (2024) to the reference list.

- The uploaded track change document of the main manuscript is... not a track change document. This substantially complicates a second revision.
  We regret to read that the reviewer could not find the documents with tracked changes. There were two documents including all tracked changes, one with tracked changed for the reviewers and one with tracked changes for the editor comment. Before the manuscript was send to the reviewers, we confirmed with the journal that the reviewers would have access to both documents.

- The uploaded pdf of the supplements looks like a track change document, which shouldn't be the case.
  We have uploaded the tracked changes in the supplementary materials because these were important changes regarding some comments of the reviewers. The final upload will not have tracked changes.

- Many hyperlinks for figures and tables are missing.
  We found no statement of the journal requiring or recommending hyperlinks for figures and tables so we did not apply hyperlinks in the manuscript.

- A random line break appears at l. 118. Double parenthesis at l. 41. Odd punctuation l. 180 and l. 300.
  We fixed these mistakes.

- The use of the hyphen is very inconsistent. For instance, "ice-sheet model" vs. "ice sheet model"; "sea level rise" vs. "sea-level rise". This was already pointed out in the last review round. It is not limited to these examples and should be checked thoroughly.
  We have changed this throughout the manuscript.

- The caption of Tab. 1 and 2 of the supplements reads "RMSE [m]" although the values from the table look like they are unitless.
  We confirm that the caption is correct. The unit is in meter and not unitless.

Reading the document with better care, from the beginning until the end, would easily prevent such oversights, which I believe are not appropriate for a second round of revision. As a reviewer who should mostly focus on the scientific aspect of the paper, it is very frustrating to be forced to comment the form so much.

From the scientific side, I think the authors do not highlight a quite important message that results from their work. ELRA with homogeneous relaxation timescale never fits the transient GIA response, since there is always a drift in the error over time. Of course some parameter choices do better than others, but I feel like the final recommendation should be: "do not use ELRA for AIS projections (even with tau = 300 years!), since (1) a single relaxation timescale just can't fit things correctly as soon as the retreat occurs both in the west and the east, and (2) using LVELRA is equally simple while greatly reducing the error drift".

We agree that LVELRA would be preferred over ELRA300 but we would also like to provide a recommendation for a uniform relaxation time for ice sheet modellers that do not want to include LVELRA.

We add the following to the conclusion section:

Line 667: "If the ELRA model with a uniform relaxation time were to be used, we …"

Line 687: "We stress that the relatively high root mean square error of ELRA with a uniform relaxation time can be significantly reduced by using LVELRA and 1D GIA models, which are the preferred models."

The authors don't mention, in the introduction, that the deformational responsehas an elastic and a viscous component. I believe this is misleading for the result interpretation because ELRA and LVELRA are tuned to fit the elastic + viscous displacement of the 3D GIA model, although their underlying relaxation equation is only adequate for the viscous response. This should be discussed more extensively, especially because the elastic response is important on such short projection horizons. As you mention in l. 501-504, this is the main reason why the 1D GIA model gives a better uplift pattern than LVELRA. This points to an obvious limitation of the present work: estimating tau while accounting for the elastic response (simple models exist for that) would give better results and should be mentioned as an outlook.

We agree that this could be stressed more clearly in the text. We add in line 61: "Furthermore, ELRA includes the flexural elastic response of the lithosphere, but it neglects the elastic part of the viscoelastic response."

Line 518: "Furthermore, the elastic response of the upper mantle is not taken into account in the ELRA model, which could lead to an underestimation of uplift compared to the viscoelastic mantle response in the 3D model."

We extended the description of the effect of flexural rigidity and add explicitly how elastic behaviour is taken into account, starting in line 525: "In the ELRA model, the elastic response of the lithosphere is computed using the flexural rigidity of the lithosphere. The lithospheric beneath the West Antarctic Ice Sheet can be as thin as tens of kilometers (Lloyd et al., 2019). We therefore test the impact of using a flexural rigidity of $1.92 \cdot 10^{24}$ km$\cdot$m$^2$/s$^2$, which roughly corresponds to a lithospheric thickness of 60 km. The combination of a lower flexural rigidity and higher relaxation time yields a similar result to the combination of a higher flexural rigidity and somewhat lower relaxation time. Therefore, decreasing the lithospheric thickness does not improve the fit of ELRA to the 3D Average (Supplementary Fig. 6)."

I wish the best of luck to the authors for the last modifications.

Jan Swierczek-Jereczek

**Unaddressed comments of the previous round of revision**

- L. 90: You said you would mention that Bueler et al. (2007) and Swierczek-Jereczek et al. (2024) capture the dependence on the load wavelength… but you don't do so explicitly. The same applies for the evolving and heterogeneous sea level (that is coupled to deformation) in the case of Coulon et al. (2021) and Swierczek-Jereczek et al. (2024).
  We acknowledged the advantages of the model presented in Swierczek-Jereczek et al. (2024) over ELRA in the text in line 112. We have added in green the description of the model: "A computationally efficient Earth model based on fast Fourier transforms has been developed that approximates lateral variations in mantle viscosity and lithospheric thickness in the Earth structure and takes into account the effect of a spatially and time varying sea level on deformation (Swierczek-Jereczek et al., 2024)."

  We have added that the dependence on the wavelength of the load is one of the advantages of FastIsostasy over ELRA and added in line 114: "Coulon et al. (2021) coupled ELRA with a gravitationally consistent geoid calculation so compute near-field relative sea-level changes. Furthermore, ELRA uses a single relaxation time and is therefore independent of load wavelength, but the framework could in principle be extended to a scale-dependent formulation where the relaxation time becomes a function of wavenumber."
  Furthermore, I don't think you can consider that the errors over a full glacial cycle, presented in Swierczek-Jereczek et al. (2024), are "notable" since (1) the maximal error is lower than with a 1D GIA model and (2) you are not making any comparison to FastIsostasy in the present paper.
  Concerning point (1), Fastisostasy has lower errors than a 1D GIA model for part of the glacial cycle, but for other parts of the cycle, the 1D GIA model has lower

erros. Our manuscript states in lines 114-115 that the error of FastIsostasy is notable compared to the 3D GIA model, which is still valid. Concerning point (2), we explicitly state that the error relates to modelling a glacial cycle rather than projections (line 113) because FastIsostasy have not been used in projections yet.

As I stated in the previous review round, you have such an easy justification for your work, which does not need to invoke the specificies of other GIA models: "going from ELRA to LVELRA is extremely simple in terms of code adaptation". Why not just focus on this appealing aspect of your work to motivate it?

The reviewer stated in the first round that the description of existing models with lateral variability needs more nuance and context and we agree that it is a valuable addition to discuss efforts to overcome computational infeasibility. We therefore implemented the suggestions in round 1 of the reviewer themselves to not only focus on ELRA and LVELRA in the introduction, but to discuss also other models (Swierczek-Jereczek et al., 2024; Nield et al., 2018; Book et al., 2022; Weerdesteijn et al., 2023). We still highlight the simple adaptation of LVELRA in multiple places in the manuscript, for example in the conclusion section in line 703.

- L. 199: You don't specify that the layers refer to layers where the solid-Earth parameters can vary along the depth, and that this is not the vertical resolution.

  We mention in the text: "A high resolution area is defined over Antarctica with a horizontal and vertical grid resolution of 30 km wide and deep between the surface and 670 km depth."

- L. 255: Be consistent with your index notation (either with comma or without).
  We added a comma.

- L. 271: A field tau should either be $tau_{ij}$ if discrete or $tau(x, y)$ if continuous.
  Since tau is derived from a spatially varying viscosity, we chose tau(x,y). However, for this context mentioning the 2D grid we switched to $tau\_ij$.

- You don't include the problematic use of a very low viscosity in your 1D GIA model, for instance when representing the global response to ocean load.
  This is already stated in lines 724-726: "Furthermore, using the suggested upper mantle viscosity would lead to an overestimation of the response to changes in global ocean loading and to changes in ice loading in East Antarctica over millennial timescales."

- The research questions mentioned in the introduction are inconsistently cited in the conclusion.
  We updated all research questions to the correct version.

**Specific comments**

- L. 65-66, l. 484-485, l. 491-492: Golledge, Rodehacke, Klose and Kachuck use the Lingle-Clark-Bueler approach, which uses a single viscosity as parameter but is not a 1D GIA model. However, the sentences are confusing and should be rephrased.
  Thank you for pointing this out. We have now separated these two types of models as "1D GIA model" and "bed deformation model with a viscoelastic half-space using a homogeneous upper mantle viscosity" in lines 75 and 605-614 (corresponding to the parts of the text previously in lines L. 65-66, l. 484-485, l. 491-492).

- L. 132: "Basal melt at the ice shelf is computed using the local Favier quadratic method and the surface mass balance is computed using a temperature and radiation parametrization (Favier et al., 2019; Berends et al., 2022)." → "Sub-shelf melting follows a quadratic local law (Favier et al., 2019) and the surface mass balance is computed using a temperature and radiation parametrization (Berends et al., 2022)."
  We switched the location of the reference and changed the name of the basal sliding law.

- L. 135: Does "ice shelf instabilities" refer to marine ice sheet instability? It would make much more sense in this sentence.
  We changed "shelf" to "sheet".

- L. 142: "Besides the stabilising effect of bedrock deformation on ice-sheet evolution, there is also a sea surface height component, and together these comprise the sea-level feedback. The loss of gravitation from the ice sheet causes a local sea-level drop...". This reads quite bad (e.g. there is no "loss of gravitation" but rather a reduced gravitational pull), please rephrase.
  Rephrased according to your suggestion.

- L. 202-204: You already say that above. Please remove redundancy.
  We have removed the duplicated information.

- L. 242: "Earth properties are" → "viscous response is"
  We changed it to "bedrock response" since the full bedrock response is simplified to the ELRA approximation and not just the viscous response.

- L. 302: Isn't the uplift rate a direct output of your model, which computes it more accurately than what you do based on the displacement curve?
  The displacement curve follows directly from the uplift rate output by the model. To make this more explicit, we change the sentence in lines 353-354 to: "For each simulation, the resulting displacement over time for each surface load/Earth model combination is computed, yielding a displacement curve."

- L.360: PISM and Kori use parameterizations for sub-shelf and grounding-line melting that yield similar sensitivities to those used here. I don't think this explains the large sensitivity of IMAU-ICE that you observe.

  Different initialisations yield distinct ice thickness fields, grounding-line configurations, and stress regimes. Basal melt and grounding-line parameterisations depend on local ice thickness, shelf geometry, and basal traction. Different initialisations place the model in different parts of this parameter space, so the same perturbation (e.g., a given melt or grounding-line parametrisation) can produce different dynamical outcomes. Because basal-melt and grounding-line parameterisations operate on these background states, identical parameter choices between models can induce different perturbations to melt, buttressing and grounding-line migration.

- L. 427: I would mention earlier that ELRA300 minimizes the RMSE. This way, the reader understands much better why you spend so many lines on describing this specific parameter realization.

  We have moved the paragraph about the optimal choice of relaxation time to straight after mentioning the research question in the beginning of the section.

- Fig. 5 (and all other figures with the same format): the figure would be much more legible if you would use a different colour bar for displacement and ice thickness anomaly.

  We understand the reviewer's point and would prefer different colormaps for ice thickness and bedrock elevation as well. However, we also wanted to keep the grounding line in the same color on both the ice thickness and bedrock elevation maps, and we could not find a second colormap in which the grounding lines are distinguishable and colorblind proof. The colors of the grounding line are now also consistent with the colors of the time series (3D stronger is pink, 3D weaker is orange and ELRA300 is green). We therefore chose to use the same colormap.

---

## Author Response (AR4)

Review by Editor : Approximating 3D bedrock deformation in an Antarctic ice-sheet model for sea-level projections; van Calcar et al.

We sincerely thank the editor for the constructive comments. Please find our response in blue.

L115 I suggest to remove "and it shows notable discrepancies compared to the 3D GIA model it was benchmarked against" as such a statement would require a more robust analysis of these discrepancies. I assume all models of reduced complexity will lead to notable discrepancies compared to 3D GIA over long timescales.
We have followed this suggestion.

L115-116 "so compute" -> "computing"
We adjusted this.

L169 "a quadratic local law" -> "a quadratic local parameterization"
We adjusted this.

L171 I suggest to change "The model does simulate marine ice sheet instabilities, but not marine ice cliff instabilities" to "The model does not include a marine ice cliff instability parameterization". MICI involves an explicit parameterization while MISI usually emerges as a result of the flow law, bedrock geometry etc.
We adjusted this.

L443: Regarding your response to the reviewer comment "L.360: PISM and Kori use parameterizations for sub-shelf and grounding-line melting that yield similar sensitivities to those used here. I don't think this explains the large sensitivity of IMAU-ICE that you observe."
Your response highlights the impact of initialization, which, as you correctly state, would lead to different basal melt rates even for the same basal melt parameterization. However, your explanation covers the imprint of initialization not the melt parameterization itself. If you want to stick to this response then you would have to remove the part of the sentence stating : "the basal melt scheme, and the melt parametrization at the grounding line".

Indeed, the main impact is the initialisation and we removed "the basal melt scheme, and the melt parametrization at the grounding line" from the sentence.

So far, your paragraph explaining the high SLE in your simulation is very general. There is no information on your initial model state expect that present bedrock topography is used, so the reader cannot evaluate this aspect of your work. Did you e.g. use inversion or a freely evolving spinup? I would suggest either explicitly stating that you did not

investigate the reasons for the high model sensitivity (and that it is not a focus of this paper) or providing a more elaborate analysis in the supplements by providing a figure of the initial model state (e.g. thickness differences to present day observations, model drift). If you use an initial state from a previous publication, then it would be helpful to state this somewhere in the methods. If not it, I would suggest adding a short paragraph on the model initialization and a figure showing the present day initial state in the supplements.

We added text in lines 371-372: "We did not investigate the sensitivity of IMAU-ICE to the initialisation and ice sheet model parameters as it is not the focus of this study".

Irrespective of this, the explanation on the initial state of the ice sheet was indeed missing, we apologize for that. We used the initial state from a previous publication so we have added the following to the text in lines 148-151:

"The basal friction and sub-shelf ocean temperature are calibrated using an inversion procedure over a period of 10000 years to obtain ice sheet velocities in equilibrium with the present-day bedrock topography taken from Bedmachine version 3 (Morlighem et al., 2020). The calibration is discussed in detail in van Calcar et al. (2025). The present-day ice surface topography and grounding line position follow from this calibration."

L448-450: the forcing e.g. in the Amundsen Sea used in your study could be compared to the forcings used e.g. in ISMIP6. This way you could either remove the sentence "Other climate models provide significantly different thermal forcings in terms of magnitude and spatial distribution, which might not be captured by the two climate models used in this study" or explicitly state the quantitative differences in the thermal forcing and whether they lead to the high SLE contributions in your simulations. I suggest either quantifying the differences in your forcing (e.g. the thermal forcing in our simulations is substantially higher compared to ..) or remove the sentence.

We agree that this sentence could be removed and did so.

L 450-455 I appreciate the modifications to this sentence, but I do not think you can state "retreat significantly whereas the rest of the West Antarctic Ice Sheet is relatively stable". Figure 4 shows a continued ice loss until the end of your simulation, i.e. ongoing retreat. Chances are that the remaining parts of the WAIS in SSP1-2.6 will melt as well over time would the simulation(s) run longer. My point being, that stability cannot be inferred from your figures.

Thank you for clarifying your point. We agree with you and acknowledge the importance of mentioning the time scale. We therefore changed the sentence you are referring to in line 375 to: "… and a scenario where the Thwaites and Pine Island glaciers retreat significantly whereas the retreat of the rest of the West Antarctic Ice Sheet is relatively

small.". Additionally, we clarify your point by adding the following sentence at the end of the paragraph in lines 377-378: "Since there is ongoing ice mass loss at 2500, the West Antarctic Ice Sheet might collapse even in this scenario on longer time scales."

L457 "Together, these simulations capture both rapidly retreating and relatively stable drainage basins across different Antarctic regions." I think this sentence is valid for many simulations as there will be always relatively stable drainage basins at least for the forcings and timescales considered here. The relevant basins here are however the major marine drainage basins which are associated with MISI. Those are all retreating or collapsing in your model ensemble for both chosen forcings.

We see your point and removed this sentence since we already give a detailed explanation of the important differences between the scenarios and basins in the sentences before.

L626 "A widely used mantle viscosity for a 1D Earth model is 1021 Pa·s (hereafter referred to as 1D21) (Gomez et al., 2015; Konrad et al., 2015; Rodehacke et al., 2020; Golledge et al., 2019)"
I am not sure whether the references are fitting here. E.g. Konrad et al 2015 don't use a single viscosity value, Golledge et al 2015 don't mention the viscosity they use, but they indeed use 10^21 in previous publications. Gomez et al use a range of viscosities. So, strictly, none of the references show that 1D21 is used widely. Some model studies make an effort to treat this viscosity as a free tuning parameter. I would suggest either referencing a study showing that 1D21 is widely used or referencing studies which explicitly use 1D21.

Thank you for pointing this out. To clarify, we have changed the following sentences in lines 67-73 in the introduction:

"Some current existing ice sheet projections that are derived in conjunction with a 1D GIA model or bed deformation model with a viscoelastic half-space use a homogeneous upper mantle viscosity of $10^{21}$ Pa·s (Rodehacke et al., 2020; Golledge et al., 2015; Klose et al., 2024). However, using a relatively high viscosity value, or a relaxation time of 3000 years, does not affect sea-level rise projections significantly compared with excluding bedrock deformation entirely, and it overestimates sea-level rise by up to 20% by the year 2500 compared with projections that use GIA models that consider a lower 1D viscosity (Konrad et al., 2015; Gomez et al., 2015) or a 3D Earth structure, which we refer to as 3D GIA models (van Calcar et al., 2025)."

We removed the sentence you refer to from the results section to prevent receptiveness.

Fig 5: It is good that you try to be consistent with the color-code for the grounding line

and the timeseries plots but maybe cmaps such as cm_crameri.broc or cmocean.cm.tarn allow for readability of both grounding line and cmap. However, as one reviewer pointed out, it could ease interpretation and readability if you would use a different colormap for bedrock response and ice thickness changes. Also, currently only thickness changes of grounded ice at the end of the respective simulation are shown while the majority of the bedrock response is due to the thickness change over all areas (i.e. including previously ice covered regions).

Thank you for the suggestions and we have now changed the colorscheme of all the figures showing changes in bedrock elevation. Figures S4 and S5 show the ice thickness and bedrock elevation difference in the year 2300.